# Human *OPRM1* and murine *Oprm1* promoter driven viral constructs for genetic access to μ-opioidergic cell types

Gregory J. Salimando[1,2], Sébastien Tremblay [1,2], Blake A. Kimmey[1,2], Jia Li[3], Sophie A. Rogers[1,2], Jessica A. Wojick[1,2], Nora M. McCall[1,2], Lisa M. Wooldridge[1,2], Amrith Rodrigues[4], Tito Borner[1,5], Kristin L. Gardiner[6], Selwyn S. Jayakar[7], Ilyas Singeç [8], Clifford J. Woolf [7], Matthew R. Hayes [1,5], Bart C. De Jonghe[1,5], F. Christian Bennett [1,9], Mariko L. Bennett[9], Julie A. Blendy [10], Michael L. Platt[1,2], Kate Townsend Creasy [4,5], William R. Renthal[3], Charu Ramakrishnan [11], Karl Deisseroth [11,12,13,14] ✉ & Gregory Corder[1,2] ✉

With concurrent global epidemics of chronic pain and opioid use disorders, there is a critical need to identify, target and manipulate specific cell populations expressing the mu-opioid receptor (MOR). However, available tools and transgenic models for gaining long-term genetic access to MOR+ neural cell types and circuits involved in modulating pain, analgesia and addiction across species are limited. To address this, we developed a catalog of MOR promoter (*MORp*) based constructs packaged into adeno-associated viral vectors that drive transgene expression in MOR+ cells. *MORp* constructs designed from promoter regions upstream of the mouse *Oprm1* gene (*mMORp*) were validated for transduction efficiency and selectivity in endogenous MOR+ neurons in the brain, spinal cord, and periphery of mice, with additional studies revealing robust expression in rats, shrews, and human induced pluripotent stem cell (iPSC)-derived nociceptors. The use of *mMORp* for in vivo fiber photometry, behavioral chemogenetics, and intersectional genetic strategies is also demonstrated. Lastly, a human designed *MORp* (*hMORp*) efficiently transduced macaque cortical *OPRM1*+ cells. Together, our *MORp* toolkit provides researchers cell type specific genetic access to target and functionally manipulate mu-opioidergic neurons across a range of vertebrate species and translational models for pain, addiction, and neuropsychiatric disorders.

Mu-Opioid receptors (MORs) are inhibitory G-protein coupled receptors expressed across molecularly and functionally distinct cell types in the peripheral (PNS) and central nervous systems (CNS), where they act as the primary effector of endogenous opioid peptides to fine-tune neural transmission, cellular excitability, and influence gene transcription[1]. Exogenous opioid compounds, whether employed clinically or recreationally, leverage this evolutionarily conserved MOR system at non-physiological concentrations and timescales leading to intended benefits like analgesia, and unwanted detriments like opioid dependence and respiratory depression[2]. In order to disentangle the underlying mechanisms and specific cells involved in opioidergic functions, there is a need for a widely adoptable, robust, and reliable method that can be employed across numerous in vivo and in vitro model systems for neuroscientific investigations.

The isolation and targeting of MORs, and the neuronal cell populations and circuitry that harbor them, initially relied on pharmacological and radiolabeled reagents to visualize and characterize physiological function in diverse species, such as mice, rats, pigeons, and non-human primates[3–5]. Recent improvements in the development of transgenic animals, including conditional *Oprm1* knock-out, fluorescently tagged MORs and Cre-recombinase lines[6–10], continue to refine our understanding of opioid receptor neurobiology and provide the ability to perform cellular level manipulations in complex behavioral tasks germane to pain and addictive behaviors, but are limited to rodent model systems. By contrast, recent advances in viral construct design and gene delivery have led to the production of highly efficient vectors for directing transgenes to specific neural structures and cell types by targeting unique genetic elements exclusive to target populations[11–14].

Taking advantage of these improvements in adeno-associated virus (AAV)-based gene delivery, we created two suites of constructs based on the mouse MOR promoter (*mMORp*) and the human MOR promoter (*hMORp*) elements that can transduce fluorescent reporters, chemogenetic actuators, biosensors, and recombinase effector transgenes in MOR+ neuronal cell types across multiple structures within the PNS and CNS. The size and design of these constructs make them amenable to the packaging limits of AAV vectors, and thus able to take advantage of the growing catalog of serotypes available for this vector that increase cell type transduction selectivity[15]. Here, we show that these viral tools are effective in rodents and other mammalian models of interest for the in vivo study of pain, addiction, emesis, and neuropsychiatric disorders involving opioid system dysfunction, including smaller mammals and non-human primates. We further show that these viruses can also be used in vitro when working with human iPSC-derived cell types, collectively demonstrating that these new viral constructs are capable of complementing and enhancing the current techniques and tools available to gain access to opioidergic cell types across germane to both basic and clinical research endeavors[16].

## Results

### Designing viral constructs to target promoter sequences within the *Oprm1* gene

To design a new promoter system for viral-assisted, selective access to MOR+ cell types, we analyzed a 1.9 Kb genomic region immediately upstream of the translation start codon (ATG) of the mouse mu opioid receptor gene (*Oprm1*) for the presence of transcriptional initiation elements with complex transcription factor binding topology using PROMO[17] and the Eukaryote Promoter Database[18] (Fig. 1a, Supplementary Fig. 1a, b). Prior reports on the putative *Oprm1* promoter sequence describe a "proximal" (−450 to −249 bp upstream of the ATG site) and a "distal" (−1326 to −794 bp upstream of the ATG site) region, each with transcription start sites (TSSs), which respectively account for 95% and 5% of the overall activity of the gene[19,20]. Therefore, we designed four complimentary sequences that covered both the proximal and distal promoter regions within the mouse *Oprm1* gene: *mMORp1* (−1797 to −265, 1532 bp), *mMORp2* (−1900 to −265, 1635 bp), *mMORp3* (−1621 to −1, 1620 bp), and *mMORp4* (−1621 to −18, 1603 bp; Fig. 1a, Supplementary Fig. 1c). In addition to these sequences targeted to the murine *Oprm1* gene, we also applied our construct design protocol to generate sequences targeted to the promoter region within the human *OPRM1* gene, resulting in the production of *hMORp1* (−1841 to −257, 1532 bp; Fig. 1a, Supplementary Fig. 1d). All promoter sequences were assembled into plasmids encoding an enhanced yellow fluorescent protein (eYFP) reporter fluorophore and packaged into serotype 1 AAV vectors (AAV1). Administration of *mMORp1-4* and *hMORp1* AAVs in vitro into well plates containing primary cultured rat hippocampal neurons at concentrations of $1-3 \times 10^{12}$ genome copies per milliliter (gc/mL) revealed both *mMORp1* and *hMORp1* constructs to robustly transduce cells, as evidenced by the presence of the eYFP

reporter (Supplementary Fig. 2a, b). The four *mMORp* and single *hMORp* constructs were then tested in vivo via intracranial injection at similar concentrations into the cortex of C57BL/6J mice. The AAV1-*mMORp*1-eYFP and AAV1-*hMORp*1-eYFP constructs once again produced robust transduction and expression of reporter eYFP, while no evidence of successful transduction was noted in tissue from animals injected with *mMORp2-4*, possibly due to the neuron-restrictive silencer element (NRSE) and transcription factor Sp1/Sp3 binding sites near the ATG site[21]. Further testing of *mMORp1* and *hMORp1* demonstrated similar successful transduction of other brain regions in vivo, including the central nucleus of the amygdala (CeA) and periaqueductal gray (PAG) (Supplementary Fig. 2c–g). Using *mMORp1* as a template, we then designed several constructs to drive the expression of different transgenes for cell labeling and circuit dissection approaches within MOR+ cell types (Fig. 1b). These variants included constructs encoding a genetically encoded calcium indicator (*mMORp*-GCaMP6f), a chemogenetic inhibitor (*mMORp*-hM4Di-mCherry), a Cre recombinase (*mMORp*-mCherry-IRES-Cre), a Flp recombinase (*mMORp*-FlpO), and the red fluorescent protein oScarlet (*mMORp*-oScarlet), which can be used as a reliable alternative to eYFP for labeling putative MOR+ cells (Supplementary Fig. 2i).

### *mMORp* viral construct shows selective expression on *Oprm1*/ MOR positive neurons across multiple brain regions

Before further examining the functional applications of the constructs described above, we first sought to confirm that the expression profile of our *mMORp1* AAV was both restrictive and selective within predominantly neuronal cell types with putative *Oprm1* promoter activity. To do so, we first intracranially injected C57BL/6J mice with AAV1-*mMORp1*-eYFP ($3 \times 10^{11}$ gc/mL), targeting the mPFC as a representative, MOR+ cell containing region ($N = 3$ male mice). The profile of transduced cells within the mPFC showed promising restriction across cortical layers 2, 5 and 6 within the cingulate (Cg1), prelimbic (PL) and infralimbic (IL) regions (Fig. 1d). These virally transduced mPFC cells were co-labeled for the neuronal marker NeuN, while none appeared to overlap with glial markers, such as the microglial marker Iba1, a cell type that has previously been speculated to harbor active *Oprm1* promoters[22–24] (Fig. 1e).

To determine the transduction efficiency and fidelity of *mMORp1* expressed in neurons with endogenous *Oprm1* promoter activity, we leveraged a rigorous methodological pipeline to assess mRNA, protein, and promoter-driven gene co-expression with *mMORp* encoded transgenes. First, we intracranially injected AAV1-*mMORp1*-eYFP into the central nucleus of the amygdala (CeA), the ventral tegmental area (VTA), and the dorsomedial striatum (DMS) of C57BL/6J mice, brain regions previously shown to express MOR, to observe viral spread and expression patterns. In the CeA, we found that most cells showed robust co-expression of *mMORp*-YFP and anti-MOR immunofluorescence (via a validated MOR antibody, Supplementary Fig. 3), with little expression in the adjacent basolateral amygdala (BLA) and other surrounding brain regions, which matches published reports of MOR expression in this subcortical region[25] ($N = 3$ male mice, Fig. 1f). Since MOR antibody staining labels diffuse dendrites and axons, with hard-to-resolve somas, we next performed fluorescent in situ hybridization (FISH) on CeA tissue sections to determine if endogenous *Oprm1* and transduced *EYFP* mRNA transcripts more clearly co-localized on CeA, DAPI-labeled nuclei. We observed a near total overlap of both transcript species within *mMORp*1 transduced cells (Fig. 1g), suggesting a high degree of specificity for our construct at the level of mRNA. However, given the dynamic and circadian expression of *Oprm1*[26,27], and the relatively short stability of the mRNA transcripts detectable by FISH at any one instance[28], we next used two transgenic mouse lines: an *Oprm1*^Cre knock-in/knock-out line developed by Dr. Richard Palmiter[29] and a bicistronic *Oprm1*^2A-Cre line developed by Dr. Julie Blendy using CRISPR knock-in[10] that both express Cre

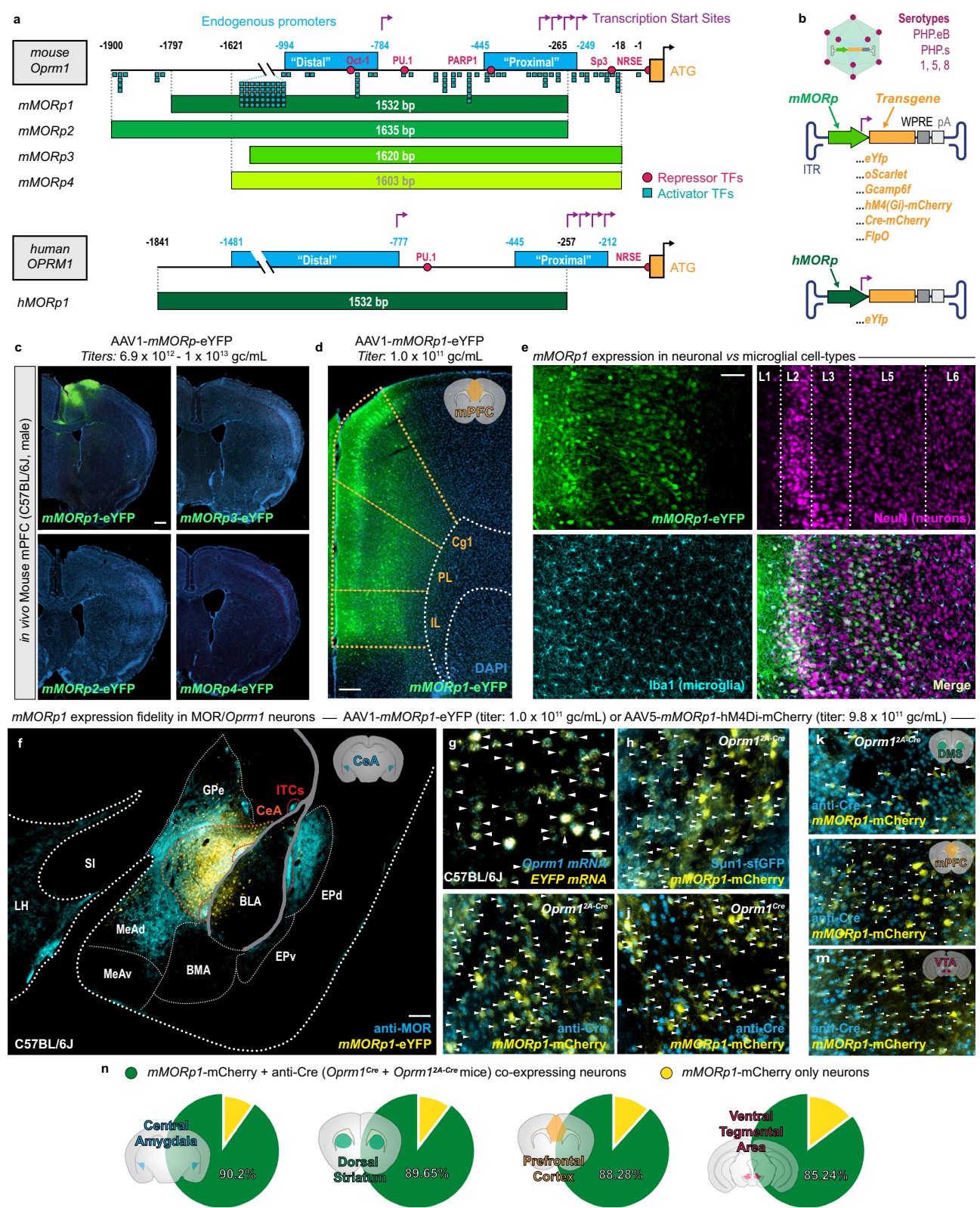

recombinase under the endogenous *Oprm1* promoter. Mice from the two *Oprm1*-Cre lines, either heterozygous for Cre (*Oprm1*[Cre], N = 2 male and 3 female mice) or crossed with the nuclear-envelop targeted fluorophore Sun1-sfGFP (*Oprm1*[2A-Cre]:Sun1, N = 2 male and 2 female mice), were injected with AAV5-*mMORp1*-hM4Di-mCherry at a standardized volume and titer (400 nL, 1 × 10[11] gc/mL) across four regions of interest (CeA, VTA, DMS, and mPFC), and tissue slices from successfully transduced regions were examined for mCherry co-labeling

with anti-Cre immunofluorescence or Sun1-sfGFP. Sun1-sfGFP and anti-Cre staining were observed to overlap with *mMORp1*-mCherry signal within the CeA of *Oprm1*[2A-Cre]:Sun1 mice (Fig. 1h, i), while all four transduced regions within *Oprm1*[Cre] or *Oprm1*[2A-Cre]:Sun1 mice showed similarly robust co-labeling of cells for both anti-Cre and *mMORp1*-mCherry signal (Fig. 1j–m). Analysis of the transduced regions of interests pooled from both Cre lines revealed the majority *mMORp1*-mCherry+ labeled neurons to be co-labeled for anti-Cre staining

**Fig. 1 | Development and validation of murine and human mu opioid receptor promoter (MORp) driven viral constructs. a** DNA sequence for the murine *Oprm1* (upper) and human *OPRM1* (lower) promoter regions, including the approximate locations of several transcriptional elements such as repressor and activator transcription factors (TFs), transcription start sites and promoter elements, as determined via PROMO and Eukaryotic Promoter Database & UCSC Genome Browser analyses of the murine and human genes (Supplementary Fig. 1). The promoter region encoded by the four murine promoter constructs (*mMORp1-4*) and single human promoter construct (*hMORp*) are depicted beneath each sequence map. **b** *mMORp* and *hMORp* construct designs and packaging schema within adeno-associated viral (AAV) vectors of multiple different capsid serotypes. **c** Transduction efficacy from initial in vivo intracranial injections of the *mMORp1-4-eYFP* constructs into C57BL/6J mouse medial prefrontal cortex (mPFC), scale bar = 500 μm. **d** mPFC expression pattern with AAV1-*mMORp1*-eYFP across mPFC subregions, including the cingulate (Cg1), prelimbic (PL) and infralimbic (IL) cortex, scale bar = 200 μm. **e** Higher magnification images of the mPFC following transduction with the *mMORp1* viral construct. Amplification of the eYFP signal, along with staining for both neuronal (NeuN) and microglial (Iba1) markers demonstrate selective transduction of neurons, with staining for additional glial markers to further verify this shown in Supplementary Fig. 5. Cortical layer division markers (Layers 1–6) highlight viral spread and efficiency, scale bar = 100 μm. **f** Overlap of *mMORp1*-eYFP viral expression and endogenous mu opioid receptor (MOR)

immunoreactivity within the central amygdala (CeA) (denoted by blue outline), but not surrounding amygdalar subregions of a C57BL/6J mouse, using a knock-out mouse-validated anti-MOR antibody (Supplementary Fig. 3). Basolateral amygdala (BLA), medial anterodorsal amygdala (MeAd), medial anteroventral amygdala (MeAv), basomedial amygdala (BMA), dorsal entopeduncular nucleus (EPd), ventral entopeduncular nucleus (EPv), intercalated cells (ITCs), globus pallidus externa (GPe), substantia innominota (SI), and lateral hypothalamus (LH); scale bar = 200 μm. **g** RNAscope FISH in the CeA of a *mMORp1*-eYFP injected mouse examining co-localized of *Oprm1* and *eYfp* mRNA transcripts. CeA co-localization of AAV5-*mMORp1*-hM4Di-mCherry and *Oprm1*[2A-Cre]:Sun1-sfGFP reporter nuclei (**h**) or anti-Cre staining (**i**). Co-expression of AAV5-*mMORp1*-hM4Di-mCherry with anti-Cre immunoreactive cells in the CeA of *Oprm1*[Cre] mice (**j**), and the dorsomedial striatum (DMS) (**k**), mPFC (**l**) and ventral tegmental area (VTA, **m**) of *Oprm1*[2A-Cre] mice. **n** Averaged number of *mMORp1*-mCherry+/anti-Cre+ cells compared to *mMORp1*-mCherry+/anti-Cre− cells quantified from successfully transduced brain regions of interest (from left or right hemispheres, or both) of *Oprm1*[Cre] and *Oprm1*[2A-Cre]:Sun1 mice within the CeA (-90.2%, N = 5, n = 9), DMS (-89.65%, N = 5, n = 7), mPFC (-88.28%, N = 3, n = 4), and VTA (-85.24%, N = 5, n = 8). Detailed information of total counts and quantification for each region within individual mouse lines can be found in Supplementary Fig. 4. Scale bar = 100 μm for **g**–**m**. White arrow heads denote cells in which co-labeling for *Oprm1* and *EYFP* transcript (**g**) or *mMORp1*-mCherry and anti-Cre signal (**h**–**m**) is observed.

(CeA = 90.2%%, *n* = 9 ROIs, from *N* = 5 mice; DMS = 89.7%, *n* = 7, *N* = 5; mPFC = 88.3%, *n* = 4, *N* = 3; VTA = 85.2%, *n* = 8, *N* = 5; Fig. 1n; quantification for individual mouse lines available in Supplementary Fig. 4a–h, with sample quantification of a single region of interest [ROI] shown in Supplementary Fig. 4i). Minimal expression of *mMORp*1-mCherry signal was observed on neurons negative for anti-Cre or anti-Cre and Sun1-eGFP, further indicating that successful transduction of cells via *mMORp*1 (hence referred to as *mMORp*) is restricted predominately to putative MOR+ cells-types with active genomic *Oprm1* promoters.

To further demonstrate that our *mMORp* constructs afford a level of selectivity and utility for targeting putative MOR+ cell populations that stand apart from other more generic promoter driven constructs, we performed co-injections of an AAV5-*hSyn*-mCherry reporter virus mixed one to one with a serotype-matched variant of our *mMORp*-eYFP virus (AAV5-*mMORp*-eYFP) into two of the representative MOR+ regions assessed in our previous specificity studies (CeA and VTA, Supplementary Fig. 2k, l). In both the CeA and VTA, we noted that the vast majority of *hSyn*-mCherry and *mMORp*-eYFP cells comprised separate populations (CeA: mCherry = 61.6%, eYFP = 24.1%, mCherry/eYFP = 14.3%, *n* = 4 ROIs from *N* = 2 mice; VTA: mCherry = 37.3%, eYFP = 40.7%, mCherry/eYFP = 22.0%, *n* = 2, *N* = 1; Supp. Fig. 2m, n), with mCherry + /eYFP+ cells only making up a small percentage of total *mMORp* transduced cells (CeA = 37.4%; VTA = 35.1%). Transduction patterns for co-injections into the CeA also showed noticeably greater spread of the *hSyn*-mCherry virus into the MOR- BLA, while *mMORp*-eYFP transduced cells were found to be more restricted to the general boundaries of the CeA itself, further supporting the transduction selectivity of the *mMORp* constructs over those which may utilize more common promoter sequences in their design.

As a final test of fidelity and specificity, we wanted to determine if the design of our *MORp* constructs would allow for transduction to occur predominantly in neurons, as opposed to more broadly across neuronal and glial cell types. AAV1-*mMORp*-eYFP was injected in a cohort of mice into mPFC, CeA and VTA, and a glial marker IHC panel was conducted to examine the overlap of *mMORp*-eYFP signal with CC1 (oligodendrocytes), PDGFRα (oligodendrocyte precursors), GFAP (astrocytes) and Iba1 (microglia) antibody staining (Supplementary Fig. 5). In all cases, minimal to no signal for glial marker staining was noted on cells positive for *mMORp*-eYFP across all regions of interest (CC1: mPFC = 1.9%, *n* = 4 ROIs from *N* = 2 mice; CeA = 1.4%, *n* = 4, *N* = 2; VTA = 4.3%, *n* = 2, *N* = 1; Supplementary Fig. 5a, b; GFAP: mPFC = 1.2%, *n* = 3, *N* = 2; CeA = 1.6%, *n* = 3, *N* = 2; VTA = 0.9%, *n* = 2, *N* = 1; Supplementary Fig. 5c, d; PDGFRα: mPFC = 1.0%, *n* = 2, *N* = 1; CeA = 5.1%, *n* = 3,

*N* = 2; VTA = 3.5%, *n* = 2, *N* = 1; Supplementary Fig. 5e, f; Iba1: mPFC = 1.3%, *n* = 2, *N* = 1; CeA = 1.6%, *n* = 2, *N* = 1; VTA = 1.2%, *n* = 3, *N* = 2; Supplementary Fig. 5g, h), indicating a higher transduction preference of our viral constructs for neurons. Additional gene expression analyses performed on cultured human neuronal (SHSY5Y), astrocytic (A172), microglial (C20) and murine microglial (N9) cell lines following transduction with AAV1-*hMORp*-eYFP or AAV1-*mMORp*-eYFP, respectively, revealed the normalized expression of *m/hMORp*-eYFP to be lower in C20 cells compared to SHSY5Y cells, and reduced across other cell lines (one-way ANOVA, *F* = 3.764, *P* = 0.0409, SHSY5Y v. A172: *P* = 0.073, SHSY5Y v. C20: *P* = 0.0251, SHSY5Y v. N9: *P* = 0.0645; Supplementary Fig. 6f), while *OPRM1/Oprm1* expression across glial lines was significantly reduced compared to SHSY5Y cells (one-way ANOVA, *F* = 413.5, *P* < 0.0001, SHSY5Y v. all: *P* < 0.0001; Supplementary Fig. 6e). Apart from the underlying predilection of AAVs for infecting neurons over other glial cell types[30–32], the exclusion of expression specifically in microglia may result from the inclusion of a PU.1 transcription factor binding region, which has been previously demonstrated to repress MOR expression in myeloid-lineage cells, including microglia[33,34].

## *mMORp* viral construct displays a robust expression profile in multiple brain regions across small mammalian model organisms

Following the above specificity and selectivity studies, we next wanted to examine the possible applications of our viral constructs in targeting populations of MOR+ neurons across several animal model systems utilized in the broader opioid research field. While many groups utilize mice to study aspects of opioid use disorder (OUD), withdrawal, and acute/chronic pain due to a growing wealth of transgenic lines[8,9,29], tools[35–38], and behavioral paradigms[16], there remain key advantages in the use of additional mammalian model systems, in particular rats and shrews, for addressing important questions regarding the endogenous opioid system's function and dysfunction[39–41]. Thus, we evaluated the ability of *mMORp* to transduce neurons within the brains of both Sprague-Dawley rats and Asian house shrews (*Suncus murinus*, a unique animal model to study the opioid system in the context of emesis and nausea[42–44]), compared with additional C57BL/6J mice. We intracranially injected AAV1-*mMORp*-eYFP (1.4 × 10$^{12}$ gc/mL) in two conserved, MOR+ representative structures in mice and rats: the CeA and VTA, and the area postrema/nucleus tractus solitarius (AP/NTS) within the dorsal vagal complex of the shrew, a structure known to express mu opioid receptors that contributes to feeding, emesis and the hypercapnic/hypoxic ventilatory response to low-oxygen[39,45–47].

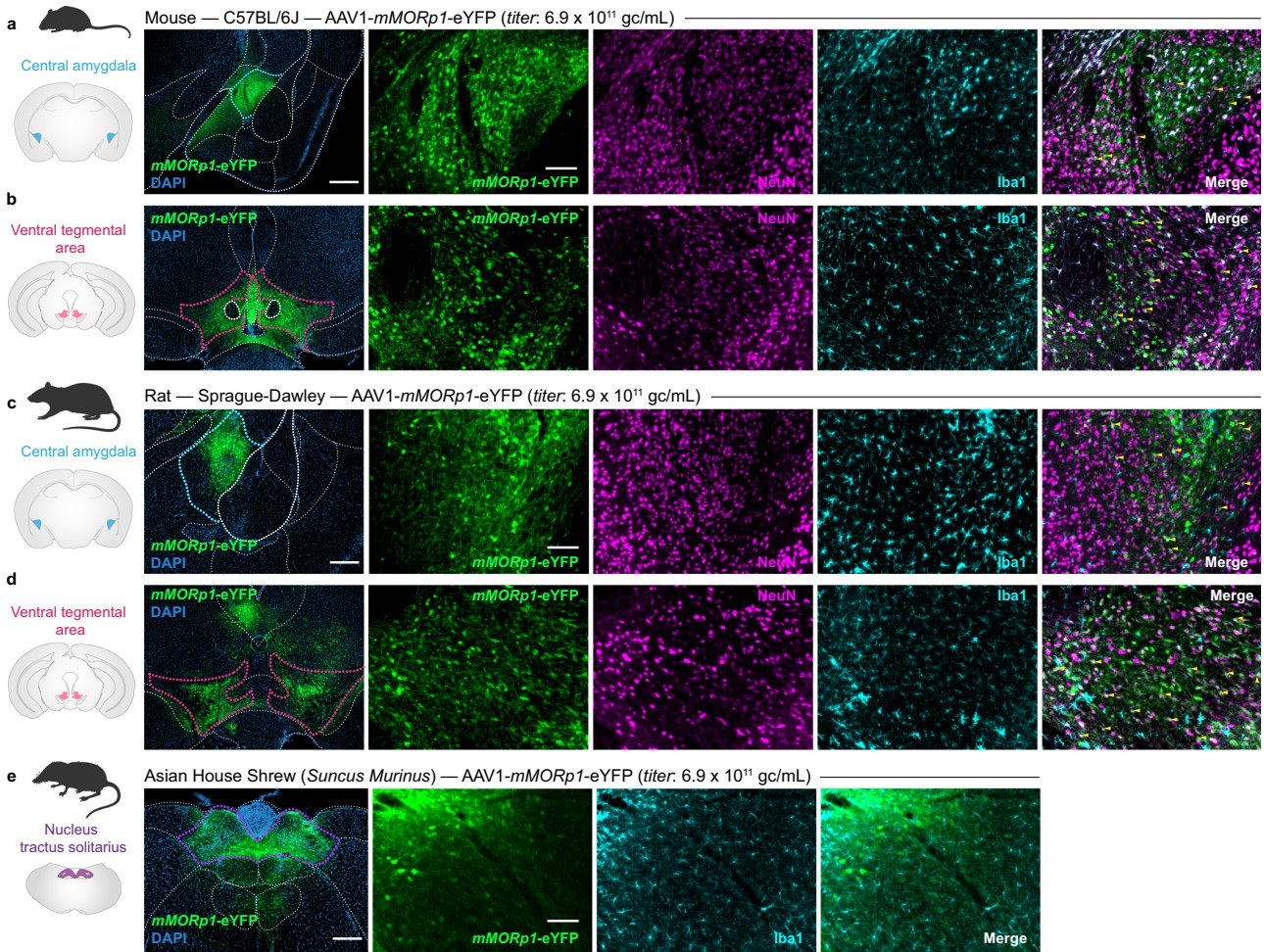

**Fig. 2 | *mMORp* viral transduction within putative MOR+ cells in multiple brain regions across mammalian model organisms. a** Mouse (C57BL/6 J) CeA expression of AAV1-*mMORp*-eYFP (titer: $1 \times 10^{11}$ gc/mL). *Left*: overview of the CeA (blue borders) and surrounding regions (white borders). *Right*: higher magnification images of *mMORp*-eYFP (anti-GFP amplified) with neuronal (NeuN) and microglial (Iba1) cell type markers. **b** Mouse VTA expression of AAV1-*mMORp*-eYFP: *Left*: VTA (magenta borders). *Right*: anti-GFP amplified eYFP, NeuN and Iba1. **c** Rat (Sprague-Dawley) CeA expression of AAV1-*mMORp*-eYFP (titer: $1 \times 10^{12}$ gc/mL). *Left*: overview of the CeA (blue borders) and surrounding structures. *Right*: anti-GFP amplified eYFP, NeuN and Iba1. **d** Rat VTA expression of *mMORp*-eYFP. *Left*: overview of VTA (magenta borders). *Right*: anti-GFP amplified eYFP, NeuN and Iba1. **e** Shrew (Asian house shrew) area postrema/nucleus tractus solitarius (AP/NTS) expression of AAV1-*mMORp*-eYFP (titer: $1 \times 10^{12}$ gc/mL). *Left*: overview of NTS (purple borders) and surrounding structures. *Right*: anti-GFP amplified eYFP, NeuN and Iba1. Scale bars = 100 μm (far left), 200 μm (right) for **a**–**e**. Staining for additional glial markers to demonstrate transduction of predominantly neurons in rat and shrew tissue is shown in Supplementary Fig. 8 (including quantification for Iba1 and *mMORp*-eYFP staining demonstrated in representative images above). Yellow arrows indicated representative NeuN/eYFP positive cells within merged images.

Viral expression in mouse CeA and VTA (N = 3 male mice, Fig. 2a, b) displayed similar patterns of restriction to targeted regions of interest to what we reported above, with additional staining once again showing much of the eYFP signal to be contained within NeuN+ cell bodies and not within any cells labeled for glial markers such as Iba1 once again. Tissue taken from rats targeted within the CeA and VTA (N = 3 male rats, Fig. 2c, d) showed complementary transduction patterns to those observed in mice, with eYFP signal primarily overlapping with NeuN+ cell bodies via IHC. Additional FISH staining of CeA tissue also showed similar results to those in mouse tissue, with most transduced cell bodies positive for *EYFP* transcript co-labeled for native rat *Oprm1* transcript (71.8%, n = 2, N = 1; Supplementary Fig. 7a–e). Transduced neurons were once again observed to be restricted primarily to the targeted regions of interest, indicating nominal levels of off-target spread or transduction of MOR- neurons. To confirm that this expression profile was selective for neurons within rat tissue as well, a similar glial marker IHC panel was conducted using CC1, GFAP and Iba1 antibody staining. As in mice, cells within rats transduced by *mMORp*-eYFP showed little to no overlap with signals for glial marker antibody staining within CeA (CC1: 1.0%, n = 2, N = 1; GFAP: 0.6%, n = 2, N = 1; Iba1: 0.9%, n = 2, N = 1; Supp. Fig. 8a, b) or VTA (CC1: 0.5%, n = 2, N = 1; GFAP: 1.5%, n = 2, N = 1; Iba1: 2.3%, n = 2, N = 1; Supp. Fig 8c, d). Within shrew tissue, similar patterns of transduction were also found, with *mMORp*-eYFP restricted exclusively to non-Iba1+ cells (0.6%, n = 3, N = 2; Supp. Fig. 8e), and overall spread of the virus noted to be relatively restricted to the borders of the AP/NTS structure where the majority of MOR+ cells reside[48] (N = 2 female shrews, Fig. 2e). Taken together, these cross-species studies demonstrate that our constructs are highly viable in both rat and shrew models systems and show a similar pattern of restricted expression within brain structures known to harbor MOR+ neuronal populations and respond to MOR manipulation.

### *mMORp*-hM4Di produces robust anti-nociception in spinal nociceptive circuits

To demonstrate the application of our constructs for encoding transgene products useful for interrogating opioidergic circuits in pain/nociceptive pathways, we conducted a series of functional and

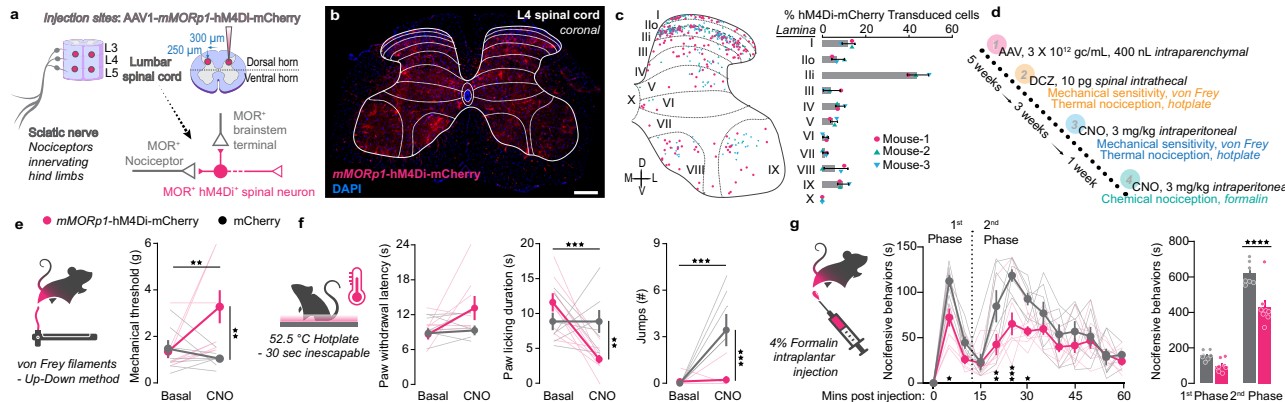

**Fig. 3 | *mMORp*-hM4Di spinal cord expression and chemogenetic-induced analgesia. a** AAV1-*mMORp*-hM4Di-mCherry injection schema within the lumbar spinal cord in C57BL/6J mice to inhibit dorsal horn MOR+ cells and not MOR+ nociceptors or descending brain stem circuits. **b** *mMORp*-hM4Di-mCherry expression and spread across the L4 spinal cord; scale bar = 100 μm. **c** Location map and quantification of *mMORp*-hM4Di-mCherry+ cells across the Rexed laminae in the dorsal and ventral horns (*N* = 3 mice). **d** Experimental timeline for viral injections and chemogenetic behavioral testing. **e** Mechanical sensory thresholds (von Frey Up-Down testing) in *mMORp*-hM4Di-mCherry injected mice (*N* = 9 mice) compared with *hSyn*-mCherry injected controls (*N* = 7 mice) at baseline and 30 min following systemic CNO administration (3 mg/kg; Two-way ANOVA + Bonferroni: main effect: *P* = 0.011 [viral treatment × CNO treatment]; multiple comparisons: basal v. CNO, *P* = 0.990 [mCherry], *P* = 0.006 [hM4Di]). Average response changes per group shown as thick gray (mCherry) or red (hM4Di) lines. Individual mice are shown as thin gray and red lines. **f** Nocifensive behaviors observed on an inescapable 52.5 °C hot plate over a 30-sec trial for the same animals (*N* = 9 hM4Di-mCherry, *N* = 7 mCherry): latency (sec) to hind paw withdrawal (two way ANOVA + Bonferroni: main effect: *P* = 0.085 [viral treatment], *P* = 0.162 [CNO treatment]; multiple comparisons: basal v. CNO, *P* > 0.999 [mCherry], *P* = 0.067

[hM4Di]), hind paw licking duration (two way ANOVA + Bonferroni; main effects: *P* = 0.008 [viral treatment × CNO treatment], *P* = 0.008 [viral treatment]; multiple comparisons: basal v. CNO, *P* > 0.999 [mCherry], *P* = 0.0007 [hM4Di]), and total jumping bouts (two way ANOVA + Bonferroni; main effects: *P* = 0.004 [viral treatment × CNO treatment], *P* = 0.002 [viral treatment], *P* = 0.006 [CNO treatment]; multiple comparisons: basal v. CNO, *P* = 0.0006 [mCherry], *P* > 0.999 [hM4Di]). **g** Time course (left) and cumulative global scoring (right) of nocifensive behaviors (licking, biting, jumping, etc.) observed in the formalin injection test during the first and second phases of testing in *mMORp*-hM4Di-mCherry (*N* = 9) and control (*N* = 7) animals at basal and post-CNO time points (*Time course:* two-way ANOVA + Bonferroni; main effects: *P* = 0.006 [time bin × nocifensive behaviors], *P* < 0.0001 [time bin], *P* = 0.001 [nocifensive behaviors]; multiple comparisons: mCherry v. hM4Di, *P* = 0.014 [5 min], *P* = 0.008 [20 min], *P* = 0.0002 [25 min], *P* = 0.48 [30 min]. *Global scoring:* two-way ANOVA + Bonferroni; main effects: *P* = 0.006 [stage × nocifensive behaviors], *P* < 0.0001 [stage], *P* = 0.001 [nocifensive behaviors]; multiple comparisons: mCherry v. hM4Di, *P* = 0.222 [1st stage], *P* < 0.0001 [2nd stage]). All data are presented as means ± SEM *P* < 0.05, **P* < 0.01, ***P* < 0.001, ****P* < 0.0001.

behavioral studies using the *mMORp*-h4MDi-mCherry and *mMORp*-GCaMP6f constructs. Intrathecal morphine and other opioids produce strong anti-nociception when binding to spinal MORs that are expressed on the presynaptic terminals of primary afferent nociceptors and on descending inputs from the brain stem, as well as MORs expressed on intrinsic dorsal horn neurons[1]. These compounds engage inhibitory Gi signaling cascades that silence these different neural populations to disrupt the transmission of peripheral nociceptive information to supraspinal brain regions. However, behavioral pharmacology does not allow for testing the role of different MOR+ neural populations within this critical nexus of the nociceptive pathways regarding analgesia. Since the DREADD GPCR, hM4Di, couples to similar Gi cascades as MOR, we hypothesized that *mMORp*-hM4Di could mimic intrathecal morphine antinociception[49], but in a selective manner that tests the necessity of spinal cord MOR+ neurons to induce analgesia, independent of afferent and brainstem terminal opioid-mediated inhibition[50]. Thus, for chemogenetic manipulation of spinal MOR+ neurons in mice experiencing nociceptive stimuli, we directly injected C57BL/6J mice with AAV1-*mMORp*-h4MDi-mCherry at the L4-L5 lumbar sections of the spinal cord (intraparenchymal, 400 nL, ~3 × $10^{12}$ gc/mL, per injection site; Fig. 3a). Robust *mMORp*-hM4Di-mCherry expression was detected throughout the dorsal and ventral horns, primarily in lamina II inner, consistent with expression patterns observed in MOR-mCherry mice[25], without detection in dorsal root ganglia axonal inputs (*N* = 3 male mice; Fig. 3b, c). We then conducted a series of well-validated, common nociceptive behavioral assays on these mice with and without different DREADD agonists to measure anti-nociceptive effects (Fig. 3d). We found that *mMORp*-hM4Di injected animals displayed an increase in mechanical threshold sensitivity in the von Frey filament Up-Down test following either intrathecal deschlorocclozapine (DCZ, 10 pg; two-way ANOVA, *F* = 8.911, *P* = 0.0105;

Supplementary Fig. 9a) or systemic clozapine N-oxide (CNO, i.p., 3 mg/kg; two-way ANOVA, *F* = 8.521, *P* = 0.0112) compared to the within-subjects baseline thresholds and mCherry control mice (Fig. 3e). Similarly, the *mMORp*-hM4Di group showed an increase in reflexive latency to withdrawal, and decreases in the duration of paw licking behavior and jumping or escape-related events on an inescapable hot plate (52.5 °C) compared to control animals following either DCZ (hot plate latency: two-way ANOVA, *F* = 19.81, *P* = 0.0007, duration: *F* = 2.058, *P* = 0.1751, jump/escape: *F* = 4.921, *P* = 0.0450; Supplementary Fig. 9b), or CNO treatment (duration: two-way ANOVA, *F* = 9.514, *P* = 0.0081; jump/escape: 2-way ANOVA, *F* = 13.65, *P* = 0.0024; Fig. 3f). Lastly, *mMORp*-hM4Di and mCherry groups received systemic CNO 30 min prior to an intraplantar injection of a 4% formalin solution to induce TRPA1-nociceptor hypersensitivity and subsequent central sensitization[51,52]. In the *mMORp*-hM4Di mice we observed significant decreases in nocifensive behaviors (paw licking, biting, escape behaviors, etc.) during both the first stage of behavioral observations (direct activation of afferent nociceptors) and the later second stage (reflective of the inflammatory and central sensitization component) of the test (Fig. 3g; time: two-way ANOVA, *F* = 22.27, *P* < 0.0001; phase: two-way ANOVA, *F* = 401.5, *P* < 0.0001). In total, these tests demonstrate the functional validity of our chemogenetics-based viral constructs in behaviorally relevant tasks, as exemplified by the ability of *mMORp*-hM4Di to produce similar antinociceptive effects to those of spinal opioid agonist administration[22].

### *mMORp*-GCaMP6f permits readout of neuronal population activities in response to morphine, opioid withdrawal, and noxious stimuli

We next assessed the ability of the *mMORp*-GCaMP6f construct for use in fiber photometry calcium imaging in awake, behaving animals to

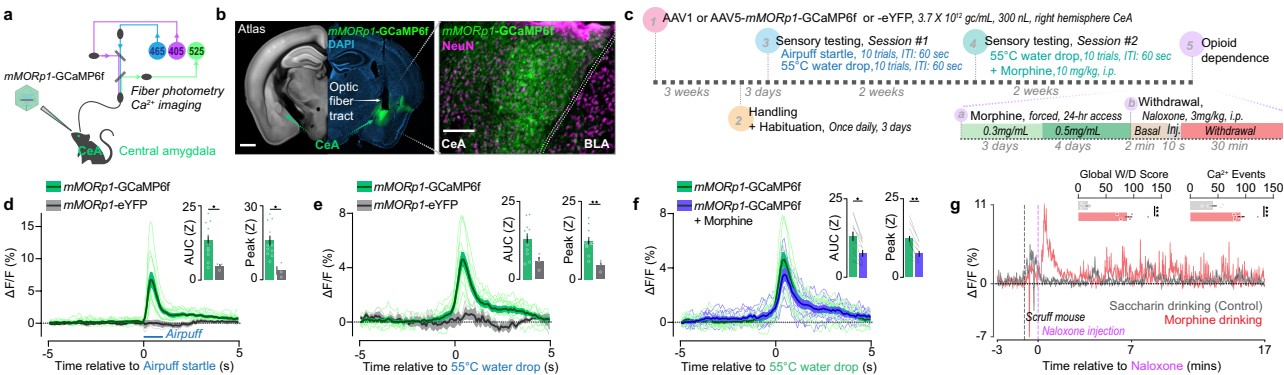

**Fig. 4 | *mMORp*-GCaMP6f in vivo photometry recording of MOR+ neural population calcium activity in response to noxious stimulation, opioid administration, and opioid withdrawal. a** In vivo fiber photometry recording set up and overall experimental design, as well as AAV1/5-*mMORp*-GCaMP6f injection schema in C57BL/6J mice. **b** *mMORp*-GCaMP6f viral targeting and transduction efficiency within CeA neurons (NeuN+) and fiber optic implant placement; scale bars = 500 μm (left), 100 m (right). **c** Photometry experimental timeline for *mMORp*-GCaMP6f (*N* = 12) and *mMORp*-eYFP (*N* = 3) injected mice. **d** Normalized calcium-mediated activity responses observed in the CeA of *mMORp*-GCaMP6f and *mMORp*-eYFP mice in response to 10 air puff applications (1-min inter-stimulation interval. Thick lines = averaged responses, thin lines = individual mice average responses to all 10 stimulations). Inset graphs depict differences in the area under the curve (AUC) quantified between averaged GCaMP6f and eYFP control response (two tailed unpaired t-test, *P* = 0.035) and peak z-score between groups (two tailed unpaired t-test, *P* = 0.011). **e** Plots for normalized, calcium-mediated responses to the application of 10 - 55 °C hot water droplets to the left hind paw (1-min inter-stimulation interval). Thick lines = average group response, thin lines = individual mice averaged response. Insets show AUC (two tailed unpaired t-test, *P* = 0.051) and

peak z-score (two tailed unpaired t-test, *P* = 0.009) comparisons for eYFP (*N* = 3) and GCaMP6f (*N* = 12) expressing mice. **f** Normalized, calcium-mediated responses to an additional round of hot water hind paw stimulations in the *mMORp*-GCaMP6f mice (*N* = 12) before and 30 min after morphine (i.p., 10 mg/kg). Opioid-naïve responses (green) and post-morphine responses (blue) are shown as grouped averages (thick lines) and individual averages (thin lines) following 10 stimulations. Insets show AUC (two tailed unpaired t-test, *P* = 0.016) and peak z-scores (two tailed unpaired t-test, *P* = 0.005) comparisons between the pre- and post-morphine responses in *mMORp*-GCaMP6f mice. Gray lines track changes in AUC and peak z-score for individual mice in pre- and post-morphine treatment conditions. **g** Normalized, calcium-mediated responses from *mMORp*-GCaMP6f mice undergoing an escalating forced morphine drinking paradigm (Fig. 4c) for morphine drinking (red, *N* = 6 mice) or saccharin drinking (gray, *N* = 5 mice) subjects following naloxone induced precipitated withdrawal (i.p., 3 mg/kg). Insets show the global withdrawal behavior score (left) and total calcium-mediated events (right) post-naloxone between morphine and saccharin drinking mice (W/D score: two tailed unpaired t-test, *P* = 0.0001; Ca²⁺ events: two tailed unpaired t-test, *P* = 0.0007). All data are presented as means ± SEM **P* < 0.05, ***P* < 0.01, ****P* < 0.001.

assess population activity changes of transduced MOR+ CeA neurons, a sub-region of the amygdala shown previously to express MORs and to be implicated in aspects of nociceptive processing[53,54]. We injected C57BL/6J mice with AAV1-*mMORp*-GCaMP6f (400 nL, ~2 × 10¹² gc/mL; *N* = 12 male mice) or an eGFP encoding control virus (*N* = 3 male mice) in the right CeA (Fig. 4a) prior to implantation of a fiber optic head post just dorsal to the injection site. As with the other *mMORp* constructs, *mMORp*-GCaMP6f expressed strongly, with relatively restricted expression in the CeA and minimal spread into the neighboring BLA or striatum (Fig. 4b). Following a three-week incubation period, mice were run through a battery of behavioral experiments to gauge relative calcium activity changes in response to noxious stimuli (Fig. 4c). Initially, mice were exposed to brief air puffs delivered to the abdomen (compressed air canister, 1 s puff) and hot water drops applied to the left hind paw (~55 °C, ~25 μl drop), and the recorded calcium mediated events were time-locked to the stimulus application. In response to air puffs, calcium activity in the CeA of *mMORp*-GCaMP6f injected mice increased significantly relative to eGFP control animals when examining both the area under the curve (AUC) and peak z-score of recorded events (AUC: GCaMP = 13.2, eGFP = 4.7, two tailed unpaired t-test, *P* = 0.0351; peak z-score: GCaMP = 15.9, eGFP = 3.7, two tailed unpaired t-test, *P* = 0.0112; Fig. 4d), thereby indicating proper functionality of our GCaMP6f construct in CeA neurons, a region known for its elevated activity profile in response to unexpected external stimuli[55,56]. Similarly, hot water applications reliably produced robust, time-locked calcium mediated events when compared with controls (AUC: GCaMP = 13.0, eGFP = 5.7, two tailed unpaired t-test, *P* = 0.051; peak Z-score: GCaMP = 12.3, eGFP = 4.5, two tailed unpaired t-test, *P* = 0.009; Fig. 4e). To assess if *mMORp*-GCaMP6f-tranduced CeA neurons are opioid-sensitive, we next administered morphine (i.p., 10 mg/kg) 30 min prior to another round of hot water hind paw stimulations. Morphine treated *mMORp*-GCaMP6f animals displayed a marked reduction in calcium mediated event AUC and peak amplitude

(AUC: no morphine = 13.0, morphine = 7.7, two tailed unpaired t-test, *P* = 0.016; peak z-score: no morphine = 12.3, morphine = 7.6, two tailed unpaired t-test, *P* = 0.005; Fig. 4f), indicative of a potential effect of morphine on the activity of transduced, putative MOR+ CeA neurons.

We next sought to extend the utility of our *mMORp*-GCaMP6f construct to examining applications aimed at modeling chronic opioid use and withdrawal. Using the same cohort of *mMORp*-GCaMP6f mice, we placed animals on a forced morphine drinking paradigm using home cage water bottles containing either an increasing concentration of morphine, or the artificial sweetener saccharin (*N* = 5 male mice, saccharin; *N* = 6 male mice, morphine). Drug treated *mMORp*-GCaMP6f groups were supplied with water containing 0.3 mg/ml morphine, 0.2% saccharin for 3 days, and 0.5 mg/ml morphine, 0.2% saccharin for 4 days, while control *mMORp*-GCaMP6f groups were only provided 0.2% saccharin laced water (Fig. 4c). After the final day of drinking, mice underwent a precipitated withdrawal challenge (naloxone, i.p., 3 mg/kg) prior to the start of behavioral testing and fiber photometry recording. Morphine drinking, naloxone treated *mMORp*-GCaMP6f animals displayed significantly greater characteristic opioid withdrawal behavioral phenotypes (wet dog shakes, biting, jumping/escape behavior) immediately following naloxone injection, with an associated, significant increase in the total number of calcium-mediated events scored within a 20-minute observation window. By comparison, saccharin drinking *mMORp*-GCaMP6f animals showed little to no changes in overt behavior and calcium-mediated activity profiles for the duration of testing (average post naloxone global withdrawal score: saccharin = 18.4, morphine = 88.5, two tailed unpaired t-test, *P* = 0.001; average post naloxone calcium events: saccharin = 41, morphine = 91, two tailed unpaired t-test, *P* = 0.0007; Fig. 4g). Taken together, these use case experiments demonstrate that our *mMORp*-GCaMP6f construct can be paired with widely adopted techniques for

recording in vivo neural activities in pain and OUD-related studies.

### *mMORp* driven recombinases and PHP.eB/PHP.s capsid packaging allows for intersectional genetic access to PNS and CNS opioidergic cell populations and circuits

Increasingly, many contemporary neuroscience investigations aim to both manipulate specific neuronal subpopulations and circuits based on multiple dimensions, including molecular expression, connectivity, function, and location within the nervous system, with viral tools becoming an increasingly invaluable means to achieve these levels of specificity[15]. Determining which cell types are accessed can be partly controlled by the capsid proteins of a specific virus (see Supplementary Fig. 16), such as the recently engineered PHP.eB and PNS PHP.S capsids variants, which are capable of selectively transducing cells within the CNS or the PNS over cells present in non-neural tissue types, respectively[12,57], as well as the inclusion of specific promoter and/or enhancer element into constructs packaged within AAVs. Furthermore, hundreds of existing Cre and Flp recombinase transgenic mouse lines are also in use in labs around the world which can be used to achieve cell and circuit specific genetic access[58,59]. To capitalize on these advances in targeting strategies and tool development, we created four example *mMORp* constructs useful for intersectional neuronal labeling and tracing studies: a *mMORp*-mCherry-IRES-Cre construct encoding Cre recombinase, a *mMORp*-FlpO construct encoding Flipase, and two *mMORp*-eYFP constructs packaged in AAV-PHP.eB and AAV-PHP.s capsids. Single-cell RNA sequencing studies show that *Oprm1* is expressed in both glutamatergic pyramidal cell types as well as GABAergic interneurons[60–63]. Selective access to these different classes of cortical neurons however has not been possible, leaving gaps in our basic understanding of cortical opioidergic processes. Thus, to test our *mMORp*-driven recombinases for their effectiveness in targeting GABAergic subtypes of MOR+ cortical neurons, we first co-injected a mixture of AAV8-*mMORp*-mCherry-IRES-Cre (titer: $7 \times 10^{12}$ gc/mL) combined with AAV9-*hDlx*-FLEx-eGFP (titer: $2.2 \times 10^{13}$ vg/mL) viruses into the mPFC and somatosensory cortex (S1) of C57BL/6 J mice ($N = 3$ male mice). The latter construct uses the human *Dlx* enhancer element to promote viral transduction in forebrain GABAergic cells and contains a FLEx switch making eGFP transgene expression dependent on the presence of Cre recombinase within a transduced cell[13]. We observed the same expression pattern of mCherry in mPFC across the cortical layers as shown in Fig. 1d for our first round of *mMORp*-eYFP injections, which was mirrored in S1 brain sections, while eGFP expression was restricted to a subset of small cells with non-pyramidal morphology in both mPFC and S1 (Fig. 5a, b). Next, we co-injected AAV1-*mMORp*-FlpO (titer: $1.4 \times 10^{12}$ gc/mL) virus with a pan-neuronal, human Ef1α promoter-driven Flp-dependent mCherry reporter (AAV9-*Ef1a*-fDIO-mCherry, titer: $2.4 \times 10^{13}$ vg/mL) into a separate cohort of C57BL/6J mice ($N = 3$ male mice) to demonstrate an additional recombinase-based targeting strategy to label putative MOR+ neurons. We found that *mMORp*-FlpO successfully drove Flp recombinase dependent recombination within *Ef1a*-fDIO-mCherry transduced cells and observed numerous mCherry labeled cells in both mPFC and S1 (Fig. 5c, d). Taken together, these two strategies show that our *mMORp* constructs can also be combined with the ever-expanding catalogs of recombinase-based viral tools to access a multitude of cell types in the brain.

We next evaluated the use of the PHP.eB and PHP.s capids to deliver *mMORp*-eYFP into broad, CNS and PNS distributed MOR+ cells. In adult C57BL/6J mice, we delivered AAV-PHP.eB-*mMORp*-eYFP (titer: $8.6 \times 10^{12}$ gc/mL) via retro-orbital injection (50 μl) in order to facilitate better systemic viral spread and distribution, and observed robust expression of eYFP in cells throughout the spinal cord and brain in both sagittal (Fig. 5e) and coronal tissue preparations (Supplementary Fig. 10), with high magnification insets shown next to representative

coronal sections to provide a clearer view of the *mMORp*-eYFP+ cells observed in several of the structures of interest targeted in our intracranial focal injection studies. In C57BL/6J P2 pups, we performed an intracerebroventricular injection of AAV-PHP.S-*mMORp*-eYFP (titer: $1.4 \times 10^{13}$ gc/mL, Fig. 5f, upper) and after 6 weeks incubation, performed FISH on dorsal root ganglia sections for *EYFP* and *Oprm1* mRNA transcripts. When compared to pups that had been similarly injected with a generic AAV-PHP.S-CAG-tdTomato encoding virus (titer: $2.1 \times 10^{13}$ vg/mL, Fig. 5f, lower), cells counted in DRG sections from the *mMORp* injected animals ($n = 11$ ROIs) showed robust co-localization of *EYFP* and *Oprm1* transcripts on/around the same DAPI delineated nuclei, while co-localization of *tdTomato* and *Oprm1* transcripts ($n = 14$ ROIs) was significantly more sparse (*Oprm1+tdTomoto/tdTomato* = 0.60, *Oprm1* + *EYFP/EYFP* = 0.83, two tailed unpaired t-test with Welsh's correction, $P = 0.0437$, Fig. 5g). Similarly, we noted that the majority of *Oprm1* positive cells in DRG sections collected from mice injected with the AAV-PHP.S-*mMORp*-eYFP virus were co-labeled for *EYFP* transcript compared to those cells labeled for *EYFP* alone (*EYFP/Oprm1+* = -82.5%, *EYFP*+ only = -17.5%, Fig. 5h). Taken together, these representative findings highlight the use of our *mMORp* constructs in combination with viral capsid engineering to selectively target MOR+ primary afferents and large numbers of spinal and brain neurons.

### Human and mouse *MORp*s provide unique access to opioidergic cells in non-human primates in vivo, and human-derived neuronal cell cultures in vitro

Having characterized our *MORp* constructs in rodents and other small mammalian model systems, we next sought to determine the viability of our human promoter derived *MORp* construct (*hMORp*). We first performed a series of intracranial injections using the AAV1-*hMORp*-eYFP virus (titer: $1.17 \times 10^{12}$ gc/mL) in a single male rhesus macaque, targeting a region anatomically complementary to one we had examined for transduction efficiency of the *mMORp* constructs in our rodent studies, the dorsal anterior cingulate cortex (dACC), as well as the insular cortex and medial thalamus, with additional injections of the AAV1-*mMORp*-eYFP construct (titer: $1.4 \times 10^{13}$ gc/mL) performed in the amygdala[64–66] (Fig. 6a, b, Supplementary Figs. 11 and 12). Tissue sections taken from dACC revealed robust transduction of NeuN+ neurons throughout multiple layers of the cortex, with the greatest densities of eYFP+ neuronal cell bodies localized to cortical layers II, V and VI, and the processes of these neurons noted to extend into/throughout layers I, III, V and VI (Fig. 6c, Supplementary Fig. 12). Immunostaining in the dACC for microglia with anti-Iba1 showed no overlap with eYFP+ cells, suggesting that like the *mMORp* constructs, our *hMORp* construct preferentially targets putative MOR+ neurons and not glial cells (Fig. 6d). To confirm *hMORp* selectivity in MOR+ neuron, we performed both a similar glial marker IHC panel and FISH on dACC sections and quantified the co-localization of transgene eYFP signal with that of glial marker staining, or *EYFP* mRNA transcripts with macaque *OPRM1* mRNA transcripts across different regions of interest within the dACC. Glial marker staining with antibodies against CC1, GFAP, PDGFRα and Iba1 in macaque dACC transduced tissue (Supplementary Fig. 13a–c) revealed similar results to those observed in mouse, rat and shrew tissue assessed for *MORp*-eYFP signal, in that few to no cells were noted to co-label for eYFP and any glial marker signal (CC1: 1.6%, $n = 2$ ROIs, $N = 1$ macaque; GFAP; 1.2%, $n = 2$, $N = 1$; PDGFRα: 1.5%, $n = 2$, $N = 1$; Iba1: 0.5%, $n = 4$, $N = 1$; Supplementary Fig. 13d–g). For FISH staining, we noted that the majority of *OPRM1* labeled cells throughout the transduced region of the dACC were co-labeled for *EYFP* transcript compared to those cells labeled for *OPRM1* or *EYFP* alone (*OPRM1+/EYFP+* = -81.4%, *EYFP+/OPRM1−* = -18.6%; Fig. 6e, f; expanded quantification and individual ROI counts presented in Supplementary Fig. 14). Successful transduction of neurons expressing the *OPRM1* promoter was also observed in the amygdaloid structure via

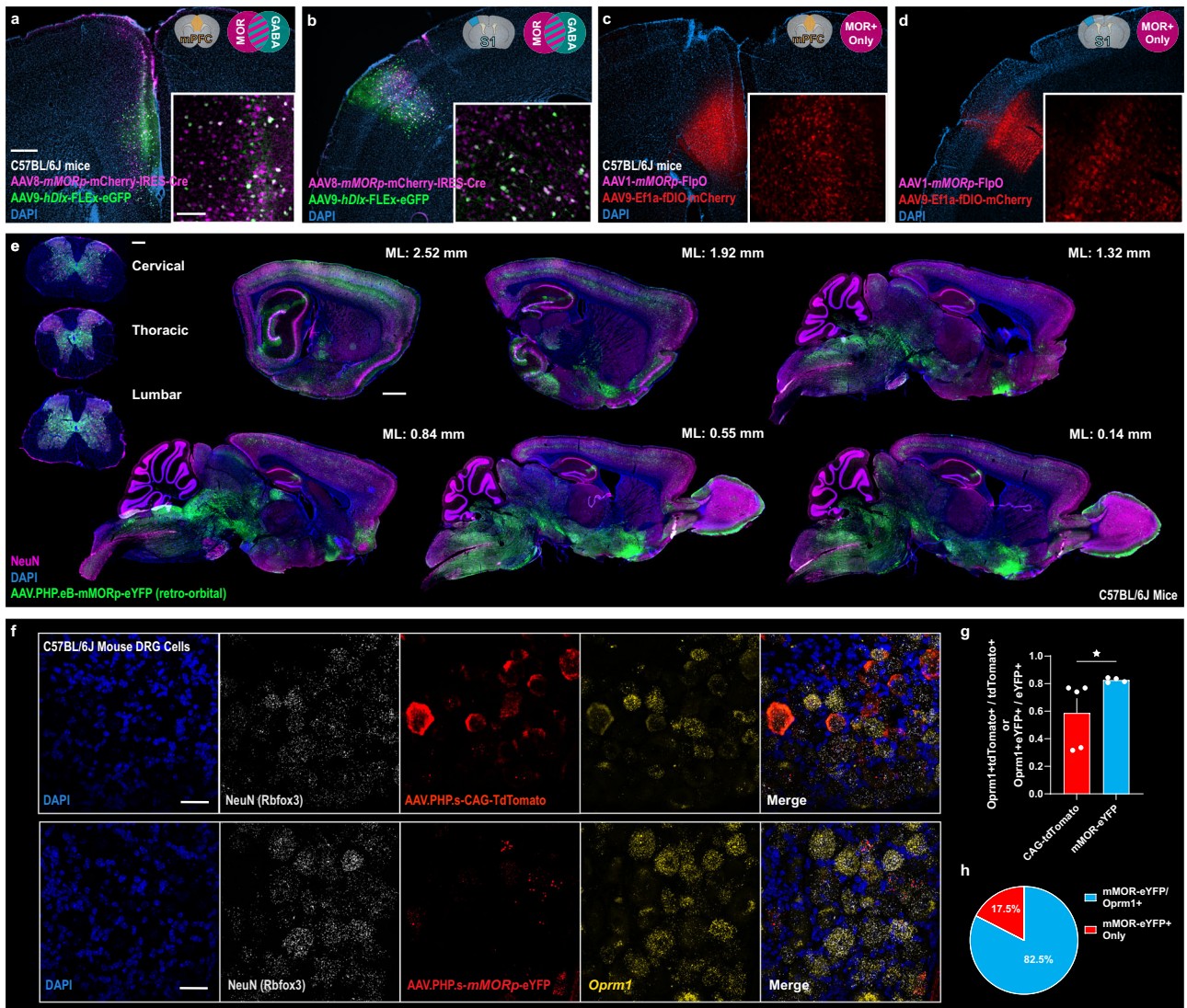

**Fig. 5 | *mMORp* driven recombinases and PHP.eB/PHP.S capsid packaging support intersectional viral strategies for gaining genetic access to CNS and PNS opioidergic cells/circuits.** C57BL/6J mice injected with a 9 μl:1 μl mix of AAV8-*mMORp*-mCherry-IRES-Cre and a Cre-dependent AAV9-*hDlx*-FLEx-eGFP with intersectional expression in putative MOR+/GABAergic neurons in mPFC (**a**) and somatosensory cortex (S1, **b**). Insets in **a** and **b** show higher magnification images of *hDlx*-FLEx-eGFP cells (green) overlap with *mMORp*-mCherry-IRES-Cre cells (magenta); scale bars = 200 μm and 100 μm for insets. C57BL/6J mice injected in mPFC (**c**) or S1 (**d**) with a 9 μl:1 μl mix of AAV1-*mMORp*-FlpO and Flp-dependent AAV9-*Ef1α*-fDIO-mCherry. Inset high magnification images show mCherry+ cells. **e** CNS expression in representative sections from the spinal cord (coronal) and brain (sagittal along the medial-lateral axis relative to Bregma; coronal sections are shown in Supplementary Fig. 10) of C57BL/6J mice injected retro-orbitally with AAV.PHP.eB-*mMORp*-eYFP virus; scale bars = 500 μm (spinal cord sections) and 1000 μm (sagittal sections). **f** PNS dorsal root ganglia (DRG) FISH from C57BL/6J mice with intracerebroventricular injection of either AAV.PHP.S-*mMORp*-eYFP (upper) or AAV.PHP.S-*CAG*-tdTomato (lower). Custom cDNA probes targeting *Rbfox3* (NeuN), *Oprm1*, *EYFP* and *tdTomato* transcripts; scale bars = 50 μm. **g** Quantification of total *Oprm1*+ cells co-labeled for *tdTomato* in control animals (*N* = 5 mice, *n* = 14 ROIs) compared to total *Oprm1*+ cells co-labeled for *EYFP* in PHP.S-*mMORp*-eYFP injected mice (*N* = 4 mice, *n* = 11 ROIs) across treatment groups (two tailed unpaired t-test with Welsh's correction, *P* = 0.0437). **h** Summary quantification of the percent total number of cells positive for *EYFP* transcript (i.e., transduced by the AAV.PHP.s-*mMORp*-eYFP virus) that were also either positive for *Oprm1* transcript (*mMORp*-eYFP/*Oprm1*+, 82.5%) or negative for *Oprm1* transcript (*mMORp*-eYFP+ only, 17.5%) to demonstrate AAV.PHP.s-*mMORp*-eYFP specificity within mouse DRG neurons. Data presented as means ± SEM, *P < 0.05.

the *mMORp* construct, as well as the insula and mediodorsal thalamus via the *hMORp* construct (Fig. 6g), with tissue samples containing the insular cortex and/or the amygdala also showing robust expression of the eYFP tag within neuronal cell bodies and processes in both regions (Supplementary Fig. 12), suggesting a broad application for *hMORp* to provide genetic access to MOR+ cells in multiple regions of the brain in non-human primate, genetically intractable subjects.

Lastly, we wanted to determine whether our *MORp* viruses would be able to provide genetic access to these same putative MOR+ cells within human derived model systems. To test this, we cultured and differentiated human iPSCs (LiPSC-GR1.1 line) to produce either

nociceptor-like neuronal cells or non-neuronal cardiomyocytes. Differentiated nociceptor-like cells have previously been shown to possess gene expression profiles consistent with those of nociceptors and sensory neurons observed in vivo. These neurons also show distinct expression profiles of select genes during differentiation in culture[67]. For instance, *OPRM1* specifically reached peak expression at day 21 and then decreased by day 28 (Supplementary Fig. 15a)[67]. When treating these cells with direct administrations of either our AAV-PHP.S-*mMORp-eYFP* or a control *CAG-tdTomato* virus at four different titers (1 × 10^9 – 10^12 gc/mL) around days 21–28 in culture, we observed broad transduction of cells with both viruses in nociceptors that increased

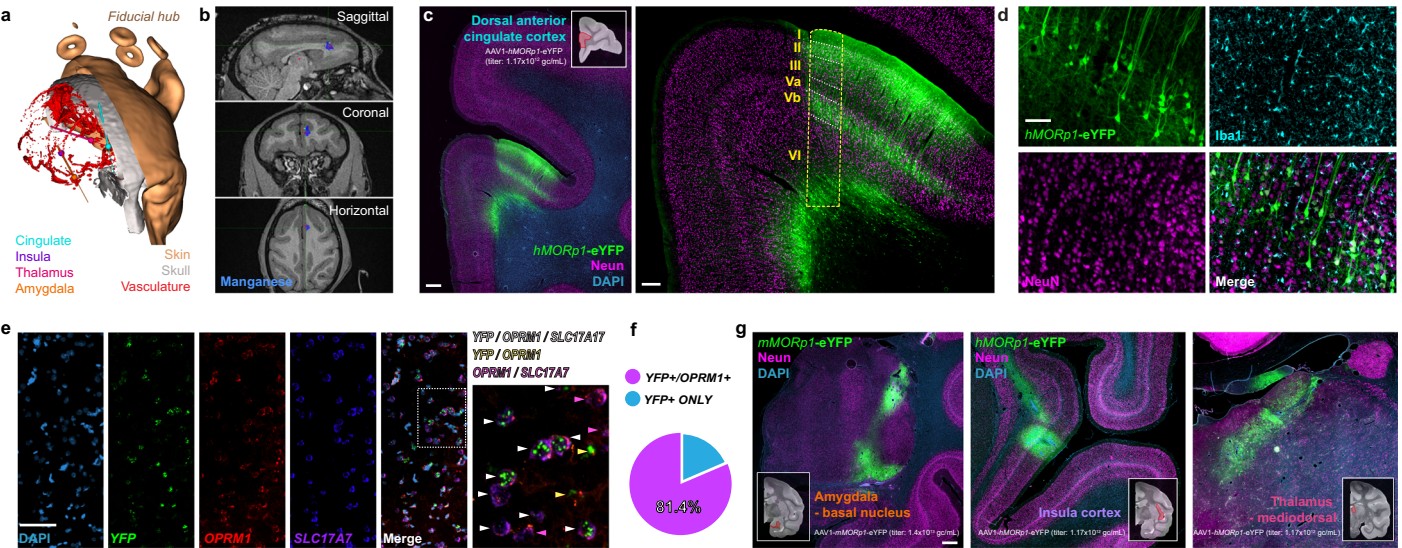

**Fig. 6 | *hMORp* viral constructs drive robust transduction of putative opioidergic cells in non-human primate neural tissue. a** 3D reconstruction of the skin, skull and underlying vasculature of a rhesus macaque for pre-operative intracranial injection planning. **b** Post-injection manganese-enhanced MRI for AAV1-*hMORp*-eYFP in vivo targeting accuracy assessement (virus mixed 1:100 with 100 mM manganese solution). **c** *Left*: Dorsal anterior cingulate cortex (dACC) expression of *hMORp*-eYFP; scale bar = 1000 μm. *Right:* dACC, higher magnification with cortical layer markers; scale bar = 200 μm. **d** Co-expression of *hMORp*-eYFP with neuronal (NeuN) but not microglial (Iba1) markers; scale bar = 100 μm. Staining for additional glial markers and relevant quantification to demonstrate transduction of predominantly neurons in macaque tissue is shown in Supplementary Fig. 13. **e** RNAscope FISH in dACC tissue for co-expression of *YFP*, *OPRM1*, and *SLC17A7* (VGLUT1) mRNA transcripts; scale bar = 100 μm, far right image = digital zoom of merged image. **f** Summary quantification of the total *EYFP* transcript positive cells quantified in sample regions of interest within dACC (upper, *N* = 1 dACC slice, *n* = 4 ROIs) with either *EYFP+/OPRM1−* (18.6%) or *EYFP+/OPRM1+*(81.4%) to demonstrate *hMORp*-eYFP virus expression specificity. **g** *mMORp*-eYFP expression within the basal nucleus of the amygdala, and *hMORp*-eYFP expression within the insular cortex and the mediodorsal thalamus following intracranial viral injections into these putative MOR expressing regions. Scale bar = 1000 μm.

with titer concentration, with robust expression of both reporters noted at the $1 \times 10^{12}$ gc/mL titer most prominently (Fig. 7a–d). By contrast, while expression of tdTomato signal remained prominent at both low and high viral titer concentrations in cultured cardiomyocytes, no eYFP signal was noted within these cultures following AAV-PHP.S-*mMORp-eYFP* treatment (Fig. 7e–h). These results indicate that our current iteration of *MORp* constructs appear to show selectivity for human cell types known to express MOR (and/or the *OPRM1* gene) when compared to cells shown to possess low expression of this receptor/gene, as demonstrated both from previous studies[68,69] and our own gene expression analyses of these two cultures, which found nociceptor cells to indeed express higher trending levels of *OPRM1* than the cardiomyocyte cells in which our transduction experiments were performed (Supplementary Fig. 15b).

## Discussion

The use of up and downstream genetic elements to target virally deliverable transgenes to select neural structures, subtypes of neuronal populations and circuits within them represents a burgeoning area of research within the gene delivery field, with several recent studies utilizing the integration of specific enhancer and promoter elements into their construct designs to target cells expressing unique genetic or molecular markers of interest[14,15]. Our results demonstrate that these principles can be extended to targeting mu opioidergic neuronal populations throughout the CNS and PNS with a high level of specificity and selectivity, and that the study of these transduced cells via effector transgenes yields behavioral and physiological readouts consistent with MOR manipulation. These findings present broad implications for the use of these viral tools in the study of the neurobiology of opioid-related fields and open the door to the translational use of such tools for potential therapeutic development and screening platforms.

The versatility of both the *mMORp* and *hMORp* constructs to transduce putative MOR+ neurons in multiple conserved brain structures across multiple species represents an important step towards expanding the use of different model systems in opioid circuit neuroscience investigations. Indeed, as alignment analyses of the sequences used to generate both of our constructs show (Supplementary Fig. 1e) both the *mMORp* and *hMORp* sequences demonstrate a high level of homology to the native *OPRM1/Oprm1* promoter sequence found in rat, macaque and human (no reference genome is currently available for the shrew), suggesting the possibility to significantly enhance and expand the investigation of the opioidergic system both within and outside of the murine research community. While five *Oprm1*-Cre transgenic mouse lines and a single *Oprm1*-Cre rat line[70] have been created, at present only one mouse line is commercially available (Jackson Labs, Strain:035574). These mice are haplo-insufficient or total knockouts for *Oprm1* when hetero- or homozygous for the Cre allele, respectively, which needs to be accounted for when designing experiments to investigate MOR function itself[29]. The use of AAVs and our *MORp* constructs can provide a more cost-effective and rapid method for labs to dissect the mu-opioid receptor system in vivo than expensive and slow rodent breeding schemes. Importantly, there are no other transgenic species models available that provide genetic access to MOR cell types. The fact that non-human primate and other small mammalian systems lack the tools or transgenic animals to achieve the same level of granularity in the study of opioid neurobiology has placed an overreliance on mouse systems that may not be the most appropriate model of studies for areas such as pain perception, opioid addiction-like behaviors, or cognitive processes involving the endogenous opioid system. The success of our constructs to transduce MOR+ cells in vivo in mouse, rat, shrew and macaque model systems demonstrate them to be potentially useful for researchers working across species that wish to target these populations for anatomical and functional studies.

Regarding the functional identity of the cells transduced with these viruses, and the utility of the effector transgenes encoded by them, our validation studies in mice suggest these cells to respond to

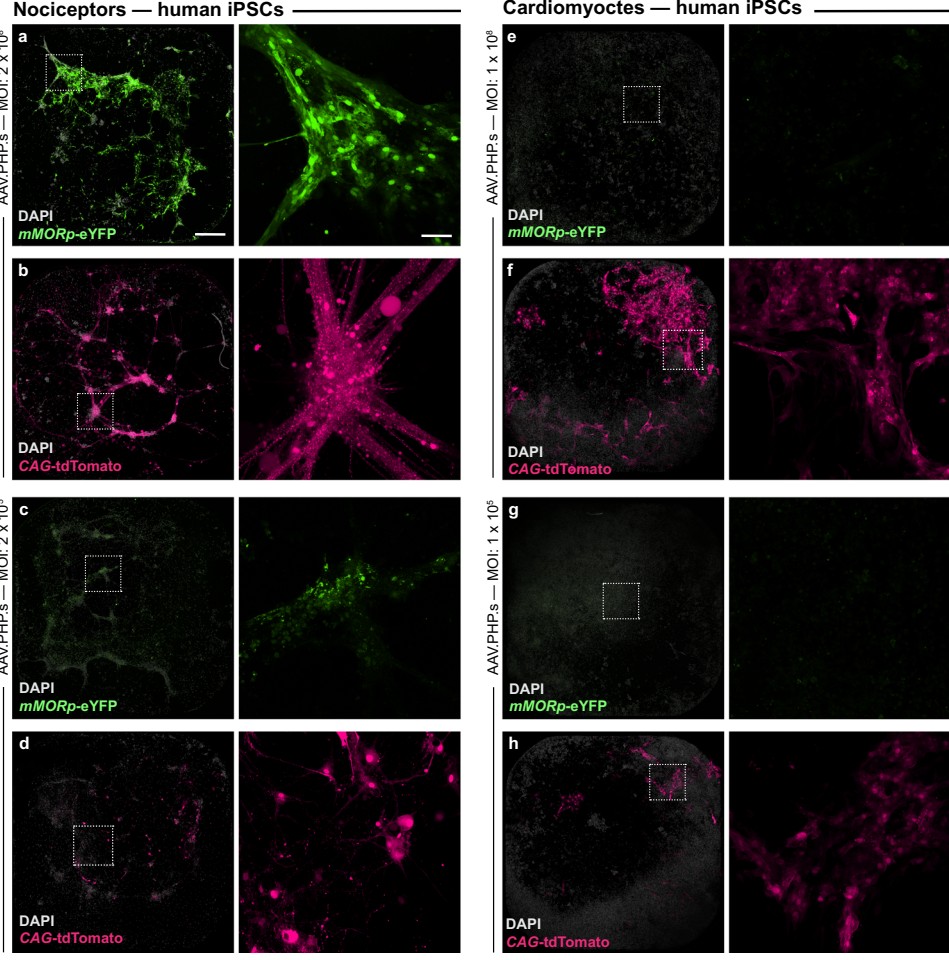

**Fig. 7 | *mMORp* viral transduction in human iPSC-derived MOR+ nociceptors.** Representative low (left, scale bar = 50 μm) and high (right, scale bar = 100 μm) magnification images of cultured human nociceptors treated with higher titer (1 × 10¹² gc/mL, MOI: 2 × 10⁸ [nociceptors], 1 × 10⁸ [cardiomyocytes]) AAV.PHP.S-*mMORp*-eYFP virus (**a**), or an AAV.PHP.S-*CAG*-tdTomato virus (**b**). High magnification sample regions are denoted in lower magnification images via a white box. **c, d** Similar low (left) and high (right) magnification images of cultured nociceptors treated with lower titer *mMORp*-eYFP (1 × 10⁹ gc/mL, MOI: 2 ×10⁵ [nociceptors], 1

×10⁵ [cardiomyocytes], **c**) or *CAG*-tdTomato virus (**d**). Images of cultured human cardiomyocytes treated with high titer *mMORp*-eYFP (**e**) or *CAG*-tdTomato (**f**) viruses, with regions boxed in white denoting high magnification sample areas shown on the right. Images of cardiomyocytes treated with low titer *mMORp*-eYFP (**g**) or *CAG*-tdTomato (**h**) viruses, at both low (left) and high (right) magnification. Evidence of *OPRM1* gene expression within cultured nociceptor cells is demonstrated in Supplementary Fig. 15.

both behavioral and pharmacological manipulation in a manner consistent with MOR agonism and/or antagonism. Within the spinal cord, the activation of mu receptors on cells in the DRGs has previously been demonstrated to produce a robust decrease in overall cellular activity and excitability[71–73], consistent with MOR's function as an inhibitory G protein coupled receptor[1], as well as a reduction in nocifensive behaviors following intrathecal administration of MOR agonists[74,75]. Our chemogenetic studies using an hM4Di encoding *mMORp* construct produced similar effects when administering either systemic CNO or intrathecal DCZ, consistent with a more selective modulation of MOR and putative MOR+ cells within the region. Noted increases in the number of calcium mediated events in the CeA and the overall total withdrawal score for behavioral responses in mice injected with our *mMORp*-GCaMP construct were also consistent with previous studies demonstrating pain-responsive CeA neurons to increase their excitatory activity in response to noxious stimuli[76–78], and the CeA in general to show an overall increase in molecular markers of neuronal activity in paradigms examining the effects of naloxone precipitated opioid withdrawal[78–81]. The potential shortcomings of a technique such as fiber photometry, which examines the bulk fluorescent signal from a population of cells transduced with a given calcium indicator, should be noted though. It is possible that the less than complete inhibition of

signal from *MORp*-GCaMP noted in these studies may be attributed to the competing activity patterns of different MOR+ subtypes known to be present within a region such as the CeA[82–84] in response to select types of stimuli or overall differences in MOR receptor density that may be present on transduced cells (which may alter the responsive of certain cells to drugs like morphine, thus influencing the fluorescent signal noted for the broader population of transduced cells). As such, care should be taken when designing experiments using viral tools such as ours to attempt to both target and restrict expression of certain transgenes to the desired population or subpopulation of cells most germane to a specific research question, an approach that our suite of constructs may help to further facility when using them in combinations with other viruses and transgenic animals lines, as demonstrated through additional studies we've presented here. Despite this though, current IHC and ISH data clearly show that it is likely our viral constructs are indeed transducing the desired target neuronal populations within these nuclei. Indeed, the restricted expression of our viral constructs to predominantly neuronal cells is in agreement with not only the overall design of these constructs, which included transcriptional elements known to repress expression in myeloid-like cells such as microglia[33], but also the natural predilections of AAVs to show greater transduction efficacy at neurons when

compared to glia[85], a detail supported by the findings presented in our glial marker staining panels across species. Additional gene expression assays used to address the concern of possible transduction events specifically in microglia seem to further support the higher preference for neuronal transduction inherent to our constructs, despite minimal upticks in *mMORp*-eYFP or *hMORp*-eYFP observed across cultured cell lines, which may be explained by cell line heterogenicity and/or minimal viral leak that may be present at higher titers. Despite the identity of the cells primarily transduced by our constructs across the select brain regions of interest discussed above appearing to be neurons, further testing will be necessary to ensure this to be the case in other regions of interest possessing MOR+ cells responsive to, and participatory in, the modulation of pain salient stimuli. Similarly, future testing will be equally necessary to determine which viral serotypes and titer concentrations will provide the greatest levels of transduction efficacy across brain regions, species, and sex variables outside of the sampling presented here. Outside of experiments in which the serotype of co-injected viruses was matched to reduce confounds from potential differences in transduction efficacy, the choice of serotype of a virus used for each of our studies was not informed by prior knowledge of viral tropism (ability of a virus to infect specific cells or tissue types[86]) inherent to any specific serotypes as reported across different brain regions or neuronal cell types within the literature[87]. This question of tropism and determination of the correct serotype to employ to best transduce a specific tissue or cell type is not unique to our viral constructs, however, as consensuses on the specific capsid proteins and construct elements that provide the greatest transduction efficacy within different tissue types and cell populations is still a matter of debate and investigation, requiring more meticulous documentation and data sharing amongst researchers across disciplines[88]. Examining the extent of the combinatorial approaches that can be taken with the use of our viruses with other transgenic animal lines and tools for the intersectional targeting of unique CNS and PNS cell populations[59] or the dissection of mu opioidergic circuits within the brain will thus require careful testing and planning when using our *MORp* constructs across any and all applications. Regardless, we believe that the results of these initial use case studies highlight the power and versatility of our *MORp* tools in the study of the opioidergic system not only in regards to the modulation of physiological pain responses, but also potentially in the study of chronic opioid addiction, withdrawal and other emerging complex behavioral features of pain and OUD as well.

While most of the work presented here delves into the applications of the *mMORp* constructs and their basic research utility in small animal models, the success of our *hMORp* and *mMORp* constructs to transduce MOR/*OPRM1*+ cells in rhesus macaque in vivo, as well as human iPSC-derived nociceptor cultures in vitro, respectively, suggests greater applications for these tools in translational research. Apart from potentially opening a door into the study of pain responsive neuronal populations in higher order animal models, the ability to directly target and manipulate opioidergic cells in human culture samples could greatly improve the specificity of drug screening studies for emerging therapeutics targeted to MOR in pharmaceutical research, as well as the prolonged effects of such compounds on specifically the cell populations of interest they are targeted to with chronic treatment paradigms. Indeed, the broadening of opioid research models is further extended to in vitro culture systems, as suggested in our exploratory use case for human-differentiated iPSCs. Patient- and disease-specific iPSC lines are currently under development as powerful in vitro disease models that provide unique exploration of nociceptor mechanisms of pain, including use in high-throughput screens for novel analgesics and as diagnostics to identify individuals at risk for transitioning from acute to chronic pain[89–92]. The great majority of peripheral nociceptors express MOR[93,94], and as demonstrated by the success of our *MORp* constructs to transduce human iPSC-derived nociceptors shown to express an apparent increase in gene expression for *OPRM1*, future uses could be leveraged to drive the expression of various gene editing tools or voltage indicators to be used in iPSCs for more complex phenotyping and screening studies. It is important to reiterate again, however, that our initial findings presented in these cultured human nociceptors only represent data acquired from the use of a single cell line shown to present an expression profile for genes such as *OPRM1* and several others indicative of a nociceptor and/or sensory-like cell types over the course of their development[67], with the expression of *OPRM1* examined in these cells shown to be somewhat variable at the time of collection for the samples used in our experiments. As such, while this does not detract from the clear evidence that *OPRM1* gene is present (and the *OPRM1* promoter active) within these cells, further study across other human-derived cell lines and/or differentiated cell types of both nociceptor and non-nociceptor lineage will be necessary to further demonstrate the utility of our viral tools for additional applications in such culture systems.

The growing promise of gene therapy for addressing highly intractable neuropathologies at the level of the CNS and PNS may warrant further thought on the potential for viral constructs such as our own and others utilizing design strategies aimed at providing selective genetic access to specific cell populations, circuits or brain regions, including MOR+ cells implicated in conditions such as chronic pain[95]. Complemented by the low immunogenicity and stable transgene expression associated with AAV-based vectors[15,96] and recent successes in the application of AAV-based therapeutics to treat several patient populations[97,98], further translationally minded studies could explore the utility of construct design schemas focused on targeting distinct elements identified within putative genes of interest as a mean for creating new and powerful treatments for myriad neurological disorders known to have distinct genetic or molecular signatures. Regardless of any future clinical application, it is ultimately our hope that these tools prove to be of interest to those in the opioid research community, and that their application aids in the expansion of the questions and complex features of opioid neurobiology that both clinical and basic researchers can pursue.

## Methods

All experimental procedures followed the National Institutes of Health guidelines, and protocols for studies conducted across all animal species were approved by the Institutional Animal Use and Care Committees of the University of Pennsylvania (in vivo viral testing and histological validation/specificity studies in mouse, rat, shrew and rhesus macaque, all mouse chemogenetic, fiber photometry and chronic and acute morphine dosing behavioral studies, and gene expression studies conducted in commercial human and mouse cell lines), Stanford University (viral production and in vitro testing in primary cultured neurons collected from rat) and Brigham and Women's Hospital/Harvard Medical School (in vivo viral testing and histological validation/specificity studies in mouse DRGs and gene expression analyses and histology conducted in human iPCS cells), respectively.

### Plasmid production

**Promoter selection.** A 1.9 Kb genomic region immediately upstream of the mouse mu opioid receptor gene (GenBank: AH005396.3) was analyzed for the presence of transcription factors using the PROMO database maintained by the Universitat Politecnica de Catalunya (Farre et al. 2003, 10.1093/nar/gkg605; http://www.lsi.upc.es/~alggen), and the Eukaryotic Promoter Database (EPD) for mmEPDnew, the Mus musculus (mouse) curated promoter database (https://epd.epfl.ch/mouse/mouse_database.php?db=mouse). Based on the results, a 1.5 Kb segment was selected (*mMORp*) and amplified from mouse genomic

DNA using cgcacgcgtgagaacatatggttggacaaaattc and ggcaccggtggaag ggagggagcatgggctgtgag as the 5' and 3' end primers respectively.

**Molecular cloning.** All *mMORp* and *hMORp* plasmids were constructed on an AAV backbone by inserting either the *mMORp* or *hMORp* promoter ahead of the gene of interest using M1uI and AgeI restriction sites. Every plasmid was sequence verified.

### Viral production

All AAVs (serotypes 1, 5, 8, PHP.S, and PHP.eB) were produced at the Stanford Neuroscience Gene Vector and Virus Core. In brief, AAV1 was produced by standard triple transduction of AAV 293 cells (Agilent). At 72 h post transduction, the cells were collected and lysed by a freeze-thaw procedure. Viral particles were then purified by an iodixanol step-gradient ultracentrifugation method. The iodixanol was diluted and the AAV was concentrated using a 100-kDa molecular mass–cutoff ultrafiltration device. Genomic titer was determined by quantitative PCR of the WPRE element. All viruses were tested in cultured neurons for fluorescence expression prior to use in vivo.

### Cell culture

**Primary neuronal cell cultures.** Primary cultured hippocampal neurons were prepared from P0 Sprague-Dawley rat pups (Charles River). CA1 and CA3 were isolated, digested with 0.4 mg/mL papain (Worthington), and plated onto glass coverslips pre-coated with 1:30 Matrigel (Becton Dickinson Labware). Cultures were maintained in a 5% $CO_2$ humid incubator with Neurobasal-A media (Thermo Fisher) containing 1.25% fetal bovine serum (FBS, HyClone), 4% B-27 supplement (Gibco), 2 mM Glutamax (Gibco) and 2 mg/ml fluorodeoxyuridine (FUDR, Sigma), and grown on coverslips in a 24-well plate at a density of 65,000 cells per well.

**HEK293 cell cultures.** HEK293FT cells (Thermo Fisher) were maintained in a 5% $CO_2$ humid incubator with DMEM media (Gibco) supplemented with 10% FBS (Invitrogen), and 1x Penicillin-Streptomycin (Invitrogen). They were enzymatically passaged at 90% confluence by trypsinization.

**Primary neuronal culture transduction.** 2.0 µg plasmid DNA was mixed with 1.875 µl 2 M $CaCl_2$ (final $Ca^{2+}$ concentration 250 mM) in 15 µL $H_2O$. To DNA-$CaCl_2$ we added 15 µL of 2× HEPES-buffered saline (pH 7.05). After 20 min at room temperature (20–22 °C), the mix was added dropwise into each well (from which the growth medium had been removed and replaced with pre-warmed minimal essential medium (MEM) and transduction proceeded for 45–60 min at 37 °C, after which each well was washed with 3 × 1 mL warm MEM before the original growth medium was returned. Neurons were allowed to express transfected DNA for 6-8 days prior to experimentation.

**Human and murine cell line cultures & transduction.** The human neuronal SH-SY5Y cells (American Type Culture Collection, ATCC CRL-2266) were seeded at $5 × 10^4$ cells/well density in 24-well plates in quadruplicate wells. The human astroglial line A172 (ATCC CRL-1620), human microglial C20 cells and murine N9 microglial cells (provided by Dr. Daniel Rader[99–101]) were seeded on 24-well plates at $1 × 10^4$ cells/well in quadruplicate wells, and cells were cultured with normal growth conditions (4.5 g/L glucose DMEM supplemented with 10% FBS and 1% antibiotic/antimycotic, incubated at 37 °C with 5% CO2). After 48 h, cells were transduced by adding $1 × 10^{11}$ gc/mL viral genomic content of the species appropriate AAV (MOI = $1 × 10^7$ [A172, C20, N9] and $2 × 10^6$ [SHSY5Y]) per well. After 5 days, media was exchanged with 1X PBS prior to imaging the transduced, cultured cells.

**Human iPSC-derived nociceptors and cardiomyoctes & transduction.** Human iPSC-derived nociceptors and cardiomyocytes cell cultures were produced as described previously[67,102,103]. Briefly, iPSCs (LiPSC-GR1.1, single donor, host sex: male, source: umbilical cord) established by the NIH Common Fund (Regenerative Medicine Program) were plated at 5000/well density and co-cultured with 2000 glial cells in 384-well plate, while iPSC-derived cardiomyocytes were plated 10,000/well without co-culture. Both nociceptor and cardiomyocyte derivations were the result of a single round of differentiation. Cultured cells were transduced by directly adding concentrated AAV viral particles at multiple titers ($1 × 10^9$, $1 × 10^{10}$, $1 × 10^{11}$, $1 × 10^{12}$ gc/mL) into well plates (MOI = $2 × 10^5$, $2 × 10^6$, $2 × 10^7$, $2 × 10^8$ [nociceptors] and $1 × 10^5$, $1 × 10^6$, $1 × 10^7$, $1 × 10^8$ [cardiomyocytes]). Cells were fed every two days with maintenance medium (Nociceptors: DMEM/F-12 [Glutamax, Cat# 10565018], N-2 Supplement [Cat# A1370701], B-27 Supplement [Cat# A3353501], BDNF [Cat# 248-BDB-050/CF], GDNF [Cat# 212-GD-050/CF], ß-NGF [Cat# 256-GF-100/CF], NT-3 [Cat# 267-N3-025/CF]; Cardiomyocytes: RPMI 1640 medium [Cat# 11835055], 1x B27 supplement with insulin [Cat# 17504-044]).

### Experimental animals

**Mice.** Adult male C57BL/6J wild type and male and female *Oprm1*^Cre:GFP knock-in/knock-out mice (The Jackson Laboratory, stock #00064 and #035574) of at least 8 weeks of age, and both adult male and female *Oprm1*^2A-Cre:Sun1-sfGFP knock-in mice (generously provided by the lab of Dr. Julie Blendy[10]) were used across all studies. Mice were group housed 2–5 individuals per cage, or individually for select experiments, and maintained on a 12-h reverse light/dark cycle (lights off at 0930 h and on at 1830 h) under controlled temperature (~20–25 °C) and humidity (~30–50%) levels. Animals were given *ad libitum* access to food and drinking water in primary housing and all secondary behavioral testing suites in the vivarium, unless otherwise noted for experimental manipulations.

**Rats.** Adult male Sprague-Dawley rats (Charles River, strain code #400) were used for viral, fluorescent in situ hybridization (FISH), and histological studies. Rats were individually housed in hanging wire cages and maintained on a 12-h light/dark cycle under temperature and humidity-controlled conditions. Animals were allowed *ad libitum* access to food and drinking water in all housing containers.

**Shrews.** Adult female Asian house shrews (*Suncus murinus*) were used for viral and histological studies. Shrews were bred and maintained at the University of Pennsylvania and derived from a Taiwanese strain initially supplied to investigators by the Chinese University of Hong Kong. Animals were individually housed in plastic cages (Innovive) and maintained on a 12/12-h light/dark cycle under temperature and humidity-controlled conditions, as above. Shrews were similarly granted *ad libitum* access to food (mixture of 75%, laboratory feline food [5003, Lab Diet] and 25% ferret food [5LI4, Lab Diet]) and drinking water in all housing containers.

**Rhesus macaque.** A single adult male rhesus macaque (*Macaca mulatta*), aged five years and weighing 6.3 kg, was utilized for viral and histological studies. Subject was housed in an enclosure exceeding regulatory standards to provide ample room for exercise and was maintained on a 12-h light/dark cycle in a temperature and humidity control environment with visual and touch access to conspecifics. Subject was provided with *ad libitum* access to drinking water, food, supplementary fruits and vegetables, and enrichment materials. No deprivations were enforced for the duration of the subject's time in university facilities. Health was monitored daily by veterinary and animal care staff.

### Stereotaxic surgery & viral delivery procedures

**Mice.** Adult mice (~8 weeks of age) were anesthetized with isoflurane gas in oxygen (initial dose = 3%, maintenance dose = 1.5%), and fitted

into Kofp stereotaxic frames for all surgical procedures. 10 μL Nanofil Hamilton syringes (WPI) with 33 G beveled needles were used to intracranially infuse AAVs into the medial prefrontal cortex (mPFC), central nucleus of the amygdala (CeA), the ventral tegmental area (VTA), and the dorsomedial striatum (DMS) at a rate of 100nL/min. The following coordinates were used, based on the Paxinos and Franklin mouse brain atlas (2019), to target these regions of interest: mPFC (from Bregma, AP = +1.25 mm, ML = ± 0.25 mm, DV = −2.0 mm), CeA (AP = −1.45 mm, ML = ± 2.91 mm, DV = −4.7 mm), VTA (AP = −3.2 mm, ML = ± 0.5 mm, DV = −4.7 mm), DMS (AP = +1.10 mm, ML = ±1.25 mm, DV = −3.0 mm). For initial immunohistochemistry and in situ hybridization in vivo validation, mice were bilaterally injected with ~300 nL of recombinant AAV1-*mMORp*-eYFP (titer: 6.9 × 10^{11} gc/mL) and given a 3–4-week recovery period to allow ample time for viral diffusion and transduction to occur. For cell counting studies, mice were injected with either a mix of recombinant AAV5-*mMORp*-hM4Di-mCherry (titer: 9.8 × 10^{11} gc/mL) and AAV5-*hSyn*-DIO-eGFP (Addgene, #50457, titer: 1.3 × 10^{13} vg/mL) at a ratio of 9ul:1ul, respectively (in *Oprm1*-Cre mice) or AAV5-*mMORp*-hM4Di-mCherry alone (in *Oprm1*^{2A-Cre}:Sun1 mice), with similar recovery periods allowed for these animals prior to tissue collection. For additional viral specificity and selectivity studies regarding the *MORp* promoter construct, C57BL/6J mice were injected with a mix of recombinant AAV5-*mMORp*-eYFP (titer: 2.7 × 10^{13}) and AAV5-*hSyn*-mCherry (Addgene, #114472, titer: 2.8 × 10^{13}) at a ratio of 1 μl:1 μl, with similar recovery periods applied prior to tissue processing as well.

For spinal cord chemogenetic studies, mice were injected intra-parenchymally at four sites (bilaterally at L4 and L5) with 400 nL of AAV1-*mMORp*-hM4Di-mCherry (titer: 3.1 × 10^{12} gc/mL) to drive transduction of inhibitory DREADD receptor in spinal cord neurons. Using a modified, minimally invasive method for microinjection in the mouse spinal dorsal horn without laminectomy[102], mice were first anesthetized with isoflurane gas in oxygen (initial dose = 4%, maintenance dose = 2%), the back shaved and wiped with alcohol and povidone-iodine followed by a 2 cm incision made through the skin, and then placed within a modified Kopf stereotaxic frame to stabilize the T13 and L3 spinal column vertebrae. Next, the spinal column was slightly arched upwards by ~30°, 0.2 mg/kg lidocaine was administered to the overlying muscle, which was then was gently dissected to expose the interspace above and below the T13 vertebrae, allowing direct access to the L3-5 spinal cord dorsal surface. A Nanofil 35 G beveled flexible steel needle on a 10 μL syringe (WPI) was connected to the stereotaxic arm and positioned over the midline of the T13 transverse process and moved medial-laterally ±0.3 mm and lowered down −0.25 mm to puncture through the spinal dura and place the tip of the needle in the substantia gelatinosa. The virus was then infused at a rate of 100 nL/min and the needle left in place for ~5 min post-injection before retracting and completing the remaining injections. The skin was sutured and wiped with povidone-iodine and an intraperitoneal (i.p.) injection of 5.0 mg/kg meloxicam was administered. Mice were observed for recovery and assessed for proper motor coordination over the next three days, with behaviorally testing then following approx. 5 weeks after surgery.

For fiber photometry studies, mice were injected with ~300 nL of recombinant AAV1-*mMORp*-GCaMP6f (titer: 3.69 × 10^{12} gc/mL) in the right CeA, followed immediately by the placement of a fiberoptic implant (~4.7 mm fiber, Doric Lenses) approximately 0.1–0.2 mm above the DV coordinate of the injection site (same as listed above). After setting the fiberoptic in position, MetaBond (Parkell) and Jet Set dental acrylic (Lang Dental) were applied to the skull of a mouse to rapidly and firmly fix the fiberoptic in place. In brief, after exposure of the skull, the bone was scored with a scalpel blade to provide grooves for the MetaBond to settle into and provide better adhesion. After lowering the fiberoptic into the craniotomy hole, the skull was cleaned with saline and then fully air dried. The MetaBond reagent was then

mixed and liberally applied over the skull and up and along the fiber-optic casing. Once dried, the MetaBond was then covered with a layer of Jet Set acrylamide to create a reinforced head cap, as well as to cover the exposed skin of the incision site. Mice were then given a minimum of 4 weeks to recover and allow for optimal viral spread and transduction throughout the region of interest prior to beginning in vivo calcium signal recordings. For all surgical procedures in mice, meloxicam (5 mg/kg) was administered subcutaneously at the start of the surgery, and a single 1 mL injection of sterile saline was provided upon completion. All mice were monitored for up to two days following surgical procedures to ensure the animals' proper recovery.

For cellular subtype labeling studies, C57BL/6J mice were injected with recombinant AAV8-*mMORp*-mCherry-IRES-Cre (7.0 × 10^{11} gc/mL) mixed with AAV9-*hDlx*-FLEX-eGFP (Addgene, #83895, titer: 2.2 × 10^{13} vg/mL) at a ratio of 9 μl:1 μl, respectively, and allowed 3-4 weeks for recovery and for viral transduction to occur throughout the mPFC and S1 regions. For additional recombinase-driven labeling studies, C57BL/6J mice were injected with a mix of recombinant AAV1-*mMORp*-FlpO (1.42 × 10^{11} gc/mL) and recombinant AAV9-*Ef1a*-fDIO-mCherry (Addgene, # 114471, titer: 2.4 × 10^{13} vg/mL) at a similar ratio of 9 μl:1 μl, respectively, and given a similar recovery period in order to allow for successful transduction and viral spread to occur prior to tissue collection and processing for histological analysis.

For pan CNS labeling studies, a 30 G insulin syringe was loaded with ~50 μl of recombinant AAV-PHP.eB-*mMORp*-eYFP virus (titer: 8.6 × 10^{12} gc/mL) was used to perform intravenous (retro-orbital) deliveries. C57BL/6J mice were deeply anesthetized with isoflurane gas mixed in oxygen (4%) and then quickly removed from the induction chamber for the injection procedure. Mice were placed in a prone position on a sterile surgical surface with a small nose cone to maintain anesthesia at a dose of 1.5% isoflurane. Using the index finger and thumb of the non-dominant hand, the skin above and below the right eye was gently drawn back to cause the eye to slightly protrude from the socket. With the dominant hand, the needle was inserted with the beveled side down at a ~30° angle (relative to the nose) into the medial canthus and through the conjunctival membrane. The total volume of the needle was then injected over a ~2 s period, after which it was slowly withdrawn, and the mouse allowed to recover. The injected eye was monitored for two days after the injection to ensure no sides of infection or other complications. For pan PNS labeling studies examining the DRGs, intracerebroventricular (i.c.v.) injections were performed on neonatal mouse pups at postnatal day 2–4 using either recombinant AAV-PHP.S-*mMORp*-eYFP (titer: 1.4 × 10^{13} gc/ml) or an AAV-PHP.S-CAG-tdTomato (Addgene, # 59462-PHP.S, titer: 2.1 ×10^{13} vg/mL) reporter virus. Pups were cryo-anesthetized for 1–2 min and then injected with about 4ul of either virus into the cerebral lateral ventricles. Six weeks after virus administration, animals were sacrificed and DRG tissue collections were performed as previously described[104,105].

**Rats.** Adult rats (~8 weeks of age) received an i.p. injected of an anesthetic cocktail (KAX, ketamine 9.0 mg/kg, Butler Animal Health Supply; xylazine 2.7 mg/kg, Anased; acepromazine 0.64 mg/kg, Butler Animal Health Supply) and tested to confirm loss of consciousness before being fitting into a Kofp stereotaxic frame. For histological and FISH studies, ~500 nL of recombinant AAV1-*mMORp*-eYFP (titer: 6.9 × 10^{11} gc/mL) was injected bilaterally into the CeA and VTA via a 10 μL Nanofil syringe fitted with a 33 G beveled needle (WPI) at a rate of 50 nL/min. The following coordinates were used, based on the Paxinos and Watson rat brain atlas (2006) these regions, respectively: CeA (from Bregma, AP = −2.4 mm, ML = ± 4.4 mm, DV = −8.1 mm) and VTA (AP = −5.4 mm, ML = ± 0.7 mm, DV = 8.2 mm). Metacam was administered subcutaneously immediately after surgery and for two consecutive days following surgery.

**Shrews.** Adult shrews (~12 weeks of age) were injected i.p. with KAX cocktail (ketamine 9.0 mg/kg, xylazine 0.27 mg/kg, acepromazine 0.064 mg/kg) and tested to confirm loss of consciousness before being fitted into a modified Kopf stereotaxic frame with the head in a ventroflexed position. A midline incision was made above the atlanto-occipital joint, the muscles were retracted, the joint capsule opened, and the ventral hindbrain was visualized similarly as previously described in rats[106]. For histological studies, similarly to what was described above for both mice and rats, ~300 nL of recombinant AAV1-*mMORp*-eYFP (titer: $6.9 \times 10^{11}$ gc/mL) was bilaterally infused into the AP/NTS via a 10 μL Nanofil syringe fitted with a 33 G beveled needle (WPI) using the following coordinates: AP = 400 μm rostral to the obex, ML = ± 300 μm, DV = 300 μm below the surface of the hindbrain. Metacam was administered i.p. both immediately after surgery and up to two days post-surgery.

**Rhesus macaque.** A single, five-year-old male macaque was utilized for immunohistochemistry and in situ hybridization studies. The subject underwent an initial surgical procedure to implant a dedicated headpost for a fiducial marker array system used in tandem with a MRI-guided neurosurgical system (Brainsight Vet, Rogue Research). MRI images for T1 and gadolinium enhanced T1 were acquired on a 3 T scanner (Siemens Tim Trio), and the images generated from this scan were then used to create 3D reconstructions of the skin, skull, brain and vasculature of the subject in the Brainsight software. Injection trajectories were planned using Brainsight to minimize the number of craniotomy holes needed to accommodate all target regions and to avoid both major blood vessels and ventricles. Four targets were selected for focal injections based on reported fMRI responses to capsaicin in macaques[107] and fMRI activity to noxious heat in humans:[108] the prefrontal cortex (PFC), insular cortex (Ins.), amygdala (Amyg.) and medio-dorsal thalamus (MDT). On the day of the first intraparenchymal injection surgery, the subject was sedated with an intramuscular (i.m.) injection of 4 mg/kg ketamine/0.025 mg/kg dex-medetomidine, intubated and maintained under inhaled isoflurane (0.5–2.0%). The analgesic buprenorphine (0.01 mg/kg) was administered subcutaneously (s.c.) preoperatively, with a concurrent dose of dexamethasone-SP (0.5 mg/kg intravenous). Intraoperatively, the subject's vitals were continuously monitored by veterinary care staff, and the anesthetized state maintained by anesthesia veterinary technicians. The surgical site was shaved and aseptically prepped, after which an "L" shaped incision was made over the midline and the rostral aspect of the right hemisphere. The skin was retracted to expose the right temporalis muscle, which was detached from its origin using a blunt tissue elevator to the level of the auricular cartilage. The periosteum was scraped, and craniotomy holes made over each injection site using the Brainsight drilling system. A 26 G injection cannula (Rogue Research) attached by polyethylene tubing to a microinfusion system (WPI) was backfilled with sterile mineral oil prior to loading ~40 μL of the AAV1-*hMORp*-eYFP virus (titer: $1.17 \times 10^{12}$ gc/mL) mixed 1:100 with a 100 mM sterile manganese solution for a final concentration of 1 mM. Virus was infused at all sites of interest at a rate of 200 nL/min, with final injection volumes as follows: 3.5 μL in right anterior insular cortex, 2.0 μL in right amygdala and 3.0 μL in right insular cortex. Upon the completion of surgery, the craniotomy holes were filled with bone wax, the superficial temporal fascia was anchored with 2-0 vicryl using a horizontal mattress pattern to replace the right temporalis muscle in the temporal fossa, and the skin was closed with 3-0 vicryl intradermal pattern. A postoperative manganese-enhanced MRI scan was conducted immediately following the surgery and linearly co-registered with pre-operative scans to assess targeting precision via imaging of the manganese contrast signal. Once completed, isoflurane was tapered, and the animal was extubated once consciousness was regained. Sustained release buprenorphine (0.12 mg/kg) was administered s.c. as well as an additional 0.5 mg/kg

Dex-SP, and the animal was returned to its enclosure. Dex-SP was administered on the following taper schedule: 1 mg/kg/day (1 day), 0.5 mg/kg/day (1 day), and 0.25 mg/kg/day (1 day). Activity and behaviors were monitored closely over several days following each procedure, with daily observations thereafter to ensure no adverse effects of the surgery on the animal's observable wellbeing. A second intraparenchymal brain injection surgery was also conducted ~3 months after the first, using the same approach described above, with the following exceptions. The "L" shaped incision was made over the midline and the rostral aspect of the left hemisphere, in order to target the left insula, mediodorsal thalamus and amygdala. The insula and mediodorsal thalamus were injected with 3.0 μL and 4.5 μL of the AAV1-*hMORp*-eYFP virus (titer: $1.17 \times 10^{12}$ gc/mL), respectively, and the left amygdala was injected with 4.5 μL of the AAV1-*mMORp*-eYFP virus (titer: $1.40 \times 10^{13}$ gc/mL).

## Immunohistochemistry
**Mice.** Animals were anesthetized using isoflurane gas and transcardially perfused with ice-cold 0.1 M phosphate buffered saline (PBS), followed by ice-cold 10% normal buffered formalin solution (NBF, Sigma, HT501128). Brains were quickly removed and post-fixed in 10% NBF for 24–48 hours at 4 °C, and then cryo-protected in a 30% sucrose solution made in 0.1 M PBS until sinking to the bottom of their storage tube (~48 h). Brains were then frozen in Tissue Tek O.C.T. compound (Thermo Scientific), coronally or sagittally sectioned on a cryostat (CM3050S, Leica Biosystems) at 40 μm or 50 μm, respectively, and the sections stored in 0.1 M PBS. Floating sections were permeabilized in a solution of 0.1 M PBS containing 0.5% Triton X-100 (PBS-T) for 30 min at room temperature and then blocked in a solution of 0.1% PBS-T and 10% normal donkey serum (NDS) for at least 60 min before being incubated with primary antibodies (1°Abs included: mouse anti-NeuN [1:1000, EMD Millipore, MAB377], chicken anti-GFP [1:1000, Abcam, ab13970], chicken anti-RFP [1:500, Novus, NBP1-97371], rabbit anti-Iba1 [1:1000, Wako, 019-19741], rabbit anti-Cre [1:1000, Synaptic Systems, 257 003], rabbit anti-MOR [1:500, Abcam, ab134054], mouse anti-CC1/anti-APC [1:200, Millipore Sigma, Op80], goat anti-PDGFRα [1:400, R&D, AF1062] and rabbit anti-GFAP [1:500, Agilent, Z033429-2]) prepared in a 0.1% PBS-T, 10% NDS solution for ~24 h at 4 °C. Following washing four times for 10 min in 0.1 M PBS, secondary antibodies (2°Abs included: Alexa-Fluor 647 donkey anti-mouse [1:500, Thermo Scientific, A31571], Alexa-Fluor 647 donkey anti-rabbit [1:500, Thermo Scientific, A31573], Alexa-Fluor 488 donkey anti-chicken [1:500, Jackson Immuno, 703-545-155], Alexa-Fluor 555 donkey anti-rabbit [1:500, Thermo Scientific, A31572], Alexa-Fluor 594 donkey anti-chicken [1:500, Jackson Immuno, 703-585-155] or Alexa-Fluor 594 donkey anti-goat [1:500, Thermo Scientific, A11058]) prepared in a 0.1% PBS-T solution were applied for ~24 h at 4 °C, after which the sections were washed again four times for 10 mins in 0.1 M PBS and then counter-stained in a solution of distilled water containing DAPI (1:10,000, Sigma, D9542). Fully stained sections were mounted onto Superfrost Plus microscope slides (Fisher Scientific) and allowed to dry and adhere to the slides before being mounted with Aqua-Poly/Mount solution (Polysciences, 18606) and cover slipped. All slides were left to dry overnight prior to imaging.

For spinal cord histology, anesthetized mice (Fatal-PLUS, Vortech Pharm.) were transcardially perfused with room temperature 0.1 M PBS, followed by 10% NBF. The brain, dorsal root ganglia (DRG, L3-L5), and spinal cord (lumbar cord L3-L5 segments) were dissected, post-fixed overnight (brains) or for 4 hrs (DRG or spinal cord) at 4 °C, and then cryoprotected in a 30% sucrose solution made in 0.1 M PBS. Samples were frozen in O.C.T. compound and then sections prepared (50 μm for brains, 30 μm for spinal cord, and 10 μm for DRG) using a cryostat. Sections were counter stained with DAPI and then mounted on glass slides using Fluoromount-G solution (Southern Biotech). Sections were covered slipped and left to dry overnight prior to imaging.

**Rats**. Animals were deeply anesthetized with KAX and transcardially perfused with ice-cold 0.1 M PBS, followed by ice-cold 4% paraformaldehyde (PFA) in 0.1 M PBS. Brains were removed and post-fixed overnight in 4% PFA at 4 °C, and then cryoprotected in a 25% sucrose solution in 0.1 M PBS for 72 h. Brains were then frozen in cold hexane and then stored at −20 °C until sectioning. Three series of 40 µm coronal sections containing the CeA and the VTA, respectively, were cut on a cryostat (CM3050S, Leica Biosystems) as described above. Floating sections were permeabilized in 0.5% PBS-T for 30 min at room temperature and then blocked in 0.1% PBS-T, 10% NDS for 60 min before anti-NeuN, anti-CC1, anti-GFAP, anti-Iba1 and anti-eGFP primary antibodies were applied in 0.1% PBS-T, 10% NDS for ~24 h at 4 °C at the same concentrations listed above for mouse tissue samples. Alexa-Fluor 647 donkey anti-mouse, Alexa-Fluor 488 donkey anti-chicken and Alexa-Fluor 555 donkey anti-rabbit secondary antibodies were applied in 0.1% PBS-T for ~24 h at 4 °C at the same concentrations listed above. Sections were washed and counterstained with DAPI, mounted with Aqua-Poly/Mount solution, cover slipped and left to dry overnight prior to imaging.

**Shrews**. Shrews were deeply anesthetized with i.p. KAX and transcardially perfused. Brains were removed and processed as described above. 30 µm coronal sections containing the AP/NTS were cut on a cryostat, and floating sections prepared and incubated with anti-eGFP and anti-Iba1 primary antibodies and Alexa-Fluor 488 donkey anti-chicken and Alexa-Fluor 555 donkey anti-rabbit secondary antibodies at the same concentrations and using the same time courses listed above for both mouse and rat tissue. DAPI counter stained sections were mounted with Aqua-Poly/Mount solution, cover slipped and left to dry overnight prior to imaging.

**Rhesus macaque**. The animal was sedated with 4 mg/kg ketamine and 0.025 mg/kg dexmedetomidine i.m., intubated and maintained at a deep plane of anesthesia using isoflurane at 2–3%. Immediately prior to perfusion, a 125 mg/kg dose of pentobarbital-based euthanasia solution was administered i.v. by veterinary staff. Upon confirmation of death, the animal was transcardially perfused with 10% NBF, and a post perfusion necropsy was performed by veterinarian and pathology staff to rule out any signs of irregularities or prior viral infection in the vital organs. The head was removed, and after extracting the fiduciary array head cap, the skull was cut away using a Dremel saw to fully expose the brain, after which the brain was removed from the skull. After cutting away the dura to expose the neural tissue, the brain was hemisected and each hemisphere then cut into six, 4 mm sections using a tissue blade. The sections were wrapped in gauze and submerged in 10% NBF, then left to post fix for ~24 h at room temperature. Brain sections were next cryoprotected by undergoing exposure to a sucrose gradient, submerging them in solutions of 10%, 20% and 30% sucrose in 0.1 M PBS, respectively, until sinking in each. Sections were then washed in 0.1 M PBS to remove excess sucrose and cut around the putative injection sites using a razor blade to create 2 cm by 2 cm blocks of tissue. Tissue blocks were placed into molds, covered with O.C.T., placed at −80 °C to freeze, and then stored at −20 °C until ready to be sectioned on a cryostat, taking ~50 µm coronal sections for immuno-histochemistry, and ~16 µm coronal sections for in situ hybridization studies. Floating sections were prepared for all IHC work (stored in 0.1 M PBS containing 0.05% sodium azide), while sections for ISH were direct mounted onto Superfrost Plus microscope slides and then stored at −80 °C until ready to be processed for RNAscope (see below). Floating tissue sections were permeabilized in 0.5% PBS-T for 30 min at room temperature and then blocked in 0.1% PBS-T, 10% NDS for at 60 min before adding and incubating with anti-eGFP, anti-NeuN, anti-CC1, anti-GFAP, anti-PDGFRα and anti-Iba1 primary antibodies at the same concentrations for tissue samples from all species (mouse, rat

and shrew) listed above for ~24 h at 4 °C. Sections were washed in 0.1 M PBS, and then incubated in Alexa-Fluor 488 donkey anti-chicken, Alexa-Fluor 647 donkey anti-mouse, Alexa-Fluor 555 donkey anti-rabbit and Alexa-Fluor 594 donkey anti-goat secondary antibodies at the same concentrations as listed above for mouse, rat and shrew tissue samples in 0.1% PBS-T for ~24 h at 4 °C. After washing in PBS again, sections were counterstained with DAPI, mounted on Superfrost Plus slides, cover slipped using Aqua-Poly/Mount solution, and left to dry overnight prior to imaging.

## RNAscope in situ hybridization

**Mice**. Animals were anesthetized using isoflurane gas in oxygen, and the brains were quickly removed and fresh frozen in O.C.T. using Super Friendly Freeze-It Spray (Thermo Fisher Scientific). Brains were stored at −80 °C until cut on a cryostat to produce 16 µm coronal sections of the mPFC, CeA and VTA. Sections were adhered to Superfrost Plus microscope slides, and immediately refrozen before being stored at −80 °C. Following the manufacturer's protocol for fresh frozen tissue for the V2 RNAscope manual assay (Advanced Cell Diagnostics), slides were fixed for 15 min in ice-cold 10% NBF and then dehydrated in a sequence of ethanol serial dilutions (50%, 70%, and 100%). Slides were briefly air-dried, and then a hydrophobic barrier was drawn around the tissue sections using a Pap Pen (Vector Labs). Slides were then incubated with hydrogen peroxide solution for 10 min, washed in distilled water, and then treated with the Protease IV solution for 30 min at room temperature in a humidified chamber. Following protease treatment, C1 and C2 cDNA probe mixtures specific for mouse tissue were prepared at a dilution of 50:1, respectively, using the following probes from Advanced Cell Diagnostics: *EYFP* (C1, 312131) and *Oprm1* (C2, 315841-C2). Sections were incubated with cDNA probes (2 h), and then underwent a series of signal amplification steps using FL v2 Amp 1 (30 min), FL v2 Amp 2 (30 min) and FL v2 Amp 3 (15 min). 2 min of washing in 1x RNAscope wash buffer was performed between each step, and all incubation steps with probes and amplification reagents were performed using a HybEZ oven (ACD Bio) at 40 °C. Sections then underwent fluorophore staining via treatment with a serious of TSA Plus HRP solutions and Opal 520 and 620 fluorescent dyes (1:5000, Akoya Biosystems, FP1487001KT, FP1495001KT). All HRP solutions (C1-C2) were applied for 15 min and Opal dyes for 30 min at 40 °C, with an additional HRP blocker solution added between each iteration of this process (15 min at 40 °C) and rinsing of sections between all steps with the wash buffer. Lastly, sections were stained for DAPI using the reagent provided by the Fluorescent Multiplex Kit. Following DAPI staining, sections were mounted, and cover slipped using Aqua-Poly Mount and left to dry overnight in a dark, cool place. Sections from all three regions were collected in pairs, using one section for incubation with the cDNA probes and another for incubation with a probe for bacterial mRNA (dapB, ACD Bio, 310043) to serve as a negative control.

For dorsal root ganglia (DRG) dissection studies, dissections were performed as previously reported[109]. First, mice were anesthetized with isoflurane gas for ~5 min and decapitated, after which the animals were perfused with 0.1 M PBS to remove all blood. Skin and muscles were removed from the back and around the spinal cord (SC). The vertebral columns of SC were exposed and placed on a tray of ice after removal. DRGs were exposed after roughly half of the vertebral columns were removed and the surrounding tissue cut away. Finally, collected DRGs were fixed in 4% PFA for 24 h at 4 °C, after which they were cryo-protected in a 30% sucrose dissolved in 0.1 M PBS for 24 h at 4 °C. DRG were frozen in O.C.T and then stored at −80 °C until ready for sectioning. DRG were sectioned at a thickness of 14 µm on a cryostat and mounted on slides. Sections then underwent RNAscope FISH using the manufacturer's V1 kit and associated protocol (Advanced Cell Diagnostics). Transcript species of interest were detected using the following probes from the manufacturer: *tdTomato*

(C2, 317041-C2), *eYFP* (C2, 312131-C2), *Rbfox3* (C3, 313311-C3), and *Oprm1* (C1, 493251).

**Rat.** Like the procedures described for mice above, rats were similarly anesthetized using isoflurane gas in oxygen, and brains removed and fresh frozen in O.C.T. using Super Friendly Freeze-It Spray. Brains were stored at −80 °C until cut on a cryostat to produce 20 µm coronal sections of the CeA, which were adhered to Superfrost Plus microscope slides and immediately refrozen before being stored back at −80 °C. The manufacturer's protocol for fresh frozen tissue for the V2 RNAscope manual assay was then followed as described above for mice tissue sections to process and stain the rat tissue, using the following probes from Advanced Cell Diagnostics: *EYFP* (C1, 312131) and *Oprm1* (C2, 410691-C2). Sections from the CeA were similarly collected in pairs, using one section for incubation with the cDNA probes and another for incubation with a probe for bacterial mRNA (dapB, ACD Bio, 310043) to serve as a negative control.

**Rhesus macaque.** The tissue preparation process for generating the macaque brain slices used in RNAscope experiments is described in the section above. 16 µm coronal sections of the PFC were adhered to Superfrost Plus microscope slides and immediately refrozen before being stored at −80 °C. Following the manufacturer's protocol for fixed frozen tissue for the V2 RNAscope manual assay, slides were washed in 0.1 M PBS to remove excess O.C.T., baked in the HybEZ oven for 30 min at 60 °C, and then post fixed for 15 min in ice-cold 10% NBF. Sections were then dehydrated in a sequence of ethanol serial dilutions (50%, 70%, and 100%), briefly air-dried, and then incubated with hydrogen peroxide solution for 10 min. Sections were washed in distilled water, and then were subjected to an antigen retrieval step. Utilizing a steamer (Hamilton Beach), sections were submerged in distilled water warmed to ~99 °C for 10 s and then switched into a container of 1x Target Retrieval Reagent for 5 min at ~99 °C. Sections were then cooled in room temperature distilled water for 15 s before being submerged in 100% EtOH for 3 min, and then finally dried in the HybEZ over at 60 °C for 5 min. Hydrophobic barriers were drawn around the tissue sections using a Pap Pen (Vector Labs), and after drying for 5 min, all sections were incubated at 40 °C for 30 min after treatment with several drops of Protease III solution. Following protease treatment, C1 and C2 cDNA probe mixtures specific for macaque tissue were prepared at a dilution of 50:1, respectively, using the following probes: *EYFP* (C1, ACD Bio, 312131) and *OPRM1* (C2, ACD Bio, 518941-C2). The remainder of the RNAscope protocol then proceeded as described above for mouse and rat tissue processing. As for the mouse and rat tissue samples, sections of macaque tissue from all regions of interest were collected in pairs, using one section for incubation with the cDNA probes and another for incubation with a probe for bacterial mRNA (dapB, ACD Bio, 310043) to serve as negative controls.

**Quantitative PCR and gene expression assays**
**qPCR for human and murine cell line cultures.** Cellular RNA was extracted with RNAzol (Sigma-Aldrich, R4533) according to manufacturer's protocol, and cDNA was synthesized from 1 ug RNA (Applied Biosystems, #4374966). cDNA was diluted 1:10 and assessed for mRNA transcript levels by qPCR with SYBR Green Mix (Applied Biosystems, #A25741) on a QuantStudio7 Flex Real-Time PCR System (Thermo Fisher). Oligonucleotide primer sequences for target and reference genes are as follows: human_ACTIN (forward: AAGTCCCTCACCCTCC CAAAAG, reverse: AAGCAATGCTGTCACCTTCCC), human_GAPDH (forward: AACCTGCCAAGTATGATGACATCA, reverse: TGTTGAAGTC ACAGGAGACAACCT), human_OPRM1 (forward: ACTGATCGACTTGT CCCACTTAGATGGC, reverse: ACTGACTGACTGACCATGGGTCGGA CAGGT), mouse_GAPDH (forward: AACGACCCCTTCATTGACCT, reverse: TGGAAGATGGTGATGGGCTT), mouse_L30 (forward: ATGGT

GGCCGCAAAGAAGACGAA, reverse: CCTCAAAGCTGGACAGTTGTTG GCA), mouse_OPRM1 (forward: CTGCAAGAGTTGCATGGACAG, reverse: TCAGATGACATTCACCTGCCAA), YFP (forward: TGCTTC GCCCGCTACCC, reverse: ATGTTGCCGTCCTCCTTGAAG). The fold change in the target mRNA abundance, with respect to SH-SY5Y cells and normalized by the reference gene GAPDH, was calculated using the $2^{-\Delta\Delta C_T}$ method[101].

**qPCR for human iPS nociceptor and cardiomyocyte cultures.** Cellular RNA was extracted with TRIzol (Invitrogen, 15596026) according to manufacturer's protocol, and cDNA was synthesized from 1 ug RNA (Invitrogen, 18080051). cDNA was assessed for mRNA transcript levels by qPCR with SYBR Green Mix (Bio-Rad, 1725121) on a CFX Opus 384 Real-time PCR System (Bio-Rad). Oligonucleotide primer sequences for target and reference genes are as follows: human_OPRM1 (forward: ACCCTGACTCAACTGGATGG, reverse: CCCCATAGCTACAGTCTGCA), human_GAPDH (forward: GTCTCCTCTGACTTCAACAGCG, reverse: ACCACCCTGTTGCTGTAGCCAA). The fold change in the target mRNA abundance (analyzed with respect to the hiSP cardiomyocytes and normalized to GAPDH) was calculated using the ΔΔCT method. Multiple wells of each cell type were combined to obtain enough cDNA to complete each run, with three separate runs conducted in total. Transcriptomic data used to demonstrate the relative expression of *OPRM1* in nociceptor-like cells differentiated from the LiPSC-GR1.1 line is presented as normalized transcript counts using data taken from screens previously reported in Deng et al. 2023, with additional information regarding these datasets available upon request via the communicating authors[67].

**Imaging and quantification**
All mouse, rat, shrew and macaque tissue expressing AAV1-*mMORp*1-eYFP, AAV5-*mMORp*-hM4Di-mCherry or AAV1-*hMORp*-eYFP processed for IHC and FISH were imaged on a Keyence BZ-X all-in-one fluorescent microscope at 48-bit resolution using the following objectives: PlanApo- λ x4, PlanApo- λ x20 and PlanApo- λ x40. Co-localization of eYFP or mCherry signal to neurons stained with antibody amplified signal or cDNA probes of interest was achieved by adjusting the exposure time and overall gain for the green, red, far red and blue channels to visualize cells with both high and low signal in x4 magnified snapshots and x20 and x40 magnified Z-stack images while correcting for oversaturation. Similar metrics and equipment were used when imaging mouse tissue from use case experiments expressing all other constructs to validate overall viral transduction efficacy. All image processing prior to quantification was performed with the Keyence BZ-X analyzer software (version 1.4.0.1). Imaging for mouse tissue expressing AAV.PHP.S-*mMORp*-eYFP and processed for FISH was performed with a Zeiss LSM 710 confocal microscope using a 20x objective for cell quantification and a 40x oil-immersion objective for viewing transcript markers at high resolution. Human iPS cells treated with AAV.PHP.S-*mMORp*-eYFP and processed for IHC were imaged using a Zeiss LSM 710 confocal microscope via a 10x objective and a 20x objective for higher magnification. All image acquisition and initial pre-processing performed on Zeiss microscopes was conducted via the use of Zeiss Zen microscopy software (v.3.4).

Quantification of viral efficacy and selectivity in IHC processed tissue from mice was performed via manual counting of TIF images in Fiji (ImageJ, 2.3.0/1.53q) using the Cell Counter plugin or Photoshop (Adobe, 2021) using the Counter function. Counts were made using x20 magnified z-stack images of a designated regions of interest (ROI) around injection sites from both the left and right hemispheres and reported as total cells in each ROI positive for mCherry and anti-Cre, or eYFP and mCherry, signal co-localized to the same cell body, as denoted by DAPI staining. Total counts for these populations were averaged across all hemispheres with positive viral transduction within targeted ROIs from 1 male and 2 female mice for the CeA and VTA,

1 male and 1 female for the mPFC, and 1 male and 1 female for the DMS in the *Oprm1*-Cre:GFP line, and 2 male mice for the CeA and VTA, 1 male for the mPFC and 2 females for the DMS in the *Oprm1*-2A-Cre:Sun1 line, as well as 2 male C57BL/6J mice for *mMORp* and *hSyn* transduction comparisons. Similar quantification metrics were applied for the quantification of all glial marker panels conducted across mouse, rat, macaque and shrew tissue.

For FISH images, TIFs of mouse, rat and macaque tissue sections treated with the dapB negative control probe for each pair of slides were used to determine brightness and contrast parameters that minimized observation of bacterial transcripts and auto fluorescence, and these adjustments were then applied to images from the experimental sections which were treated with the cDNA probes. Adjusted experimental images were then analyzed within separate, 20x ROIs across all structures of interest in mouse, rat and macaque tissue sections. Cells in these ROIs were identified using DAPI-stained nuclei, and the total number of cells in each region were counted. Cells were then appraised for the presence of *EYFP* and *OPRM1* transcript signal to determine the total number of cells labeled for these probes either alone or in combination. Transcripts were readily identified as round, fraction delimited spots over and surrounding DAPI-labeled nuclei. Total counts for the number of cells showing co-localized signal were summated across ROIs, and replicate counting was performed and by 4 separate individuals for the macaque images, with total counts of all ROIs from each experimenter averaged and the resulting values reported. FISH images taken of mouse DRG sections were quantified manually in Fiji using ROIs to define the quantified area. AAV+ and Marker+/AAV+ neurons were counted as those with more than five dots (or transcripts) per ROI, and counts were confirmed as reasonable estimates by comparison to counts labeled with nuclei marker DAPI and pan neuron marker Rbfox3.

## Behavioral testing

All experiments took place during the dark phase of the cycle (0930–1830 h). Group and singly housed mice were allowed a 1–2-week acclimation period to housing conditions in the vivarium prior to starting any behavior testing. Additionally, three to five days before the start of testing, mice were handled daily to help reduce experimenter-induced stress. On test days, mice were brought into procedure rooms ~1 h before the start of any experiment to allow for acclimatization to the environment. Mice were provided food and water *ad libitum* during this period. For multi-day testing conducted in the same procedure rooms, animals were transferred into individual "home away from home" secondary cages ~1 h prior to the start of testing and were only returned to their home cages at the end of the test day. All testing and acclimatization were conducted under red light conditions (<10 lux), with exposure to bright light kept to a minimum to not disrupt the animals' reverse light cycle schedule. Equipment used during testing was cleaned with a 70% ethanol solution before starting, and in between, each behavioral trial to mask odors and other scents.

## Von Frey filament touch test.

To evaluate mechanical reflexive sensitivity, we used a logarithmically increasing set of 8 von Frey filaments (Stoelting), ranging in gram force from 0.07 to 6.0 g. Mice were placed on a metal hexagonal-mesh floored platform (24 in x 10 in) within a transparent red cylinder (3.5 in x 6 in). The filaments were then applied perpendicular to the left plantar hind paw with sufficient force to cause a slight bending of the filament. A positive response was characterized as a rapid withdrawal of the paw away from the stimulus within 4 s. Using the Up-Down statistical method[110], the 50% withdrawal mechanical threshold scores were calculated for each mouse and then averaged across the experimental groups. For the two rounds of inhibitory chemogenetic testing conducted using this assay, mice were administered a dose of either the drug DCZ intrathecally at a dose of

10 pg (first round, 5 weeks post-surgery) or CNO systemically at a dose of 3 mg/kg by body weight (second round, 8 weeks post-surgery, 3 weeks post-DCZ trials), with animals being returned to their home cages for ~30 min after injections to allow for complete absorption of drugs prior to the start of any behavioral testing.

## Hotplate thermal tests (static & dynamic).

To evaluate thermal sensitivity, we used an inescapable hotplate set to 50–52.5 °C. The computer-controlled hotplate (6.5 in x 6.5 in floor, Bioseb) was surrounded by a 15 in high clear plastic chamber and a web camera was positioned at the front of the chamber to continuously record animals to use for post hoc behavioral analysis. For the static temperature tests conducted for chemogenetic inhibition studies, mice were administered either a 10 pg intrathecal dose of DCZ or a 3 mg/kg systemic dose of CNO (Fig. 3c), and then returned to their home cages for ~30 min to allow for complete absorption of drugs prior to the start of behavioral testing. Mice were then gently placed on the hotplate floor and removed from the chamber after 30 secs. Behaviors were scored in real-time for the latency to the first paw withdrawal, the total duration of paw licking, and the total number of jumps/escape events observed over the entire trial period.

## Formalin chemical test.

To evaluate chemical induced nocifensive responses, mice received an intraplantar 10 µL injection of a 4% formalin solution. Mice were lightly restrained in a prone position with the left hind limb held between the thumb and fore finger of the non-dominant hand, while a formalin-loaded glass Hamilton syringe with a 30 G needle was held in the dominant hand. The needle was inserted in the middle of the left hind paw at a ~30° angle and the formalin solution was injected over a ~2 s period. Next, the mice were quickly placed within transparent red cylinder on a stage with a clear plastic floor, with a mirror angled at 45° underneath, while a web camera recorded behavior from both underneath and to the side of the apparatus for 60 min. These videos were then blind scored in 2-min bins for reflexive paw flinches, paw guarding, paw licking and jumps. Time course data was summed into the classically defined phases of the formalin test:[111] 1st phase, interphase and 2nd phase, which reflect different engagements of peripheral and central sensitization mechanisms. For all chemogenetic inhibitory studies, mice were administered a systemic injection of CNO i.p. (3 mg/kg body weight), after which they were returned to their home cage for ~30 min prior to the start of testing to allow for full absorption of the drug to occur.

## Air puff stimulation.

Response to an unexpected stimulation was assessed during fiber photometry testing to validate the overall success of viral transduction of CeA neurons in mice. Mice were placed in transparent red cylinders on top of a metal hexagonal-mesh floored platform as described above, with a web camera position to record them continuously during testing. Using a canister of compressed air (Century Cleaning Duster), a ~1 s long blast was applied to the underside of the animal to elicit a startle response. This process was repeated 10 times, with an inter stimulation interval of ~1 min. Behavioral responses time-locked to calcium mediated events were scored posthoc using the acquired video footage.

## Hot water hind paw stimulation test.

To evaluate responses to acute, noxious thermal stimulations during fiber photometry testing, we used a hot water hind paw application protocol described previously[112]. In brief, animals were placed in transparent red cylinders placed on top of a metal hexagonal-mesh floored platform, while a small hot plate was used to heat distilled water to ~70–72 °C. Using a 1 mL syringe, a drop of the hot water was quickly applied to the underside of the left plantar hind paw (temp = 55–57 °C at time of application). This process was repeated for a total of 10 applications, with each droplet applied at a 1 min interval. Animals were continuously recorded by a web camera

positioned to face the front of the cylinder in which the animal was housed, and all relevant behaviors time-locked to calcium mediated events were scored post-hoc using the video footage. Responses to these noxious stimuli were also tested following acute i.p. administration of morphine (~10 mg/kg body weight). After injection, animals were placed back in their home away from home cages for 30 min to allow for complete absorption of the drug. Hot water hind paw stimulation testing then proceeded as described above in the naïve condition.

**Chronic morphine drinking & precipitated withdrawal testing.** Withdrawal behaviors were assessed during fiber photometry studies by use of a forcing morphine drinking paradigm in mice, achieved by substituting their drinking water for glass stopper bottles (Braintree Scientific) containing morphine or saccharin laced water. Morphine drinking animals were supplied with a 0.3 mg/mL morphine, 0.2% saccharin solution for 3 days, followed 0.5 mg/mL morphine, 0.2% saccharin for 4 days prior to inducing precipitated withdrawal with an i.p. injection of naloxone hydrochloride. Control animals were supplied with 0.2% saccharin laced water only for the duration of testing. After the final day of forced drinking, animals were injected with 3 mg/kg body weight naloxone and then immediately placed in a high-walled Plexiglass container (16.5 × 16.5 cm) on top of a metal hexagonal-mesh floored platform. Two web cameras were set up to continuously record the animals from separate angles, and withdrawal related behaviors were assessed for ~20 min by an experimenter blinded to experimental condition using BORIS software[113]. A global withdrawal score was calculated for each mouse by assigning one point for each instance of jumping, paw tremors, head shakes, wet dog shakes, genital licking, digging at the sides of the enclosure, and defensive treading; one point for each fecal bolus; and one point for each minute in which a resting tremor or teeth chatter was present. Total calcium-mediated events recorded by Synapse software during withdrawal behavioral recording were also used as a read out for overall change in CeA activity in response to naloxone injection.

**In vivo fiber photometry recordings and analysis**
Optical recordings of GCaMP6f fluorescence were acquired using an RZ10x fiber photometry detection system (Tucker-Davis Technologies), consisting of a processor with Synapse software (Tucker-Davis Technologies), and optical components (Doric Lenses and ThorLabs). Excitation wavelengths generated by LEDs (460 nm blue light and 405 nm violet light) were relayed through a filtered fluorescence minicube at spectral bandwidths of 460–495 and 405 nm to a pre-bleached mono fiberoptic patch cord connected to the implant on top of each animal's head. Power output for the primary 460 nm channel at the tip of the fiberoptic cable was measured at ~25–30 mW. Single emissions were detected using a femtowatt photoreceiver with a lensed cable adapter. Signal in both 460 and 405 nm channels were monitored continuously throughout all recordings, with the 405 nm signal used as an isosbestic control for both ambient fluorescence and motion artifacts introduced by movement of the fiberoptic implant. Wavelengths were modulated at frequencies of 210–220 and 330 Hz, respectively, and power output maintained at a range of ~20–60 mA for the 460 channel and ~20 mA for the 405 channel, with a DC offset of 3 mA for both light sources. All signals were acquired at 1 kHz and lowpass filtered at 3 Hz. Mice were housed and handled as described above, with the addition of a 5 min session each handling day during which the mice were hooked up to the fiberoptic patch cord to allow them to become accustomed the tethered cable. On testing days, mice were connected to the photometry system, and following a 1–2 min habituation period, mice were placed into the respective equipment for each of the test described above. Following testing, all mice were perfused, and the tissue was assessed for proper viral targeting and

transduction efficacy, as well as optic fiber placement via immunohistochemistry (see above section for details).

Analysis of the GCaMP signal was performed with the use of the open source, fiber photometry analysis MATLAB software suite, pMAT[114]. Using pMAT, bulk fluorescent signal from both the 460 and 405 channels were normalized to compare differences in calcium-mediated event metrics for both the total duration of a recording (frequency) and at select events using peri-event time histogram (PETH) analyses locked to specific behaviors designated by the application of an external transistor-transistor logic (TTL) input (amplitude and area under the curve) across groups, with the 405 channel serving as a control signal. Linear regression was used to correct for the bleaching of signal for the duration of each recording, using the slope of the 405 nm signal fitted against the 460 nm signal. Detection of GCaMP-mediated fluorescence is presented as a change in the 460 nm/fitted 405 nm signal over the fitted 405 signal ($\Delta F/F$). Peak analysis of calcium-mediated events to determine frequency in precipitated withdrawal testing was performed by running the normalized, filtered signals generated by pMAT through Clampfit 10.6 software and performing threshold matched event detection analyses. The threshold value was set to examine the amplitude of an imputed, positive going "spikes" that exceeded the baseline noise of the combined $\Delta F/F$ signal (typically set to exceed a value of 1–2).

**Drugs and delivery**
For chemogenetic studies, deschloroclozapine (DCZ dihydrochloride, water soluble; HelloBio, HB9126) was delivered intrathecally at a dose of 10 pg in 5 µL saline, and clozapine N-oxide (CNO dihydrochloride, water soluble; HelloBio HB6149) was delivered i.p. systemically at a dose of 3.0 mg/kg body weight. For fiber photometry behavioral testing, morphine sulfate (Hikma) was delivered acutely i.p. at a dose of 10 mg/kg body weight, or chronically at concentrations of 0.3 mg/mL and 0.5 mg/mL by mixing into drinking water along with 0.2% saccharin sodium hydrate (Sigma) in the animals' drinking water. Precipitated withdrawal was inducing during photometry studies by administering naloxone hydrochloride (HelloBio, HB2451) i.p. at a dose of 3 mg/kg body weight.

**Statistical analyses and reproducibility**
The number of animals used in each experiment were predetermined based on analyses of similar experiments in the literature and supplemented as needed based on observed effect sizes. All data are presented as mean ±the standard error of the mean (SEM) for each group, and all statistical analyses were performed using Prism 9 & 10 software (GraphPad Software). We used primarily male mice for all relevant histological cell counting analyses, as well as all chemogenetic and fiber photometry based behavioral studies, and thus did not attempt to assess sex related differences in these use case experiments. Female shrews were used in cross species histological studies, but as those were primarily qualitative evaluations of the efficacy of our viruses, and no meaningful statistical analysis was performed.

For all IHC-based quantification of virally transduced cells in either *Oprm1*[Cre] or *Oprm1*[2A-Cre]:Sun1 mice, as well as transduced cells in C57BL/6J mice either co-injected with the *hSyn*-mCherry reporter virus or stained for select glial markers, total cell counts are presented as either parts of a whole for comparing *mMORp*-mCherry+/anti-Cre+ vs. *mMORp*-mCherry+/anti-Cre−, *mMORp*-eYFP+/mCherry+ vs. *mMORp*-eYFP+/mCherry−, or *mMORp*-eYFP+/glial marker+ vs. *mMORp*-eYFP+/glial marker− stained populations, and summary statistical data for total cell counts and groupings based on transgene reporter, anti-Cre or glial antibody staining signal are reported as well. Similar statistical analyses and reporting were performed for ISH-based quantification of rat and macaque tissue sections as well. ISH-based analyses of total transduced cells within mouse DRG following AAV-PHP.s-*mMORp*-

eYFP injections were conducted via unpaired two-tailed t-tests with Welsh's correction and reported as total *Oprm1+/eYFP+* out of all *Oprm1+* cells quantified. All specificity/efficiency experiments for the validation of *mMORp* viruses were run from single experiments, using a necessary number of animals, imaged tissue sections and regions of interest to sufficiently power these assays for statistically analysis. For behavior experiments, data comparing all metrics assessed between *mMORp*-hM4Di+ vs. mCherry+ controls was analyzed using repeated-measures two-way ANOVAs along with Bonferonni multiple comparison post hoc tests. Corrected P values are reported in the text as needed. For comparing behavioral results from *mMORp*-GCaMP+ vs. *mMORp*-eYFP+ controls across tasks, two-tailed unpaired t-test analyses were performed on data regarding the frequency of imputed calcium-mediated events, changes in the amplitude values, and computed area under the curve (AUC) for select time-locked events from in vivo fiber photometry recordings (all of which consisted of comparisons between only two groups/conditions), with corrected P values presented in the text. For gene expression analyses, when comparing *OPRM1, Oprm1* and *EYFP* levels across different mouse and human cell lines (SHSY5Y, C20, A172 and N9), the Ct values for these genes of interest were normalized to the housekeeping gene (*GAPDH*) Ct value (for each well and for averaged values), and then displayed relative to the expression values determined for the SHSY5Y line for *EYFP* and *OPRM1* (input normalized to 1 ug of RNA). One-way ANOVAs along with Dunnett's multiple comparison post hoc tests were used to compare expression values across all cell lines for both genes of interest relative to SHSY5Y expression. Similarly, for analyses conducted of *OPRM1* gene expression in differentiated hiPSC nociceptor and cardiomyocyte cultures, Ct values were normalized to *GAPDH* and then displayed relative to the expression values determined for *OPRM1* in the cardiomyocytes, with final summary data being presented as the average of normalized values calculated for three rounds of gene expression assays run on separate cohorts of cultured cells. For all analyses, significance levels were set at an alpha of 0.05. Detailed statistics are provided within the text and figure legends.

For all mouse, rat and shrew studies, representative images of viral expression within brain regions of interest for each species were selected following one to four sets of successful viral injections (mouse = four separate sets of injections to satisfy all IHC and ISH-based studies, rat = two separate sets for IHC and ISH studies, and shrew = a single set of injections for IHC studies along). Banked tissue sections for all species were used across multiple rounds of histology, using multiple sections containing each region of interest for individual IHC and ISH experiments, with the sections showing the best staining for either antibody or probe expression used as representative images. Specifically for mouse based studies, in C57BL/6J mice: one set of injections and imaging was conducted for initial validation of the *MORp*1 virus, one for generating tissue used to show overall transduction efficiency and spread across injected areas of interest and staining with anti-MOR Abs, one set for ISH validation and one set for all glial panel staining based studies; in *Oprm1*[Cre] mice: one set for generating tissue for specificity/selectivity assays; in *Oprm1*[2A-Cre] mice: one set for generating tissue for specificity/selectivity assays. Similar transduction efficacy was noted in mouse tissue when performing histological validation across all injection sets targeting CNS structures. For spinal cord studies in C57BL/6J mice, a single set of intraspinal injections was performed on a group of three mice to generate representative images of AAV1-*mMORp*-hM4Di-mCherry expression in L4 spinal cord region. Representative images shown for AAV1-*mMORp*-GCaMP6f in mouse CeA were taken from a single set of injections into a cohort of twelve C57BL/6J mice, with the image taken from an animal showing exemplary viral expression in the CeA and placement of the fiber optic cannula. Similar results were noted in other animals for both the accuracy of cannula placement and viral expression, as none were

omitted from analyses due to issues with either of these factors. For intersectional/recombinase based studies using the *mMORp* viruses in C57BL/6J mice, representative images shown are the result of two separate sets of successful injections of all viruses into each region of interest, across all of which expression was notable at all injection sites. Two separate sets of successful injections of the AAV.PHP.eB-*mMORp*-eYFP were performed retro-ortbitally in C57BL/6J mice to generate both the coronal and sagittal sections of all CNS and spinal cord tissue used to demonstrate the transduction efficiency of this capsid variant version of our virus, with the more exemplary images from the two injected cohorts (approx. 3-4 mice per set) used as representatives. Two sets of injections are performed on a single macaque, and two rounds of IHC (one for general staining for *hMORp-eYFP* expression, and one for glial marker staining) and a single round ISH were conducted on collected tissue to generate representative images, showing similar robust staining for each round to that observed in all lower phylogenic organisms. Gray scale images of dACC transduced neurons in the macaque tissue were taken from a separate round of IHC performed on additional slices of the dACC taken from the same macaque. For human iPSC and in vitro transduction studies, representative images were taken from a single round of viral transductions performed across two well plates of either differentiated cardiomyocytes or nociceptors. Similarly, all representative images from cultured SHSY5Y, C20, A172 and N9 cells transduced with either the *mMORp*-eYFP or *hMORp*-eYFP viruses were taken from a single round of transductions in separate well plates containing cells from each line, with the images of wells containing the most exemplary transduced cells used as representatives.

### Resource sharing
All viruses detailed in this manuscript are available for academic use from the Stanford Gene Vector and Virus Core (https://neuroscience.stanford.edu/research/neuroscience-community-labs/gene-vector-and-virus-core) and/or by contacting the lead authors. Additional information regarding ordering and MTA completion can be found at http://www.optogenetics.org/sequence_info.html.

### Reporting summary
Further information on research design is available in the Nature Portfolio Reporting Summary linked to this article.

## Data availability
All source and raw data generated in this study have been deposited in the Zenodo database (https://zenodo.org/record/8185119; https://doi.org/10.5281/zenodo.8185119) with Open-Access under a Creative Commons Attribution 4.0 International license. The dorsal root ganglia RNA-seq data reanalyzed from ref. 67, are deposited to the Sequence Read Archive under Bioproject PRJNA783035. Source data are provided with this paper.

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

## Acknowledgements

This work was funded by NIGMS DP2GM140923, NIDA R00DA043609, NIDA R01DA056599 and NIDA R01DA054374 awarded to G.C. and the NIDA Translational Addiction Research Fellowship Program grant T32DA028874 to G.J.S. T.B. was funded by SNF P2ZHP3 178114. B.A.K. and N.M.M. were funded by NIDA F32DA055458 and NIDA F32DA053099, respectively, while J.A.W. and L.M.W. were funded by NINDS F31NS125927 and NIDA F31DA057795, respectively. W.R.R. acknowledges support from The Burroughs Wellcome Fund, NINDS R01NS119476 and NIDA DP1DA054343. C.J.W. acknowledges support from NINDS R35NS105076. M.R.H. and B.C.D.J. acknowledge support from NIDDK R01DK130239. J.A.B. acknowledges support from NIDA R01DA054374 and R01DA056599. M.L.P. and S.T. acknowledge support from NIMH R37MH109728. K.D. acknowledges support from The Howard Hughes Medical Institute. We thank the NIH Common Fund (Regenerative Medicine Program) and the NIH Helping to End Addiction Long-Term (HEAL) Initiative. We thank the University Laboratory Animal Resources (ULAR) group at the University of Pennsylvania for assistance with rodent, small mammal and rhesus macaque husbandry and veterinary support, including all faculty stationed at both the Translational Research and Smilow Research Buildings. We the thank University of Pennsylvania School of Veterinary Medicine faculty for assistance with anesthesia, MRI imaging and pathology work in relation to our macaque studies. We thank other members of the Corder Lab, especially Malaika Mahmood, for additional technical assistance and support. We thank Carleigh O'Brien of the Bennett Lab for assistance with glial cell-type analysis, and the Fuccillo lab and Rader lab for providing additional technical assistance. All viral construct design and production was performed by the "Cracking the Neural Code" Program and Gene Vector and Virus Core at Stanford University.

## Author contributions

G.J.S., G.C., C.R. and K.D. conceptualized and planned out the study. G.J.S., S.T., J.L., B.A.K., S.R., N.M.M., T.B., K.L.G., W.R.R., C.R., and G.C. designed the research. C.R., K.D., G.C., and G.J.S. conceptualized and designed all viral constructs, and C.R. performed molecular biology and viral production. S.T. and K.L.G. planned and performed the non-human primate surgeries, and G.J.S. performed immunohistochemistry, in situ hybridization and analysis. T.B. performed rat and shrew surgeries and tissue processing; G.J.S. performed the immunohistochemistry and in situ hybridization with analysis assistance from J.A.W. G.J.S., B.A.K., and N.M.M. performed mouse surgeries, tissue processing, immunohistochemistry, and analyses for in vivo characterization studies, with analysis assistance from J.A.W. G.C. and S.R. performed behavioral chemogenetic experiments. G.J.S. performed in vivo fiber photometry experiments, with behavior analysis assistance from L.M.W. J.A.B. provided the *Oprm1*[2A-Cre] mouse line. W.R.R. and J.L. planned and performed peripheral nervous system studies and human iPSC transduction studies, with I.S. providing transcriptomic data. S.J., I.S. and C.J.W. provided human iPSCs. K.T.C. and A.R. planned and performed in vitro neuronal and glial studies. J.A.B., M.R.H., B.C.D.J., M.L.P., F.C.B., M.L.B., K.T.C., and K.D. provided additional laboratory supervision and support funding. G.J.S. and G.C. wrote the manuscript, and all authors contributed to the editing and revising of the document.

## Competing interests

G.C, K.D., C.R. and G.J.S. are listed as inventors on a provisional patent application filed on November 11th, 2022 through both the University of Pennsylvania and Stanford University regarding the custom sequences used to develop, and the applications of, both the *mMORp*1 and *hMORp*1 constructs (patent application number: 63/383,462 462 'Human and Murine *Oprm1* Promotes and Uses Thereof'). The remaining authors declare no competing interests.

## Additional information

[1]Dept. of Psychiatry, Perelman School of Medicine, University of Pennsylvania, Philadelphia, PA, USA. [2]Dept. of Neuroscience, Mahoney Institute for Neurosciences, Perelman School of Medicine, University of Pennsylvania, Philadelphia, PA, USA. [3]Dept. of Neurology, Brigham and Women's Hospital and Harvard Medical School, Boston, MA, USA. [4]Translational Medicine and Human Genetics, Perelman School of Medicine, University of Pennsylvania, Philadelphia, PA, USA. [5]Dept. of Biobehavioral Health Sciences, School of Nursing, University of Pennsylvania, Philadelphia, PA, USA. [6]Dept. of Pathobiology, School of Veterinary Medicine, University of Pennsylvania, Philadelphia, PA, USA. [7]F.M. Kirby Neurobiology Center, Boston Children's Hospital and Harvard Medical School, Boston, MA, USA. [8]Stem Cell Translation Laboratory, National Center for Advancing Translational Sciences, National Institutes of Health, Rockville, MD, USA. [9]Division of Neurology, Dept. of Pediatrics, Children's Hospital of Philadelphia, Philadelphia, PA, USA. [10]Dept. of Systems Pharmacology & Translational Therapeutics, Perelman School of Medicine, University of Pennsylvania, Philadelphia, PA, USA. [11]CNC Program, Stanford University, Stanford, CA, USA. [12]Dept. of Bioengineering, Stanford University, Stanford, CA, USA. [13]Howard Hughes Medical Institute, Stanford University, Stanford, CA, USA. [14]Dept. of Psychiatry & Behavioral Sciences, Stanford University, Stanford, CA, USA. ✉e-mail: deissero@stanford.edu; gcorder@upenn.edu

