## [Peer Review File · Nature Communications]

REVIEWER COMMENTS

Reviewer #1 (Remarks to the Author):

Manuscript by Salimando et al reports the generation of viral vectors that target mu opioid receptor expressing neurons in the central nervous system of mouse and macaque as well as human IPS cells. Cell type-specific targeting has been challenging but new bioinformatic and screening technologies are facilitating the identification of promoter and enhancer sequences that can confer this property.

The authors examine the distribution of their mouse promoter-designed viral vector within different areas of the mouse CNS as well as select areas of the rat and shrew CNS. They also examined the distribution of the human promoter designed viral vector within different areas of the macaque CNS. Lastly, they also test the expression of the mouse promoter-designed viral vector in hiPSCs. The authors used immunohistochemistry with an antibody to the mu opioid receptor shown to be specific by staining a mu opioid receptor KO mouse. They also used RNAScope and two genetically modified mouse lines that express Cre under control of the mu opioid receptor genomic locus.

The paper clearly demonstrates the creation of useful new tools for the study of mu opioid cells and circuitry across species as well as for potential therapeutic applications. Experiments examining the biology of cell, circuits and behavior were strictly for validation of these tools. Inclusion of species such as shrew and macaque is commendable and the creation of multi-species viral vectors for the research community is valuable. A few comments:

1. In general, it is difficult to see the reported overlap many of the images. Not that it doesn't occur but just that it isn't obvious from the pictures. It is important that the authors improve these to demonstrate what appears to be a rigorous analysis. For example, it is impossible to see that nearly 80% of the eYFP are Mor+ cells as reported from the images shown.
2. The authors should include quantification for each area of the nervous system where the viral vector was expressed, and these data should be included in the main figures. It is so crucial to the usefulness of the tool for the reader to see these data.
3. Fiber photometry makes it hard to know if the effect of the less than complete inhibition by morphine in Figure 4f is due to dose effect (not maximal for each cell) or off target effect (some cells are not inhibited). Authors should discuss.

4. A more minor point, I may be mistaken, but the work in Ref 22 (earlier study by the senior author) shows the primary afferents (not the spinal cord neurons) are responsible for the effect of spinally delivered agonist.

Reviewer #2 (Remarks to the Author):

In the manuscript titled, "Human and murine promoter driven viral constructs for genetic access to opioidergic cell-types", Salimando et al. have reported development of new tools to investigate mu-opioid receptor (MOR)+ neural cell-types across species. Through novel promoters mMORp1 (mouse) or hMORp1 (human), and interplay with cre recombinase switches and transgenic lines, the authors have performed extensive studies to demonstrate the strategies that promote AAV based interventions of MOR+ cells. Different strategies have been characterized in greater depth to demonstrate the ability to target and/or functionally manipulate MOR+ cells in mouse, rats, shews, NHPs and iPSCs.

I recommend this work for publication in N.Comm for the discovery of novel promoters which can be transformative for this field of research, and for the extensive body of work that is presented in this manuscript across species, provided the authors can address the following major concerns.

Major concerns:

1. Lack of clarity on the m/hMORp1 promoter specificity to neurons vs glia. Lines 129-144 and Fig. 1
 - a. The authors show expression of reporter in NeuN+ and Iba1+ cells based on the report of expression in literature. However, it is unclear if the promoter can express in other widely AAV transducible glia cell-types such as GFAP/S100B astrocytes or oligodendrocytes.
 - i. A quantitative data across these cell types using AAV1 (direct delivery) and/or PHP.eB (ICV or IV) is recommended.
 - ii. If the MOR+ cell specificity results only through interplay with transgenics or recombinase systems, capsid serotype and delivery, this is critical to highlight in the main text, sub-section titles etc and revise language to avoid any ambiguity among readers.

b. AAVs are generally not potent towards transducing microglia, and it is unclear whether the lack of microglia transduction is due to the capsid or the promoter in this context, and the repeated conclusions drawn by the authors to define the m/hMORp promoter to be NeuN specific due to its lack of expression in Iba1+ is ambiguous to the reader, may be misleading for non-AAV background readers as even with ubiquitous promoters, AAVs have not shown promise to transduce microglia significantly in vivo.

i. The authors need to further clarify with supporting experiments to test the transduction of microglia with the m/hMORp promoter using a delivery system that has been previously demonstrated to transduce microglia in vivo significantly. eg. Lin et al, Nat Methods, 2022 PMID: 35879607.

ii. Optional - Microglia transduction in vitro using existing AAV serotypes may be helpful but only as an additional supporting data in addition to in vivo as the field has conflicting views on the translatability of in vitro systems in vivo.

iii. If the authors choose to not go down the route of validating microglia transduction, it is critical to re-draw the conclusion across figs/text and move the microglia related data to the supplementary, clarifying the ambiguity to the readers.

2. How does m/hMORp1 promoter perform compared to the widely used pan-neuronal hSyn promoter?

a. Based on the information provided in the manuscript, it is unclear to the reader as to why they should choose the new promoter over the commonly used hSyn. The data in Fig. 1e. shows overlap of pan-neuronal marker NeuN which is used across any other neuronal promoter. hSyn promoter is less than 500 bp compared to ~1500 long m/hMORp promoter. For AAV researchers, this is a huge size difference, and may not be welcomed unless the authors can provide quantitative supporting data that shows the differences in either cellular specificity or efficiency in the relevant rodent models.

Minor concerns:

1. Since N.Comm is for broader audience, I would highly recommend the authors to revise to generalized nomenclature across main text/figs/legends. Below are a couple of examples and would encourage to apply this broadly across the manuscript.

a. Lines 114-117. Recommend using MOI (multiplicity of infection) nomenclature for AAV transduction assays across in vitro cell culture systems than using concentration as this is less meaningful without understanding the cell count per well.

b. Fig.1c. A total dose per animal or a concentration followed by volume (as shown in Fig. 1d). Consistency is helpful for readers.

2. Except for PHP.eB and PHP.S, I didn't really come across a rationale for a broader audience to understand the use of the different AAV serotypes (AAV1, 5, 8, 9) across studies. Will be helpful for a call out in the text or in methods.

3. In methods, line 192 – AAV9 is missing although I did see this capsid was being used in the experiments.

4. While the manuscript provides a detailed characterization of different systems to get to the desired cell-type targeting across species, there are several layers of control knob (capsid, promoter, genetic switch, transgenics, species, delivery) that a reader needs to understand. While the authors have tried to illustrate in figs as needed, it would be very helpful if there is any room to improve or add a supplementary summary figure.

Reviewer #3 (Remarks to the Author):

Salimando et al. reported a series of mouse and human MOR promoter-based viral tools that allow the efficient and specific expression of transgenes across research models. Mouse mMORp could recapture MOR expression pattern to a similar level to available Cre lines across many brain regions. The authors then demonstrated a few use cases for mMORp: hM4Di for inhibition; GCaMP6f for activity monitoring; recombinase (Cre and Flp) for intersectional genetics; and PhPeB/S for broad targeting of central and peripheral MOR+ neurons. Lastly, the authors demonstrate that hMORp can transfect macaque and human iPSC cells. These new constructs could provide more flexibility to the anatomical and functional investigation of opioid circuits and facilitate opioid research in other species. The results are interesting, and the paper is well-written. I have a few questions regarding data presentation and interpretation:

1. Line 135: The titer used in the paper varies a lot. The fact that the optimized titer, $3e+11$, is several folds lower than the commonly used titer ($1e+12$) could mean that the latter might introduce nonspecific expression. Please further explain how the optimized titer is determined.

2. Line 164 and 170: Fig. S4, Sun1-eGFP is observed in striatal matrices. Fig. 1h, I see lots of Sun1-sfGFP+ neurons that are mMORp1-mCherry negative. These data mean that the Oprm1-Cre-2A mouse line is not specific. What is the authors' take on this?

3. Line 180: Fig. 1n and Fig. S5, in addition to specificity/selectivity, it would also be informative to know the sensitivity, i.e., regarding Cre line as "ground truth," how much of the total targeted population can be successfully captured by the mMORp construct (basically means comparing MOR+ and MOR- % in all Cre+ cells).

4. Line 186: is the mMORp sequence also present in rats and shrews? Also related – is the hMORp sequence present in macaques?

5. Line 189: Please comment on, or summarize the various serotypes used across the manuscript and whether the authors prefer a particular one.

6. Line 233: Fig. 3e and S6b: The authors claimed that DCZ and CNO produced similar analgesic effects, but they look very different to me. It almost seems like CNO only influenced the affective responses to alleviate pain (licking, jumping) while DCZ only affected thermal sensitivity (quick onset paw withdrawal). However, CNO and DCZ should act on the same target, so these figures are hard to reconcile.
7. Line 285: Fig. 4g, the x-axis should be minutes?
8. Line 318: Fig. 5c-d: Con/Fon virus has been reported by many labs to be leaky. Do the authors have control experiments showing minimal baseline expression of Con/Fon virus in SST-Cre mice without FlpO?
9. Line 326: Fig.5e: if I understand correctly, the white color denotes the regions co-expressing mMORp-eYFP and FLEX-tdT. I couldn't see lots of white colors but instead saw blue and red colors preferentially located in different areas. The data do not support the authors' claim that eYFP and tdTomato are co-expressed in anti-Cre+ cells. If this discrepancy is due to the validity of the Cre line, then the authors should do a MOR staining and show that it correlates well with mMORp-eYFP. Please explain and provide the quantification and zoom-in images in supplementary figures.
10. Line 330: Fig.5g, the authors need to provide higher-quality images. There are clouds of NeuN mRNAs in DAPI-negative areas, which could mean that the mRNAs weren't well fixed and couldn't be used for quantification.
11. Line 387: strain number should be 035574.
12. Since this paper reports a new vector tool, it would be great if the authors could share this with the community by placing the construct on addgene with instructions for at least mouse models.

Reviewer #4 (Remarks to the Author):

In this manuscript the authors generate human and mouse MOR promoter viral constructs. This is important as MOR expressing neurons are critical in the neural circuits underlying analgesia and addiction. The generation of validated viral tools to selectively transduce and modulate these neurons is therefore a significant advance and could impact on numerous aspects of neuroscience. They have developed what looks to be an effective promoter system to drive expression in MOR cells with good selectivity. I thought that overall experiments were well performed and reported and I was impressed by the degree of validation which did not just look at expression but also functional assays (for instance calcium imaging and assessing the impact of chemogenetics on behaviour). This validation is helpful in establishing the utility of these constructs. I have a few issues that I would like to see addressed (below) that could strengthen it further.

Major

Expression in rat and shrew. In Fig 2 expression of eYFP following administration of AAV1-mMORp-eYFP in other model organisms (rat and shrew) in candidate brain regions where MOR is known to be expressed. The key issue is fidelity to endogenous MOR expression and so this section would have been strengthened by some co-expression data between the eYFP and endogenous MOR (eg. using in situ as they effectively did in the mouse).

Relationship between behavioural outcomes and transduction : In Fig 3 there is quite a lot of inter-individual variation in the behavioural assays in response to CNO eg just looking at E some animals show large shifts in mechanical thresholds, others less so and one animal no change. Given that the authors show that they can assess transduction with mMORp-hM4Di-mCherry was there a relationship between transduction efficacy and behavioural response?

Expression in human iPSCd nociceptor like sensory neurons. In Fig 7 the authors show that mMORp viral transduction leads to expression of YFP in human iPSCd nociceptor like sensory neurons and not cardiomyocytes. This should be bench marked relative to endogenous MOR expression (for instance using immunocytochemistry or in situ). Although assumptions can be made that iPSCd nociceptor like neurons express MOR that should be shown (because iPSCs do not always fully capture the expression profile in adult human neurons), furthermore there may well be heterogeneity within this neuronal population and showing a relationship between mMORp driven expression and endogenous MOR would be even more convincing.

Minor-

Generally the methods are well written and comprehensive however there is a lack of detail on the iPSC work. For instance, we know nothing about the donors. How many donors were used, ? healthy, ? sex. Was this the result of just one differentiation or were multiple differentiations performed? Were there checks on neuronal viability following viral transduction (in our hands PHPs evokes some toxicity in iPSCd neurons)?

Reference below is lacking detail:

'Bohic, M. et al. Developmentally determined intersectional genetic strategies to dissect adult somatosensory circuit function Authors. (2022).'

Point by Point Response to Reviewers

“Human OPRM1 and murine Oprm1 promoter driven viral constructs for genetic access to μ -opioidergic cell-types” (NCOMMS-22-39191A)

We would like to thank all the reviewers for the interest in our work, and for providing us with insightful and useful comments. We are very grateful to the reviewers for their enthusiasm for the manuscript, and their overall support for its publication upon the addition of several new datasets, further clarification of our results, and greater discussion on the utility and application of our viral tools across multiple biomedical research disciplines.

- Reviewer 1 noted that our manuscript, “*clearly demonstrates the creation of useful new tools for the study of mu opioid cells and circuitry across species, as well as for potential therapeutic applications.*”
- Reviewer 2 commented, “*I recommend this work for publication in Nature Communications for the discovery of novel promoters which can be transformative for this field of research, and for the extensive body of work that is presented in this manuscript across species...*”
- Reviewer 3 stated that, “*These new constructs could provide more flexibility to the anatomical and functional investigation of opioid circuits and facilitate opioid research in other species. The results are interesting and the paper is well written.*”
- Reviewer 4 further added that our development of our MOR specific promoter constructs represents, “*a significant advance and could impact on numerous aspects of neuroscience.*”

Each of the reviewers did raise a few critiques and comments, and in the majority of cases, we addressed and took their suggestions directly. Where we did not, we explain why in a point-by-point fashion as outlined in the document accompanying this letter. Therein, the reviewers’ comments are paraphrased in **italic black font** with our responses in **blue font**, and any new text that has been added to the manuscript in **green font**. The key suggested experiments have now been completed, and are presented in the revised manuscript, and within our point-by-point response document as well for ease of viewing.

Here, we would like to summarize the key new experiments that we have performed for this revision, which we hope you and the reviewers will agree are substantial and help to address the initial questions and critiques regarding the data presented in the original manuscript.

Summary of New Experiments:

1. Addition of IHC data and images showing the successful transduction and expression of another fluorescent reporter variant of our constructs that encodes oScarlet under the *mMORp1* promoter within two mu-opioid receptor (MOR) positive regions of interest (central amygdala and ventral tegmental area) after intracranial viral injection.
2. Glial marker IHC panels to verify *mMORp1* promoter expression in neurons vs. microglia vs. oligodendrocytes vs. oligodendrocyte precursors vs. astrocytes. These studies were performed across all species where staining was applicable/successful (mouse, rat, shrew and macaque), and the quantification/analysis of *mMORp1*-eYFP signal and glial antibody staining overlap is shown across multiple regions of interest: central amygdala, prefrontal cortex, and ventral tegmental area.
3. Updated retro-orbital injections and resulting IHC image sets from C57BL/6J injected with the PHP.eB-*mMORp1*-eYFP construct to better demonstrate the transduction efficacy and spread of this virus throughout the CNS with this approach.
4. Quantitative PCR and IHC analyses for the transduction efficacy of our *mMORp1* and *hMORp1* viruses in cultured human SHSY5Y neuronal, human C20 microglial, human A172 astrocytic and mouse N9 microglial cell lines. Resulting representative images and reported average fold change in expression of the reporter fluorophore, as well as native *Oprm1* expression across these cells line, is provided.
5. IHC imaging and analysis for the overlap of cells transduced by our *mMORp1* promoter driven virus (AAV5-*mMORp1*-eYFP) and a more generic, human synapsin (*hSyn*) promoter driven virus (AAV5-*hSyn*-mCherry) when the two viruses were co-injected together into the same region of interest.
6. NCBI BLAST analyses of the overall homology and coverage of our *mMORp1* and *hMORp1* promoter sequences with that of the native *Oprm1* promoter sequence across all species of interest discussed in our paper, where valid genomic data is available to conduct such analyses (*i.e.*, mouse, rat, macaque and human).
7. IHC analyses and representative images from C57BL/6J mice co-injected intracranially with a mix of our *mMORp1*-FlpO encoding virus (AAV1-*mMORp*-FlpO) and a Flp-dependent virus expressing mCherry (AAV9-*Ef1a*-fDIO-mCherry) to demonstrate another application of our viruses/constructs for performing select cell population labeling.
8. Updated RNAscope (FISH) images and analyses of the expression and quantification of our *mMORp1* viral constructs in cultured mouse DRG neurons.

9. RNAscope (FISH) for the co-labeling of cells transduced with our *mMORp1*-eYFP encoding virus (*EYFP* transcript signal) and native mu-opioid receptor (*Oprm1*) transcript in rat neural tissue following intracranial viral injection.
10. Quantitative PCR analyses performed on cardiomyocytes and nociceptors differentiated from cultured human iPSC cells to demonstrate the overall gene expression levels of *OPRM1* in these two cultured cell types.

A. Response to Reviewer #1:

1. ***“In general, it is difficult to see the reported overlap in many of the images...It is important that the authors demonstrate what appears to be a rigorous analysis.”***

We would like to thank the reviewer for pointing this out to us, particularly regarding the quality of the representative images provided in Figure 1 that are meant to demonstrate the overlap of *mMORp1*-eYFP and either anti-Cre or *Oprm1* transcript signal that we attempted to show for our initial construct specificity/selectivity analyses. In order to rectify this and make it easier for readers to notice this overlap more immediately when looking at this figure, we have produced **updated versions of Fig. 1 f-m** that make use of both a new color schema and include the addition of bolded, white arrowheads that point to the majority of the cells within each image where either *EYFP* and *Oprm1* transcript signal (**Fig. 1g**) or *mMORp1*-hM4Di-mCherry and anti-Cre or *mMORp1*-hM4Di-mCherry and Sun1-sfGFP (**Fig. 1h-m**) are overlapping on the same cell body.

Panels 1g-m as they are shown in the updated Figure 1. White arrowheads denote cells in which signal from either *Oprm1* and *EYFP* transcript (g) or *mMORp1*-mCherry and either Sun1-sfGFP (h) or anti-Cre (i-m) can be seen to overlap on the same cell body.

In addition to this, we have also provided an updated supplementary figure (**Supp. Fig. 4**) that includes an example of the quantification that we perform on the high magnification images we collected from each region of interest transduced by our AAV1-*mMORp1*-hM4Di-mCherry across the two Cre lines used. **Supp. Fig. 4i** shows an ROI taken from the central amygdala of one of the *Oprm1*^{2A-Cre} mice injected with the AAV1-*mMORp1*-hM4Di-mCherry virus and then stained with an anti-Cre Ab for quantification of the overlap of both signals following IHC. The respective channels for each stain have been presented separately and as a merged image in order to demonstrate the overlap of each signal, and underneath each of these, a complementary image of each signal presented in greyscale is provided along with markers overlaying anti-Cre+ cells (red dots), *mMORp1*-mCherry+ cells (blue dots) or the overlap of both signals on the same cell body (yellow dots). Using these new image sets, in tandem with the other quantification data provide in **Supp. Fig. 4**, we hope that we are better able to provide evidence of the ability of our viral constructs to transduce and more exclusively express transgenes of interest in putative MOR/*Oprm1* promoter (+) cells in a way that can be more immediately observed and interpreted by a potential reader of the manuscript.

New panel S4i from Supplementary Figure 4. Upper panels demonstrate anti-Cre (blue) staining (left), *mMORp1*-mCherry (yellow) fluorescent signal (middle), and an overlay of the two signal (right). Lower panels show greyscale images of the same anti-Cre (left) and *mMORp1*-mCherry (middle) signal, with markers placed over the cell bodies positive for anti-Cre staining (red dots) or *mMORp*-mCherry signal (blue dots). The right most lower panel displays a merge of the two marker types to demonstrate the relative overlap of both anti-Cre and *mMORp*-mCherry across cells in this image (yellow dots), as well as individual cells that were positive only for anti-Cre or *mMORp*-mCherry signals alone. Scale bar = 100um.

2. ***“The authors should include quantification for each area of the nervous system where the viral vector was expressed, and these data should be included in the main figures.”***

The reviewer is completely correct in pointing this out, as appropriate quantification and analysis of both efficacy and specificity of these new constructs/viral tools are tantamount to demonstrating their utility to the greater neuroscience and biomedical research communities. In our initial submission, we did provide the quantification data for the initial specificity/selectivity studies presented in **Figure 1**, which served to establish that our viruses were indeed selective and specific enough to predominantly transduce and drive transgene expression in putative MOR+ cells (or at least, cells in which the *Oprm1* native promoter sequence was present) in mouse neural tissue. As the brain regions focused on within this initial set of experiments covered the primary regions used for all mouse studies throughout the rest of the manuscript (i.e. mPFC, DMS, CeA and VTA) for further use case studies, we predominantly focused on the quantification of these regions alone with the hopes that the high level of selectivity we saw for our viruses to express in these MOR+ cells (as demonstrated by the high overlap noted with anti-Cre+ cells when compared with anti-Cre- cells across two *Oprm1*-Cre mouse lines, and summarized in **Figure 1n**), as well as the lack of spread/expression of virus/virally encoded transgenes into brain structures known to express little to no MOR (i.e. the BLA, as observable in **Figure 1f**, and further expounded upon in **Supp. Fig. 2k-n**, in which our *mMORp1*-eYFP virus was co-injected along with a generic *hSyn*-mCherry reporter into the CeA, and notably little to no *mMORp* encoded eYFP was observed in the BLA, unlike *hSyn* encoded mCherry), would instill confidence that our viruses do indeed more selectively drive transgene expression in MOR/*Oprm1* promoter+ neurons. As these datasets in question are quite sizeable, and may not fit neatly into a main text figure in such a way that would allow the data to be readily interpretable by a potential reader due to how small the corresponding plots, graphs and accompanying images would need to be scaled to in order to fit, it was our hope that providing these in a Supplementary Figure would allow for the necessary space to display the data in their entirety and in a format that is easier to read and digest. As mentioned above, the quantification data for the specificity/selectivity studies have been provided in **Supp. Fig. 4a-h**, which represents an updated supplementary figure from our previous submission that included this same data. We have also made notes both within the main text of the manuscript in the Results section that discusses the data presented in **Figure 1n** to refer to **Supp. Fig. 4** for a more extensive breakdown of these findings, as well as similar language in the figure legend for **Figure 1** that references **Fig. 1n** to direct readers to **Supp. Fig. 4a-h** for the accompanying quantification data used to create the summary graphs shown. New Supplementary Figures have also been generated and added to the manuscript to accompany the majority of studies in which our viruses were injected into either a new animal model or in which an examination of the specificity or selectivity of our viruses for transducing

specific cell types (i.e. glial v. neuronal cells) was conducted (further discussion of these figures is provided below, but in brief: **Supp. Fig. 2k-n**, **Supp. Fig. 5**, **Supp. Figs. 7-8**, and **Supp. Fig. 13**).

Panels S4a-h from Supp. Fig. 4. Quantification performed within the four major brain regions of interest (central amygdala [CeA], dorsal striatum [DMS], medial prefrontal cortex [mPFC] and ventral tegmental area [VTA]) for both the *Oprm1^{2A-Cre}* and *Oprm1^{Cre-GFP}* transgenic mouse lines used to create the summary graphs in Fig. 1n. S5a-d shows the total *mMORp*-mCherry+/Cre+, *mMORp*-mCherry+/Cre- or *mMORp*-mCherry-/Cre+ cell counts quantified within a predetermined region of interest (ROI) imaged for a given region (bar graphs on the right), and the summary data for the total percentage of cells that were either *mMORp*-mCherry+/Cre+ compared to those that were *mMORp*-mCherry+/Cre-, in order to demonstrate the average number of cells transduced by the *mMORp* virus that were also positive for *Oprm1^{2A-Cre}* staining (parts of a whole graphs on the left). Results shown for the CeA (~92.8%, N=2 mice, n=4 ROIs, a), DMS (~87.3%, N=2, n=4, b), mPFC (~97.6%, N=1, n=1, c) and VTA (~89.7%, N=2, n=3, d). S5e-h show the same types of data, but in this case for the analyses conducted using the *Oprm1^{Cre-GFP}* transgenic mouse line. Results shown for the CeA (~86.4%, N=3, n=5, e), DMS (~90.4%, N=3, n=3, f), mPFC (~86.8%, N=2, n=3, g) and VTA (~81.7%, N=3, n=5, h). Data presented in bar graphs represent the mean of the individual data points shown overlaying each, with error bar representing the standard error of the mean (S.E.M.).

Text added into the Results section of the manuscript making note of the quantification data presented in Supp. Fig. 4 that accompanies the graphs shown in Figure 1n:

“...Quantification for individual mouse lines available in Supp. Fig. 4a-h, with sample quantification of a single region of interest [ROI] shown in Supp. Fig. 4i” (Lines 172-174).

Text added into the figure legend for Figure 1:

“Detailed information regarding the total cell counts and overall quantification for each region within the individual mouse lines can be found in Supp. Fig. 4” (Lines 630-633).

3. ***“Fiber photometry makes it hard to know if the effect of the less than complete inhibition by morphine in Figure 4f is due to dose effect (not maximal for each cell) or off target effect (some cells are not inhibited). Authors should discuss.”***

The reviewer makes an important point here regarding one of the shortcomings of *in vivo* fiber photometry for performing the behavioral and drug injection studies summarized in **Figure 4f**. While the use of our AAV1-*mMORp1*-GCaMP6f virus would allow us to transduce and then record from all putative MOR/*Oprm1* promoter+ cells within the CeA for these studies, it does not allow us to parse out functionally morphine-sensitive MOR+ cells from those that respond more minimally to a dose of morphine, regardless of our use of a relatively high dose of morphine (10mg/kg). This could be explained by the attributes of separate subtypes of cell populations within the CeA that also express MOR/*Oprm1*, such as the *Pkrcd+* or *Sst+* subtypes described in Wang et al. (eLife. 2023, PMID: 36661218), which have classically been shown to display antagonistic effects in regards to their responses to and processing of certain stimuli within the CeA (Haubensak et al. Nature. 2010, PMID: 21068836; Yu et al. J. Neurosci. 2016, PMID: 27307236). This could thus argue for the presence of a level of basal, background signal that could be consistently detected from a population of MOR/*Oprm1*+ cells within the CeA that less significantly affected by morphine dosing (i.e. lower MOR density on cells) or by contrast is receiving competing stimulation/input from an exogenous or endogenous populations of cells that projects onto select MOR/*Oprm1*+ neurons transduced by our virus in response to other stimuli processing that could surround the response of the animal to the acute administration of morphine (i.e. in response to the injection, to stress, etc.). A more effective way to determine this could be by conducting a targeted study of a select subpopulation of CeA MOR/*Oprm1*+ cells that the literature may support as showing greater responsiveness to morphine administration and attempting to express our GCaMP encoding constructs more selectively in these cells. However, we felt that parsing out the potential opioidergic circuitry/microcircuitry of the CeA fell outside of the purvey of this manuscript.

As the goal of the work presented in this manuscript is to both validate and present a potential use case of *mMORp*-GCaMP6f viral tools that may suggest their utility to a wide range of neuroscience, opioid and pain researchers across fields, it is our hope and desire that if readers deem that these applications seem of interest to them, that they will take it upon themselves to expand more into the specifics of opioidergic signaling, circuitry and the responses of MOR/*Oprm1*+ cell populations throughout the brain to different forms of stimulation at the level of their individual research questions.

Regardless though, we have added new text into the **Discussion** section of the revised manuscript to address these clear shortcomings when using *in vivo* fiber photometry to conduct the studies outlined in **Figure 4**, with the hope that this discussion will prime readers interesting in making use of our tools on how to best think about and design experiments that will allow them to effectively synergize their technique of choice with the construct that would be most useful for them in answering their question.

Text added to the Discussion to address the shortcomings of fiber photometry + possible interpretations of our results:

“The potential shortcomings of a technique such as fiber photometry, which examines the bulk fluorescent signal from a population of cells transduced with a given calcium indicator, should be noted though. It is possible that the less than complete inhibition of signal from *MORp*-GCaMP noted in these studies may be attributed to the competing activity patterns of different MOR+ subtypes known to be present within a region such as the CeA in response to select types of stimuli or overall differences in MOR receptor density that may be present on transduced cells (which may alter the responsive of certain cells to drugs like morphine, thus influencing the fluorescent signal noted for the broader population of transduced cells). As such, care should be taken when designing experiments using viral tools such as ours to attempt to both target and restrict expression of certain transgenes to the desired population or subpopulation of cells most germane to a specific research question, an approach that our suite of constructs may help to further facilitate when using them in combinations with other viruses and transgenic animals lines, as demonstrated through additional studies we’ve presented here.” (Lines 474-485).

4. ***“A more minor point, I may be mistaken, but the work in Ref 22 (earlier study by the senior author) shows the primary afferents (not the spinal cord neurons) are responsible for the effect of spinally delivered agonists?”***

The reviewer is correct that **Fig. 2p-r** in the referenced Nature Medicine paper (Corder et al. Nat. Med. 2017, PMID: 28092666) shows loss of presynaptic MOR reduces intrathecal morphine anti-nociception. In the studies outlined in **Figure 3** of the current manuscript, however, we are chemogenetically inhibiting a large number of the spinal neurons (interneurons, projections neurons, ventral horn, etc.) directly with a systemic injection of CNO, which should engages hM4i:Gi/o signaling in spinal-located dendrites, spinal located cell bodies, as well as the long-range axons in the brainstem/midbrain that are far away from the spinal cord. Thus, the current studies conducted using our AAV1-*mMORp1*-hM4Di-mCherry viral constructs are not directly comparable to our previously published findings.

B. Response to Reviewer #2:

1. ***“Lack of clarity on the m/hMORp1 promoter specificity to neurons v. glia. The authors show expression of reporter in NeuN+ and Iba1+ cells based on the report of expression in literature. However, it is unclear if the promoter can express in other widely AAV transducible glia cell-types such as GFAP/S100B astrocytes or oligodendrocytes. A quantitative data across these cell types using AAV1 (direct delivery) or PHP.eB (ICV or IV) is recommended.”***

The reviewer brings up an important point regarding the completeness of our specificity/selectivity studies, in that our examination of cell types that could be transduced by our *MORp* constructs initially only extended to neurons and microglia. While this was originally conducted in an attempt to address any potential discrepancies in the literature regarding which cell types are *Oprm1+* in the mouse brain (Ferrini et al. Nat. Neurosci. 2013, PMID: 23292683), we agree with the reviewer that a more rigorous investigation of whether or not our constructs were capable of transducing additional glial cell types was very much warranted.

To address this, we have included several new supplementary figures that detail both the staining and quantification of astrocytic (GFAP), oligodendritic (CC1 & PDGFR α) and microglial (Iba1) markers across different brain regions for all the animal model species discussed in this manuscript (mouse, rat, shrew and macaque), where applicable. **Supp. Fig. 5** presents a glial marker panel conducted on tissue from mouse mPFC, CeA and VTA for CC1 (**S5a-b**), GFAP (**S5c-d**), PDGFR α (**S5e-f**) and Iba1 (**S5g-h**) staining, respectively, following intracranial injections of a new cohort of C57BL/6J male mice with our AAV1-*mMORp*-eYFP virus. Representative images from one of these regions is shown to demonstrate the general pattern and morphology of cells stained with each glial antibody (following IHC), and summary graphs and plots are provided for each region below these images to show overall quantification of the overlap of *mMORp*-eYFP and glial marker signal on the same cell bodies (as delineated by DAPI nuclei staining). **Supp. Fig. 8** presents the same kind of glial marker panel conducted on rat (**S8a-d**) and shrew (**S8e**) tissue. Analysis and quantification of the overlap of *mMORp*-eYFP and glial marker signal in rat tissue was only conducted for the CC1, GFAP and Iba1 stains, as the PDGFR α antibody we used for these assays did not produce signal of a high enough quality to allow for quantification. Analyses were also only conducted for ROIs from the rat CeA and VTA, keeping in line with the main complementary structures that we examined for transduction selectivity across both mouse and rat and present in **Figure 2**.

For shrew tissue, similar issues with both antibody efficacy and the quality of remaining tissue available to us only permitted an analysis of Iba1 and *mMORp*-eYFP overlap in the AP/NTS region. Lastly, **Supp. Fig. 13** presents the same glial marker panel as conducted on macaque anterior cingulate cortex (ACC) tissue alone. CC1 (**S13a,d**), GFAP (**S13b,e**), PDGFR α (**S13c, f**) and Iba1 (**S13g**) total cell count quantifications and analyses are shown, with representative images provided for CC1, GFAP and PDGFR α staining.

For the rat, shrew and macaque glial marker panels, representative images of Iba1 staining were not provided in the figures, as such representative images can be found in **Fig. 2c+d**, **Fig. 2e** and **Fig. 6d**, respectively. Across all panels for all species and regions of interest assessed, we noted little to no overlap whatsoever of the *mMORp*-eYFP signal (indicating virally transduced cells) with that of signal for CC1, GFAP, PDGFR α or Iba1 antibody staining. These results suggest that our *MORp* constructs selectively transduce neuronal cell types and show little to no ability to transduce most canonical glial cell types within the CNS of mouse, rat, shrew or macaque animal models.

We have included new text within the respective **Results** sections of the main manuscript to summarize these findings as well, and provided additional text within the figure legends of **Fig. 1**, **Fig. 2** and **Fig. 6** to direct readers to these Supplementary Figures in order to view the full glial marker panels if they so desire.

New Supplementary Figure 5. Glial marker panel and analysis within the mPFC, CeA and VTA of neural tissue from AAV1-*mMORp*-eYFP intracranially injected C57BL/6J male mice. Representative images showing CC1 staining in the CeA (S5a), GFAP staining in the mPFC (S5c), PDGFR α staining in mPFC (S5e) and Iba1 staining in mPFC (S5g) show the clear separation of signal for all glial markers from that of signal for virally transduced eYFP. Summary bar and parts of a whole graphs are provided below each respective marker's representative image set, which outline the total quantification of all *mMORp*-eYFP and glial marker positive cells counted across ROIs imaged for each region, and the average percentage of *mMORp*-eYFP/glial marker+ cells displayed (S5b,d,f,h). Results shown for CC1 in the mPFC (1.9%, N=2 mice, n=3 ROIs), CeA (1.4%, N=2, n=3) and VTA (4.3%, N=1, n=2), GFAP in mPFC (1.2%, N=2, n=3), CeA (1.6%, N=2, n=3), and VTA (0.9%, N=1, n=2), PDGFR α in the mPFC (1.0%, N=1, n=2), CeA (5.1%, N=2, n=3) and VTA (3.5%, N=1, n=2) and Iba1 in the mPFC (1.3%, N=1, n=2), CeA (1.6%, N=1, n=2) and VTA (1.2% N=2, n=3) indicate little to no overlap of virally transduced eYFP with any of the four glial markers across all three regions. All data in bar graphs are presented as means, with individual data points overlaying them, and error bars represent s.e.m. Scale bars are approximately 50 μ m.

Glial Marker Panel Analysis: Rat & Shrew

Rat — Sprague-Dawley — AAV1-*mMORp*-eYFP (titer: 6.9×10^{11} gc/mL)

New Supplementary Figure 8. Glial marker panel and analysis within the CeA and VTA of neural tissue from AAV1-*mMORp*-eYFP intracranially injected Sprague-Dawley male rats, and the AP/NTS of Asian House Shrews injected intracranially with the same virus. Representative images showing staining in the rat CeA for CC1 (**S8a, upper**) and GFAP (**S8a, lower**), as well as the rat VTA for CC1 (**S8b, upper**) and GFAP (**S8b, lower**) demonstrate the clear separation of signal for these glial markers from that of signal for virally transduced eYFP. Summary bar and parts of a whole graphs are provided below each respective region's representative image set, which outline the total quantification of all *mMORp*-eYFP and glial marker positive cells counted across ROIs imaged for each region, and the average percentage of *mMORp*-eYFP/glial marker+ cells displayed (**S8b,d**). Results shown for CC1 in the CeA (1.0%, N=1 rat, n=2 ROIs) and VTA (0.5%, N=1, n=2), GFAP in CeA (0.6%, N=1, n=2) and VTA (1.5%, N=1, n=2) and Iba1 in the CeA (0.9%, N=1, n=2) and VTA (2.3% N=1, n=2) indicate little to no overlap of virally transduced eYFP with any of these glial markers in these two regions. Summary bar and parts of a whole graphs are also provided for the total quantification of *mMORp*-eYFP and Iba1 positive cells counted in ROIs for the AP/NTS only (**S8e**), with results showing a similarly low level of overlap of signal for either virally transduced eYFP or Iba1 staining in this region (0.6%, N=1 shrew, n=3 ROIs). All data in bar graphs are presented as means, with individual data points overlaying them, and error bars represent s.e.m. Scale bars are approximately 50µm.

New Supplementary Figure 13. Glial marker panel and analysis within the anterior cingulate cortex (ACC) of neural tissue from a single male rhesus macaque intracranially injected with the AAV1-*hMORp*-eYFP virus. Representative images showing staining in the ACC for CC1 (S13a), GFAP (S13b) and PDGFR α (S13c) demonstrate the clear separation of signal for these glial markers from that of signal for virally transduced eYFP. Summary bar and parts of a whole graphs are provided below representative image sets, which outline the total quantification of all *mMORp*-eYFP and glial marker positive cells counted across ROIs, and the average percentage of *mMORp*-eYFP/glial marker+ cells is displayed. Results shown for CC1 (1.5%, N=1 macaque, n=2 ROIs, S13d), GFAP (1.2%, N=1, n=2), PDGFR α (1.5%, N=1, n=2) and Iba1 (0.5%, N=1, n=2) indicate little to no overlap of virally transduced eYFP with any of these glial markers. All data in bar graphs are presented as means, with individual data points overlaying them, and error bars represent s.e.m. Scale bars are approximately 50 μ m

New text added to the Results section of the manuscript to address the addition of the above glial marker panel analysis in mouse tissue:

“As a final test of fidelity and specificity, we wanted to determine if the design of our *MORp* constructs would allow for transduction to occur predominantly in neurons, as opposed to more broadly across neuronal and glial cell types. AAV1-*mMORp*-eYFP was injected in a cohort of mice into mPFC, CeA and VTA, and a glial marker IHC panel was conducted to examine the overlap of *mMORp*-eYFP signal with CC1 (oligodendrocytes), PDGFR α (oligodendrocyte precursors), GFAP (astrocytes) and Iba1 (microglia) antibody staining (Supp. Fig. 5). In all cases, minimal to no signal for glial marker staining was noted on cells positive for *mMORp*-eYFP across all regions of interest (CC1: mPFC=1.9%, n=4 ROIs from N=2 mice; CeA=1.4%, n=4, N=2; VTA=4.3%, n=2, N=1; Supp. Fig. 5a-b; GFAP: mPFC=1.2%, n=3, N=2; CeA=1.6%, n=3, N=2; VTA=0.9%, n=2, N=1; Supp. Fig. 5c-d; PDGFR α : mPFC=1.0%, n=2, N=1; CeA=5.1%, n=3, N=2; VTA=3.5%, n=2, N=1; Supp. Fig. 5e-f; Iba1: mPFC=1.3%, n=2, N=1; CeA=1.6%, n=2, N=1; VTA=1.2%, n=3, N=2; Supp. Fig. 5g-h), indicating a higher transduction preference of our viral constructs for neurons.” (Lines 191-201).

Addressing the same addition of a glial marker panel for the analysis of rat and shrew tissue:

“To confirm that this expression profile was selective for neurons within rat tissue as well, a similar glial marker IHC panel

was conducted using CC1, GFAP and Iba1 antibody staining. As in mice, cells within rats transduced by mMORp-eYFP showed little to no overlap with signal for glial marker antibody staining within CeA (CC1: 1.0%, n=2, N=1; GFAP: 0.6%, n=2, N=1; Iba1: 0.9%, n=2, N=1; Supp. Fig. 8a-b) or VTA (CC1: 0.5%, n=2, N=1; GFAP: 1.5%, n=2, N=1; Iba1: 2.3%, n=2, N=1; Supp. Fig 8c-d). Within shrew tissue, similar patterns of transduction were also found, with mMORp-eYFP restricted exclusively to non-Iba1+ cells (0.6%, n=3, N=2; Supp. Fig. 8e), and overall spread of the virus noted to be relatively restricted to the borders of the AP/NTS structure where the majority of MOR+ cells reside (N = 2 female shrews, Fig. 2e)” (Lines 236-244).

And lastly addressing the addition of this panel for analysis of macaque tissue:

“To confirm *hMORp* selectivity in MOR+ neuron, we performed both a similar glial marker IHC panel and FISH on dACC sections and quantified the co-localization of transgene eYFP signal with that of glial marker staining, or *EYFP* mRNA transcripts with macaque *OPRM1* mRNA transcripts across different regions of interest within the dACC. Glial marker staining with antibodies against CC1, GFAP, PDGFR α and Iba1 in macaque dACC transduced tissue (Supp. Fig. 13a-c) revealed similar results to those observed in mouse, rat and shrew tissue assessed for *MORp*-eYFP signal, in that few to no cells were noted to co-label for eYFP and any glial marker signal (CC1: 1.6%, n=2 ROIs, N=1 macaque; GFAP; 1.2%, n=2, N=1; PDGFR α : 1.5%, n=2, N=1; Iba1: 0.5%, n=4, N=1; Supp. Fig. 13d-g)” (Lines 396-404).

Text added to the legend for Figure 1:

“Amplification of the eYFP signal, along with staining for both neuronal (NeuN) and microglial (Iba1) markers demonstrate selective transduction of neurons, with staining for additional glial markers to further verify this shown in Supp. Fig. 5” (Lines 581-583).

Text added to legend for Figure 2:

“Staining for additional glial markers to demonstrate transduction of predominantly neurons in rat and shrew tissue is shown in Supp. Fig. 7 (including quantification for Iba1 and mMORp-eYFP staining demonstrated in representative images above)” (Lines 616-618).

Text added to legend for Figure 6:

“Staining for additional glial markers and relevant quantification to demonstrate transduction of predominantly neurons in macaque tissue is shown in Supp. Fig. 13” (Lines 723-724).

2. **“If MOR+ cell specificity results only through interplay with transgenics or recombinase systems, capsid serotype and delivery, it is critical to highlight in the main text, sub-section titles etc, and revise language to avoid any ambiguity among readers.”**

We would like to thank the reviewer for pointing out this lack of clarification to us, and reiterate that the MOR+ cell expression specificity is the result of the novel *MORp* promoter, while initial transfection spread/penetrance/etc. is due to permutations of other variables (e.g. titer, capsids, etc.). While the vast majority of the data presented in this manuscript focusing on the transduction/expression of our *m/hMORp* viral constructs *in vivo* is the result of direct intracranial (or retro-orbital) injections, and required no recombinase activity, special capsid composition or a specific serotype to be used to achieve these results, there are indeed a few experiments we outline in the **Results** section that did make use of variants of our constructs that are Cre or Flp recombinase dependent, made use of more novel AAV capsid proteins (PHP.eB and PHP.s) and which made use of a route of administration other than intracranial injection in order to deliver a select virus. Specifically, our chemogenetics-based experiments using our AAV1-*mMORp*-hM4Di-mCherry virus in the spinal cord (summarized in **Figure 3**) and our experiments used to validate our AAV.PHP.eB-*mMORp*-eYFP virus (summarized in **Figure 5**) made use of both intrathecal and retro-orbital injections, respectively, in order to deliver these two viruses for these specific studies. We have revised the text both in the **Results** section that correspond to these studies, as well as in the figure legends of both **Figure 3** and **Figure 5** to make this distinction more clear, and more fully address why we utilized a specific route of administration for a specific virus.

For studies utilizing the recombinase dependent versions of our constructs (results summarized in **Figure 5a-d**), we have included similarly updated text in the corresponding area of the **Results** section and the associated figure legend in order to more clearly point out the need of a recombinase to be expressed exogenously in the same tissue region in order to drive the transduction of the viral constructs used for these experiments. More expanded explanations regarding the use and function of the novel AAV capsid protein variants, and the versions of our constructs that make use of them (AAV.PHP.eB-*mMORp*-eYFP and AAV.PHP.s-*mMORp*-eYFP) have also been included in the **Results** section and figure

legends for **Figure 5** (for experimental results shown in **Fig. 5e+f**) to make the utility of these variants more clear to a general audience.

Lastly, in regard to the use of specific serotypes, we understand the question the reviewer may be getting at is one of which serotype may be best suited to transduce specific cell types or cells within a specific region of the brain, which is a question about viral tropism at its core. While these concerns are of course important, we did not feel it was possible for us (or within the scope of the manuscript) to test every possible combination of AAV serotypes across all the brain regions we discussed in this study, and/or most brain regions of interest to the field in general for our work here. It is our hope though that once we are able to deposit these constructs and make them widely available, that potential future users will be able to test any of the serotypes we have produced and deposited in their regions and structures of interest to more fully help us tabulate where specific serotypes work best (a need that would benefit the greater neuroscience community as a whole regarding serotypes in general), and help direct us to produce package our constructs into different viral serotypes that could have broader applications. For now though, we have provided information in all relevant **Figures**, and sections of the **Results** and **Methods** outlining the specific serotypes we used within a specific region, and representative images to show the successful transduction of those regions with a virus of a specific serotype that reader can hopefully work off of to inform their own endeavors if they so choose to work with our new viral tools.

Text updates regarding the use of direct spinal cord injections of our MORp viruses for chemogenetic studies:

“Thus, for chemogenetic manipulation of spinal MOR+ neurons in behaving mice experiencing nociceptive stimuli, we directly injected C57BL/6J mice with AAV1-mMORp-h4MDi-mCherry at the L4-L5 lumbar sections of the spinal cord (intraparenchymal, 400 nL, $\sim 3 \times 10^{12}$ gc/mL, per injection site; Fig. 3a)” (Lines 260-262).

Addressing the utility and application of the capsid variant AAVs:

“Determining which cell-types are accessed can be partly controlled by the capsid proteins of a specific virus (see Supp. Fig. 16), such as the recently engineered PHP.eB and PNS PHP.S capsids variants, which are capable of selectively transducing cells within the CNS or the PNS over cells present in non-neural tissue types, respectively, as well as the inclusion of specific promoter and/or enhancer element into constructs packaged within AAVs. Furthermore, hundreds of existing Cre and Flp recombinase transgenic mouse lines are also in use in labs around the world which can be used to achieve cell and circuit specific genetic access” (Lines 335-341).

Regarding the use of Cre-dependent viral constructs:

“The latter construct uses the human Dlx enhancer element to promote viral transduction in forebrain GABAergic cells and contains a FLEEx switch making eGFP transgene expression dependent on the presence of Cre recombinase within a transduced cell” (Lines 351-353).

Regarding the use of Flp-dependent viral constructs:

“Next, we co-injected AAV1-mMORp-FlpO (titer: 1.4×10^{12} gc/mL) virus with a pan-neuronal, human Ef1 α promoter-driven Flp-dependent mCherry reporter (AAV9-Ef1 α -fDIO-mCherry, titer: 2.4×10^{13} vg/mL) into a separate cohort of C57BL/6J mice (N=3 male mice) to demonstrate an additional recombinase-based targeting strategy to label putative MOR+ neurons. We found that mMORp-FlpO successfully drove Flp recombinase dependent recombination within Ef1 α -fDIO-mCherry transduced cells and observed numerous mCherry labeled cells in both mPFC and S1 (Fig. 5c-d)” (Lines 357-362).

Regarding the use of a retro-orbital injection of our PHP.eB virus:

“ In adult C57BL/6J mice, we delivered AAV-PHP.eB-mMORp-eYFP (titer: 8.6×10^{12} gc/mL) via retro-orbital injection (50 μ L, 50:50 mix) in order to facilitate better systemic viral spread and distribution, and observed robust expression of eYFP in cells throughout the spinal cord and brain in both sagittal (Fig. 5e) and coronal tissue preparations (Supp. Fig. 10), with insets shown across representative coronal sections of high magnification pictures of mMORp-eYFP+ cells observed in several of the key structures of interest examined in our intracranial focal injection studies” (Lines 366-371).

3. **“AAVs are generally not potent towards transducing microglia, and it is unclear whether the lack of microglial transduction is due to the capsid or the promoter in this context, and the repeated conclusions drawn by the authors to define the m/hMORp promoter to be NeuN specific due to its lack of expression in Iba1+ is ambiguous to the reader, may be misleading for non-AAV background readers as even with ubiquitous promoters, AAVs have not shown promise to transduce microglia significantly in vivo.”**

The reviewer is correct in pointing this out, as the transduction efficacy of AAVs in microglial cell types has been a topic of discussion within the literature for some time. In **Lines 140-144** of our original manuscript submission, we addressed by both noting and citing relevant studies that have directly investigated and reported on these findings, and mentioning that the design of our constructs may have produced in them a predisposition for transducing neuronal cell types over microglial cell types due to the inclusion of a PU.1 transcription binding factor region, a genetic element which has been demonstrated to repress MOR expression in myeloid-lineage cells, a category which microglia fall into. While this may indeed be directly related to the transduction profile we see for our constructs (i.e. producing more exclusive transduction in neurons compared to microglia, as quantified in **Supplemental Figures 5, 8 and 13** above), we have edited the text within the updated **Results** and **Discussion** section of the manuscript to make the assertions that constructs are NeuN+ or neuronal cell type specific less definitive, and placed emphasis again on the elements of both our constructs and AAVs as a whole that may make them less efficient at transducing microglia.

*Updated text within the **Results** section discussing the selectivity of our viral constructs for neurons over glia/microglia:*

“These virally transduced mPFC cells were co-labeled for the neuronal marker NeuN, while none appeared to overlap with certain glial marker, such as the microglial marker Iba1, a cell type that has previously been speculated to harbor active *Oprm1* promoters (Fig. 1e)” (Lines 141-143)

“As a final test of fidelity and specificity, we wanted to determine if the design of our *MORp* constructs would allow for transduction to occur predominantly in neurons, as opposed to more broadly across neuronal and glial cell types. AAV1-*mMORp*-eYFP was injected in a cohort of mice into mPFC, CeA and VTA, and a glial marker IHC panel was conducted to examine the overlap of *mMORp*-eYFP signal with CC1 (oligodendrocytes), PDGFR α (oligodendrocyte precursors), GFAP (astrocytes) and Iba1 (microglia) antibody staining (Supp. Fig. 5). In all cases, minimal to no signal for glial marker staining was noted on cells positive for *mMORp*-eYFP across all regions of interest (CC1: mPFC=1.9%, n=4 ROIs from N=2 mice; CeA=1.4%, n=4, N=2; VTA=4.3%, n=2, N=1; Supp. Fig. 5a-b; GFAP: mPFC=1.2%, n=3, N=2; CeA=1.6%, n=3, N=2; VTA=0.9%, n=2, N=1; Supp. Fig. 5c-d; PDGFR α : mPFC=1.0%, n=2, N=1; CeA=5.1%, n=3, N=2; VTA=3.5%, n=2, N=1; Supp. Fig. 5e-f; Iba1: mPFC=1.3%, n=2, N=1; CeA=1.6%, n=2, N=1; VTA=1.2%, n=3, N=2; Supp. Fig. 5g-h), indicating a higher transduction preference of our viral constructs for neurons. Additional gene expression analyses performed on cultured human neuronal (SHSY5Y), astrocytic (A172), microglial (C20) and murine microglial (N9) cell lines following transduction with AAV1-*hMORp*-eYFP or AAV1-*mMORp*-eYFP, respectively, revealed the normalized expression of *m/hMORp*-eYFP to be lower in C20 cells compared to SHSY5Y cells, and reduced across other cell lines (one-way ANOVA, $F=3.764$, $P=0.0409$, SHSY5Y v. A172: $P=0.073$, SHSY5Y v. C20: $P=0.0251$, SHSY5Y v. N9: $P=0.0645$; Supp. Fig. 6f), while *OPRM1/Oprm1* expression across glial lines was significantly reduced compared to SHSY5Y cells (one-way ANOVA, $F=413.5$, $P<0.0001$, SHSY5Y v. all: $P<0.0001$; Supp. Fig. 6e). Apart from the underlying predilection of AAVs for infecting neurons over other glial cell types, the exclusion of expression specifically in microglia cell-types may result from the inclusion of a PU.1 transcription factor binding region, which has been previously demonstrated to repress MOR expression in myeloid-lineage cells, including microglia” (Lines 191-211).

“To confirm that this expression profile was selective for neurons within rat tissue as well, a similar glial marker IHC panel was conducted using CC1, GFAP and Iba1 antibody staining. As in mice, cells within rats transduced by *mMORp*-eYFP showed little to no overlap with signals for glial marker antibody staining within CeA (CC1: 1.0%, n=2, N=1; GFAP: 0.6%, n=2, N=1; Iba1: 0.9%, n=2, N=1; Supp. Fig. 8a-b) or VTA (CC1: 0.5%, n=2, N=1; GFAP: 1.5%, n=2, N=1; Iba1: 2.3%, n=2, N=1; Supp. Fig. 8c-d). Within shrew tissue, similar patterns of transduction were also found, with *mMORp*-eYFP restricted exclusively to non-Iba1+ cells (0.6%, n=3, N=2; Supp. Fig. 8e), and overall spread of the virus noted to be relatively restricted to the borders of the AP/NTS structure where the majority of MOR+ cells reside (N = 2 female shrews, Fig. 2e).” (Lines 236-244).

“Glial marker staining with antibodies against CC1, GFAP, PDGFR α and Iba1 in macaque dACC transduced tissue (**Supp. Fig. 13a-c**) revealed similar results to those observed in mouse, rat and shrew tissue assessed for *MORp*-eYFP signal, in that few to no cells were noted to co-label for eYFP and any glial marker signal (CC1: 1.6%, n=2 ROIs, N=1 macaque; GFAP: 1.2%, n=2, N=1; PDGFR α : 1.5%, n=2, N=1; Iba1: 0.5%, n=4, N=1; Supp. Fig. 13d-g).” (Lines 400-404).

*Updated text from the **Discussion** section addressing the higher preference of our viral constructs for transducing neuronal cells over glial/microglial cell types:*

“Despite this though, current IHC and ISH data clearly show that it is likely our viral constructs are indeed transducing the desired target neuronal populations within these nuclei. Indeed, the restricted expression of our viral constructs to predominantly neuronal cells is in agreement with not only the overall design of these constructs, which included transcriptional elements known to repress expression in myeloid-like cells such as microglia, but also the natural

predilections of AAVs to show greater transduction efficacy at neurons when compared to glia, a detail supported by the findings presented in our glial marker staining panels across species. Additional gene expression assays used to address the concern of possible transduction events specifically in microglia seem to further support the higher preference for neuronal transduction inherent to our constructs, despite minimal upticks in *mMORp*-eYFP or *hMORp*-eYFP observed across cultured cell lines, which may be explained by cell line heterogeneity and/or minimal viral leak that may be present at higher titers. Despite the identity of the cells primarily transduced by our constructs across the select brain regions of interest discussed above appearing to be neurons, further testing will be necessary to ensure this to be the case in other regions of interest possessing MOR+ cells responsive to, and participatory in, the modulation of pain salient stimuli” (Lines 485-498).

4. **“The authors need to further clarify with supporting experiments to test the transduction of microglia with the *m/hMORp* promoter using a delivery system that has been previously demonstrated to transduce microglia *in vivo* significantly, eg. Lin et al, Nat. Methods, 2022 PMID: 35879607.”**

The reviewer has noted a need to attempt to package our *m/hMORp* promoter constructs into a viral delivery system that has been reported to more selectively transduce microglia over other neuronal and glial cell types, specifically the AAV capsid variants AAV-cMG or AAV-MG presented in the recent Nature Methods paper by Lin et al; in fact, we referenced and discussed this study in our original manuscript submission, see **Ref. 29**. However, contrary to the data in Lin et al., to the best of our knowledge there are few labs have been able to repeat these results (personal communications with labs at Penn, CHOP, Harvard, and Duke).

Importantly, in collaboration with our Penn colleagues Dr. Christopher Bennett and Dr. Mariko Bennett, two prominent glial researchers, we recently tested an AAV-MG-BFP *in vivo*. Specifically, we injected 1 μ L of 1.21 x 10¹³ vg/mL (MG1.2) or 1.64x10¹²vg/mL (cMG), intracranially into P4 mouse pups. Brains were harvested at P21, sectioned, and immunostained for Iba1 to label microglia and macrophages. While we observed the transduction of cells that morphologically appear to be neurons and astrocytes, there is no apparent labeling of Iba1+ microglia. As such, our results led us to the decision to not package our *m/hMORp* constructs into these AAV serotypes, as this approach would not address the reviewer’s comment nor yield reliable results.

The greater goal of the work presented in our manuscript here was not to litigate whether microglia express the native *Oprm1* promoter sequence in their genome, but to highlight that *m/hMORp* packaged into most standard AAVs and serotypes provide selective genetic access to neuronal cell types over glial cell types (**Supplementary Figures 5, 8 and 13** above). To better explain this goal within the manuscript, we have updated the text in the **Results** and **Discussion** sections to remove any strong language asserting that our work shows microglia to be MOR/*Oprm1* promoter- (as noted in the section above), and instead focus on how the design of our constructs (and the nature of AAVs in general) may lead to more prominent transduction of neurons over microglia and other glial cell types, as our new supplemental results clearly support.

Motor Cortex (C57BL/6J mice, P21)

AAV.MG1.2-CAG-BFP (titer: 1.21 x 10¹³ vg/mL)

Representative images of AAV.MG1.2-CAG-BFP viral expression in mouse cortical tissue. High magnification images show expression patterns for blue fluorescent protein in transduced cells from the motor cortex of C57BL/6J mice following intracranial injection of AAV.MG1.2-CAG-BFP virus, as indicated by the white arrow heads (**a**). Staining for Iba1 (1:500, Wako, 019-19741) as a marker for microglial and macrophages is shown in **b**, while the overlap of BFP signal and Iba1 staining is shown on the right in **c**, with cells in which both BFP and anti-Iba1 signal are present indicated by yellow arrowheads. Little to no overlap of the two signals is present within cells observed throughout the region, with

morphological patterns of cells that were transduced appearing quite different from that of cells stained for anti-Iba1 signal. Scale bar = 100um.

5. **“Optional – Microglia transduction in vitro using existing AAV serotypes may be helpful but only as additional supporting data in addition to in vivo as the field has conflicting views on the translatability of in vitro systems in vivo.”**

We have provided new data, summarized in **Supplemental Figure 6** of the updated manuscript, in which we followed the reviewer’s suggestion and transduced different cultured lines of human and murine cells. Using the commercially available cell lines A172 (human astrocytes), C20 (human microglia), N9 (mouse microglia) and SHSY5Y (human neurons), we cultured and transduced these cells with the species appropriate *MORp* construct (either AAV1-*hMORp*-eYFP for all human cell lines, or the AAV1-*mMORp*-eYFP for the mouse cell line) for ~5 days before collecting and prepping cells for qPCR analysis of total *OPRM1*, *Oprm1* or *EYFP* gene expression, or for fluorescence imaging of eYFP expression in the culture wells. We found the average normalized expression of *OPRM1* (in human cell lines) or *Oprm1* (in mouse cell lines) to be low to relatively non-existent in A172 human astrocytes (0.00051, n=4), C20 human microglia (0.00039, n=4) and N9 mouse microglia, respectively (0.000076, n=4), while human neuronal SH-SY5Y cultures showed relatively higher *OPRM1* expression overall (1.004, n=4). Similarly, averaged *EYFP* normalized expression was found to be highest in the SH-SY5Y cells (1.182, n=4) and lower by comparison in human A172 astrocytes (0.350, n=4), human C20 microglia (0.147, n=4) and mouse N9 microglia (0.325, n=4). Representative images taken of well plates containing transduced cells of each line described above also demonstrated relatively low eYFP signal in A172, C20 and N9 cells, while SH-SY5Y cells showed robust eYFP signal throughout the plate. The **Results** section has been updated to include these findings, and additional acknowledgement of how these results may indicate that our constructs, with their current design packaged into AAVs, are best able to transduce MOR/*Oprm1* promoter+ neurons over other glial cell types, especially *in vivo*, has been included in the **Discussion** as well (see above responses for text regarding **Results** and **Discussion** section updates).

New Supplementary Figure 6. Representative, high magnification images of individual wells from plates of cultured human SHSY5Y neurons (a), human A172 astrocytes (b) and human C20 microglia (c) transduced with the hMORp-eYFP viral construct, as well as mouse N9 microglia (d) transduced with the mMORp-eYFP viral construct are shown on the left. Scale bars for all images=100um. e-f, Summary data of the relative gene expression for both *OPRM1* (in human lines) or *Oprm1* (in murine line) and *EYFP* are shown on the right. Expression for genes of interest across lines is normalized to the house keeping gene GAPDH, and all normalized expression values are displayed as relative to the expression of either *OPRM1*/*Oprm1* or *EYFP* in SHSY5Y cells. Normalized expression of *OPRM1* and *Oprm1* was found to be low in A172 human astrocytes (0.00051, n=4), C20 human microglia (0.00039, n=4) and N9 mouse microglia, respectively (0.000076, n=4), while human neuronal SH-SY5Y cultures showed relatively higher *OPRM1* expression overall (1.004, n=4). Similarly, normalized *EYFP* expression was found to be highest in the SH-SY5Y cells (1.182, n=4) and lower by comparison in human A172 astrocytes (0.350, n=4), human C20 microglia (0.147, n=4) and mouse N9 microglia (0.325, n=4). All data in graphs are presented as averages \pm the s.e.m. MOI = multiplicity of infections.

6. ***“If the authors choose not to go down the route of validating microglial transduction, it is critical to re-draw the conclusion across figs/text and move the microglia related data to the supplementary, clarifying the ambiguity to the reader.”***

As mentioned above, we have added new data and the updated text also listed out above throughout the **Results** and **Discussion** sections of the updated manuscript, with a focus on making sure it is clear that due to the design of our constructs (i.e. inclusion of a PU.1 TF element) and the inherent nature of AAVs, our viral constructs may be best suited for applications aimed at studying MOR/*Oprm1* promoter+ neuronal cell types over glial ones that may or may not express the same receptor or promoter sequence in their genome, in particular microglia. While we were unable to go down the route of attempting to directly transduce microglial with the serotype variant AAVs described in Li et al. 2022 (PMID: 35879607), we hope that the supplementary data we have provided, along with the updated language to our manuscript has served to provide a more clear distinction that our constructs in their current form of packaging/design are best suited for transducing neurons, and not that they would have no utility for transducing microglia if a group that did have access to the AAV-cMG or AAV-MG1.2 serotypes in the future was to attempt to perform studies/applications of our constructs for their own purposes by packing them within these serotype variants.

Likewise, we hope that the updated text makes it clear that we are not drawing any conclusion on whether or not microglia definitively expression MOR or the native *Oprm1* promoter, as this was once again not the goal of the work presented in our manuscript. For these reasons, and since an active group of researchers within the neuroscience community specifically are still pursuing the question of microglial MOR expression, we believe it to be useful to leave the main figures which display representative IHC images of AAV1-*m/hMORp*-eYFP and Iba1 staining unchanged, as we have now updated the manuscript's contents to directly point out and show to readers that these constructs may be best used for targeting neurons over microglia for MOR specific studies. Supplementary data tabulating the quantification of Iba1 and *MORp*-eYFP overlap will remain in the Supplement due to space and size concerns for our main body text figures, but as mentioned previously, we have updated text in the **Results** and relevant figure legends to direct readers to this part of the paper to view these data themselves, and also provided commentary on this in the **Discussion**.

7. ***“How does m/hMORp1 promoter perform compared to the widely used pan-neuronal hSyn promoter? Based on the information provided in the manuscript, it is unclear to the reader as to why they should choose the new promoter over the commonly used hSyn. The data in Fig. 1e. shows overlap of pan-neuronal marker NeuN which is used across any other neuronal promoter. hSyn promoter is less than 500 bp compared to ~1500 long m/hMORp promoter. For AAV researchers, this is a huge size difference, and may not be welcomed unless the authors can provide quantitative supporting data that shows the differences in either cellular specificity or efficiency in the relevant rodent models.”***

The reviewer raises a very interesting and important point here regarding how useful our viral constructs may end up being when compared directly with more commonly available, and widely used, pan-neuronal promoter targeted constructs, like those driven by the *hSyn* promoter. We attempted to directly address the reviewer's request and have provided a new dataset and Supplementary Figure within the updated manuscript outlining *mMORp/hSyn* co-injection experiments we conducted in the CeA and VTA of C57BL/6J male mice (**Supplemental Figure 2k-n**). These experiments demonstrate that when intracranially injecting in an AAV5-*mMORp*-eYFP along with an AAV5-*hSyn*-mCherry (viruses mixed 1:1), we are able to both see robust expression of both reporter fluorophores at the site of focal injection, and note a clear distinction in the transduction patterns of both viruses. Analyses from the quantification of total cell counts across both regions show that while there is indeed transduction of the same cells by both viruses, not only did we note a greater number of eYFP+/mCherry- cells compared to eYFP+/mCherry+ cells in the ROIs used to produce this data, but that the total number of cells transduced overall within these regions by the *hSyn* promoter driven virus was greater or about equivalent to that of the cells transduced by our *mMORp* promoter driven virus. As both the mouse CeA and VTA are known to possess a high number of GABAergic cells/interneurons, a population of cells known to be more preferentially targeted by *hSyn* promoter driven viruses, our findings appear in line with what might be expected from using a more ubiquitously targeting promoter like *hSyn* when compared to one with more selective targeting, like our *mMORp* promoter construct. It is also important to note that in tissue sections we analyzed (showing the mouse amygdala, including both the central nucleus (CeA) and basolateral nucleus (BLA), we find little to no AAV5-*mMORp*-eYFP expression in BLA cells, while robust transduction of cells within this structure by the AAV5-*hSyn*-mCherry virus was seen (see **Supp. Fig 2k**). The BLA expresses little MOR compared to the CeA (Wang et al. Neuron. 2018, PMID: 29576387). Thus, as expected, *mMORp*-eYFP was not expressed in *Oprm1*- BLA cells, while *hSyn*-mCherry robustly expressed in BLA neurons, which highlights the clear experimental gains of using *mMORp* if a reader's goal is to gain genetic access to *Oprm1*-expressing cell-types.

In relation to the reviewer's comment regarding the size of our *m/hMORp* promoter sequence compared to that of the *hSyn* promoter (448 bp), while we do agree that this could complicate creating novel constructs combining very large transgenes with the *m/hMORp* promoter, we would like to note that our ~1500 bp sequence still leaves around ~3200 bps of space in most constructs that would be suitable for packaging within an AAV (note: the highly used *Camk2a* promoter is 1289 bp). This ~3200 bps is more than sufficient to package common transgene for most circuit-neuroscience

experiments, such as DREADDs and GECIs, as we have demonstrated. This is also why we have created *mMORp*-Cre-mCherry and *mMORp*-FlpO, which can be co-injected with a multitude of intersectional AAVs or rodent reporter lines. Additionally, and while it falls out of the scope of our studies here, readers who may find use for our sequences and constructs would not be barred from attempting to incorporate them into larger constructs/plasmids more suitable for packaging into viruses rated for a higher genomic load than the AAV, such as lentiviruses with 10 kb packaging capacity.

In vivo expression validation of *mMORp* construct compared w/ *hSyn* promoter construct, C57Bl6/J mice: AAV5-*mMORp*1-eYFP + AAV5-*hSyn*-mCherry (1:1)

Panels S2k-n from updated Supplemental Figure 2. Representative images and accompanying quantification and analysis of the transduction patterns for cells in either the mouse CeA or VTA transduced via co-injection of our AAV5-*mMORp*1-eYFP virus mixed 1:1 with an AAV5-*hSyn*-mCherry fluorescent reporter virus. High magnification images of the CeA shown in the **S2k** display an ROI of cells transduced by both the *hSyn*-mCherry and *mMORp*1-eYFP viruses, with the signal for each fluorophore shown separately, and merged together in order to better display the individual and overlapping patterns of transduction for each, with sample cells positive for either *hSyn*-mCherry alone (red arrows), *mMORp*-eYFP alone (green arrows) or both viruses (white arrows) denoted in the right most image. Fluorescent signal for both viruses is presented against DAPI nuclear staining in the left most image, and the borders of the CeA and BLA are also shown. It should be noted here, that signal for the *hSyn*-mCherry virus alone can be prominently noted within the BLA, a region known to possess low MOR expression. A similar set of images is shown for the VTA in **S2l**. Scale bars for both images are set at ~100um. The corresponding analyses for total *hSyn*/*mMORp* (+), *hSyn* (-)/*mMORp* (+) and *hSyn* (+)/*mMORp* (-) cells are shown below image sets in **S2m** (CeA) and **S2n** (VTA), with bar graphs displaying the total number of cells quantified within each region displayed as means (with individual data points overlaid; error bars indicate S.E.M.), and the average number of cells in each group presented as percentiles in the parts of a whole graphs. On average, it was noted that a greater number of cells transduced by the *mMORp*1-eYFP virus did not co-express *hSyn*-mCherry, indicating that the two cell populations targeted by both promoters are not 1:1 in overlap, and that the *mMORp*1 construct may indeed be able to target a more selective population of cell across putative MOR+ brain regions/cell populations than a generic promoter driven virus could.

Updated text from the Results section commenting on these co-injection studies:

“To further demonstrate that our *mMORp* constructs afford a level of selectivity and utility for targeting putative MOR+ cell populations that stand apart from other more generic promoter driven constructs, we performed co-injections of an AAV5-*hSyn*-mCherry reporter virus mixed one to one with a serotype-matched variant of our *mMORp*-eYFP virus (AAV5-*mMORp*-eYFP) into two of the representative MOR+ regions assessed in our previous specificity studies (CeA and VTA, **Supp. Fig. 2k-l**). In both the CeA and VTA, we noted that the vast majority of *hSyn*-mCherry and *mMORp*-eYFP cells comprised separate populations (CeA: mCherry=61.6%, eYFP=24.1%, mCherry/eYFP=14.3%, n=4 ROIs from N=2 mice; VTA: mCherry=37.3%, eYFP=40.7%, mCherry/eYFP=22.0%, n=2, N=1; Supp. Fig. 2m-n), with mCherry+/eYFP+ cells only making up a small percentage of total *mMORp* transduced cells (CeA=37.4%; VTA=35.1%). Transduction patterns for co-injections into the CeA also showed noticeably greater spread of the *hSyn*-mCherry virus into the MOR- BLA, while

mMORp-eYFP transduced cells were found to be more restricted to the general boundaries of the CeA itself, further supporting the transduction selectivity of the *mMORp* constructs over those which may utilize more common promoter sequences in their design.
" (Lines 178-190).

8. ***"Recommend using MOI (multiplicity of infection) nomenclature for AAV transduction assays across in vitro cell culture systems than using concentration as this is less meaningful without understanding the cell count per well."***

We thank the reviewer for reminding us of the general readership of Nature Communications and have adjusted the text throughout the **Results** and **Methods** sections (as well as within all relevant figures and figure legends concerning *in vitro* assay studies) to make use of the suggested nomenclature.

9. ***"A total dose per animal or a concentration following by volume (as shown in Fig. 1d). Consistency is helpful for readers."***

We have updated all **Figures** and figure legends to reflect the labeling schema suggested here by the reviewer.

10. ***"Except for PHP.eB and PHP.s, I didn't really come across a rationale for a broader audience to understand the use of the different serotypes (AAV1, 5, 8, 9) across studies. Will be helpful for a call out in the text or in methods."***

We agree that the reviewer is correct in that, for a more broad audience, we need to specify whether there was a rationale used to guide why specific serotypes or capsid variants were used for individual experiments. To address this, we have added new text within the **Discussion** to note that the serotype used for a specific experiment was not premeditated or done by design in most cases, but simply to make use of the viruses which were most efficient and successful for us to produce at the time with a high enough titer to be useful in *in vivo* injections and experiments, and that numerous other studies have indicated are viable for transducing neural tissue with high fidelity (Van Vliet et al.). Our collaborator, Dr. Charu Ramakrishnan, who produced all the viruses discussed in this manuscript, noted that the vector core facility which she utilized for their production (Stanford GVVC) reported higher titers when making AAV1 serotype viruses compared to other serotypes, and thus we decided to produce our initial batches of viruses using AAV1 in order to maximize success. Subsequent AAV5 serotypes for select *m/hMORp* variants (specifically the AAV5-*mMORp*-eYFP virus used in studies summarized by **Supp. Fig. 2k-n**) was selected to match the serotype of the AAV5-*hSyn*-mCherry virus co-injected along with it, as to not introduce a potential confound for the differences in transduction efficacy noted for each to be able to be attributed to a difference in serotype, and thus any effects of tropism that may have introduced within a region like the CeA or VTA of the mouse where they were injected. The serotypes of other viruses used within the experiments discussed in the manuscript, specifically for viruses that were purchased from vendors and not produced by our group, were similarly not selected for any premeditated reason, but purely due to their availability at the time. Within the **Discussion**, we address all this by noting that our studies here making use of different serotypes of viruses were in no way exhaustive or done to show that specific serotypes work better in specific brain regions or in combination with other specific viruses of the same or different serotypes, as our primary goal was mainly to demonstrate the utility of our *m/hMORp* constructs. We note that readers must thus be aware that further testing for each use case of any virus is important to account for the effects of tropism, and determine if one specific serotype will work best for the specific question they are attempting to ask outside of transduction specificity that our constructs packaged within a specific AAV might afford them.

For the PHP.eB and PHP.s capsid variants, updated the relevant text in the **Results** and **Discussion** to note their utility in transducing more exclusively CNS or PNS based cell, respectively, when compared to more generic AAVs, and more fully explained our want to demonstrate their utility for transducing/targeting more broadly all putative MOR/*Oprm1* promoter+ cells within the CNS or areas of the PNS without the need for doing multiple focal injections (see above response sections for the relevant text).

New text added to the Discussion addressing the topic of serotype and AAV variant choices throughout the manuscript:

"Similarly, future testing will be equally necessary to determine which viral serotypes and titer concentrations will provide the greatest levels of transduction efficacy across brain regions, species, and sex variables outside of the sampling presented here. Outside of experiments in which the serotype of co-injected viruses was matched to reduce confounds from potential differences in transduction efficacy, the choice of serotype of a virus used for each of our studies was not informed by prior knowledge of viral tropism (ability of a virus to infect specific cells or tissue types⁸⁴) inherent to any specific serotypes as reported across different brain regions or neuronal cell types within the literature⁸⁵. This question of tropism and determination of the correct serotype to employ to best transduce a specific tissue or cell type is not unique to

our viral constructs, however, as consensus on the specific capsid proteins and construct elements that provide the greatest transduction efficacy within different tissue types and cell populations is still a matter of debate and investigation, requiring more meticulous documentation and data sharing amongst researchers across disciplines⁸⁶. Examining the extent of the combinatorial approaches that can be taken with the use of our viruses with other transgenic animal lines and tools for the intersectional targeting of unique CNS and PNS cell populations⁸⁷ or the dissection of mu opioidergic circuits within the brain will thus require careful testing and planning when using our MORp constructs across any and all applications.” (Lines 498-513).

11. “In methods, line 192 – AAV9 is missing although I did see this capsid was being used in the experiments.”

We have fixed this error within the text of the **Methods** section, and thank the reviewer for pointing this out to us.

12. “While the manuscript provides a detailed characterization of different systems to get to the desired cell-type targeting across species, there are several layers of control knob (capsid, promoter, genetic switch, transgenics, species, delivery) that a reader needs to understand. While the authors have tried to illustrate in figs as needed, it would be very helpful if there is any room to improve or add a supplementary summary figure.”

We thank the reviewer for this excellent suggestion and have included a new supplementary summary figure (Supplementary Figure 16) at the end of the manuscript to address this.

New Supplementary Figure 16. Summary figure outlining the design, packaging, routes of administration and expression mechanisms of the MORp constructs. Cartoons outlining the coding sequence for the *Oprm1* gene, and the complementary sequence designed for the MORp constructs matching this region, are shown in the upper left, with icons indicating the endogenous promoter region of the gene (blue box), binding areas for both repressor and activator transcription factors (red dots and teal boxes, respectively) and putative transcription start sites (TSS, purple arrows). The AAV cassette designed for the MORp constructs is shown below this, which is composed of: the custom sequence designed to be complementary to the *Oprm1* promoter (orange arrow), TSS (purple arrow) sequence found after the promoter region, the transgene element driven by the promoter (green box), regulatory elements like the woodchuck hepatitis virus posttranscriptional regulatory element (WPRE) which aides transcribed mRNA in forming a tertiary structure and in nuclear export and a polyadenylation sequence (pA) that caps the tail of the mRNA and aides in nuclear export. All of these elements are contained between two inverted terminal repeat sequences (ITR). The cassette, contained within an AAV transfer plasmid, can be packaged into different variants of AAV depending on the specific serotype of the AAV desired to use. The serotype of the AAV is determined by the unique antigenicity and surface structure of the viral proteins (VP1-3) that comprise the viral particle’s outlay layer, known as the capsid. Different conformations and folds of the capsid, as well as capsid protein interactions, define each serotype, and impart on them different levels of selectivity for the tissue and cell types they can most successfully transduce, with AAV1, 2, 5 and 8 historically showing higher preference for neuronal tissue and cell types. AAV serotypes 1-11 are naturally occurring, while synthetic variants such as AAV.PHP.eB or

AAV.PHP.s have been produced using AAV targeted evolution platforms. Packed AAVs can then be introduced into tissue or cell types of interest via multiple routes of administration, depending on the model system desired. For *in vivo* applications, these AAVs can be administered directly into a tissue or organ of interest via injection using a syringe, with our manuscript outlining the use of intra-cranial (brain) intra-thecal (spine) or retro-orbital (eye) based routes to deliver AAV virus either focally within a single tissue type/region of interest or throughout the entire body, respectively. For *in vitro* applications, administration of an AAV virus directly on top of cells or tissue within culture is usually sufficient to drive transduction. Once administered, viral particles will then be able to bind to cells by targeting surface receptors along their membranes and then enter said cells, where it they will release their genomic material. As AAV are usually not integrating, this exogenous DNA will instead co-opt the necessary transcriptional machinery present within the cells to drive the transcription of it's DNA into mRNA, and then eventually into protein via the use of the cell's translational machinery as well. In the case of the our *MORp* constructs, transcription will only occur in cells in which a unique combination of transcription factors are able to bind to the native *Oprm1* promoter and recruit additional transcriptional machinery (i.e. polymerases like pol II) necessary to transcribe the *Oprm1* gene to produce it's cognate mRNA transcripts, and thus eventually, mu opioid receptor protein. As such, cells transduced by our AAV-*MORp* virus that contain active endogenous *Oprm1* promoter, will also transcribe the virally introduced *MORp* construct. By contrast, cells that do not possess an active *Oprm1* promoter, even if they are transduced by the virus, will not be able to recruit the necessary transcriptional elements the *MORp* construct will need to co-opt to drive the transcription, and eventual translation, of its genetic material.

C. Response to Reviewer #3:

- 1. “The titer used in the paper varies a lot. The fact that the optimized titer, $3e+11$, is several folds lower than the commonly used titer ($1e+12$) could mean that the latter might introduce nonspecific expression. Please further explain how the optimized titer is determined.”**

We agree with the reviewer that clarification is required for this, especially so that our results can be made more interpretable to a broader audience without a background in the use of AAV viruses for *in vivo* research purposes. All viruses utilized across the experiments described in this manuscript were produced by Dr. Charu Ramakrishnan and the Stanford GVVC across multiple different runs which each had a set maximal yield. As such, we were often provided with different batches/lots of certain viral preparations that possessed different titers and inherent batch to batch variability. As such, we attempted to dilute select viruses down to differing concentrations/titers primarily if we noted any significant evidence of cell death *in vivo* when injecting viruses at the base concentrations/titers provided (ranging from $E+13$ to $E+11$), and refrained from using viruses at a dilution in which no expression was noted when checking for targeting/expression efficacy with IHC. As such, the titer of a given virus needed to be determined via a process of optimization and testing that varied from lot to lot of the same viral constructs at times, as well as for individual viral preps across construct variants (i.e. *mMORp-eYFP* v. *mMORp-hM4Di-mCherry* v. *mMORp-FlpO*). Thus, we have removed any text from the **Figures** (and any associated figure legends) and the **Results** section which may indicate that a blanket “optimized titer” may exist for each individual virus/construct. We have similarly noted in the **Discussion** section that it will be necessary for researchers who make use of our viral tools, as is the case with most other commercially available viral tools, to perform their own optimization tests with each virus/construct to ensure that it produces the desired extent of transduction to fit their specific questions or experimental parameters (see above response to Reviewer #2 for the text in question).

- 2. “Fig. S4, Sun1-eGFP is observed in striatal matrices. Fig. 1h, I see lots of Sun1-sfGFP+ neurons that are mMORp1-mCherry negative. These data mean that the *Oprm1-Cre-2A* mouse line is not specific. What is the authors' take on this?”**

Questions regarding the specificity of transgenic Cre mouse lines are always crucial to have when considering the impact they exert across many different fields and studies, and we thank the reviewer for broaching this topic. While it is true that there are many Sun1-sfGFP+/mCherry- neurons that can be noted in regions such as the dorsomedial striatum imaged from the *Oprm1^{Cre-2A}* mouse that we examined following the injection of our AAV1-*mMORp-hM4Di-mCherry* virus, there are several explanations that may exist outside of this transgenic Cre line lacking specificity alone as to why we observed this. As mentioned above, the matter of tropism, and/or the ability of a specific serotype of AAV to drive effective transduction within a specific region/tissue type or cell population could serve as one explanation as to why we did not see 100% transduction of every single Sun1-sfGFP (and by proxy, Cre+) cell within a region like the DMS. Indeed, as our intention with the work presented in this manuscript was to demonstrate the utility of our new viral tools and not definitively investigate which AAV serotypes are the most effective for driving transduction across every region of the brain, it is possible that the use of another AAV serotype (2, 5, 8, or 9) could prove to be more efficient at transducing the majority of cells within the DMS, regardless of their expression of MOR or the *Oprm1* promoter sequence, as opposed to AAV1. While an exact consensus has not been reached regarding which AAV serotypes are best suited to each specific brain structure or cell type, there has been data compiled to suggest that AAV5 may be a more effective serotype to use

within the striatum if maximal transduction of cells within that region is desired, as presented by Haggerty et al. in a recent review (Mol. Ther. Methods Clin. Dev. 2020, PMID: 31890742). Similarly, the majority of AAV viruses tend to not yield a 100% transduction rate of every single cell within a given focal injection site, with factors such as serotype once again, as well as titer and volume of the injected virus coming into play to determine how efficiently and broadly transduction occurs and how prominently a virus spreads throughout a structure. While we were able to determine a titer of our AAV1-*mMORp1*-hM4Di-mCherry virus that was sufficient to drive mCherry expression within the DMS, we did not conduct a rigorous series of tests to determine the appropriate titer and volume of virus needed to maximally transduce the majority of cells within this structure, as again, this appeared to us to be outside of the scope of our overall goals of the manuscript. As we were able to observe that, while every Sun1-sfGFP cell within the DMS of the *Oprm1*^{Cre-2A} was not transduced by our virus, the majority of cells that were transduced by the virus were indeed Sun1-sfGFP (and Cre) positive (87.3%, N=3 mice, n=4 ROIs, see **Supp. Fig. 4b**), a result which was also mirrored in an additional *Oprm1*-Cre line when examining Cre+ cells transduced by this virus (90.4%, N=2, n=3, **Supp. Fig. 4f**), we took this result to at least indicate that our viruses retain a relatively high degree of specificity, regardless of serotype, titer or volume complicating overall penetrance in the DMS. While it may thus prove worthwhile for researchers interested in investigating the specificity of the *Oprm1*^{Cre-2A} line, and overall Sun1-sfGFP expression in the DMS, to make use of an AAV5 variant of our *m/hMORp* constructs, this kind of study and speculation on the specificity of this line seemed to be outside of the work presented in our current manuscript. As a paper outlining all the specificity and selectivity studies undertaken to validate the *Oprm1*^{Cre-2A} line has recently been peer reviewed and published (Mengaziol et al. PLoS One, 2022. PMID: 36534642), it is our hope that this work can be referenced and reviewed when questioning the veracity of the *Oprm1*^{Cre-2A} as opposed to the work presented in our manuscript here, which focused primarily on our viral tools. Lastly, we note that we rigorously quantified *mMORp* expression specificity using multiple methods, including a second *Oprm1*-Cre line that was created in Dr. Richard Palmiter's lab (U. Washington), reporter mouse line fluorescence (Sun1-eGFP) plus anti-Cre immunohistochemistry to account for possible lineage recombination of the reporter any time prior to tissue collection, and fluorescent *in situ* hybridization of endogenous *Oprm1*.

3. ***“Fig. 1n and Fig. S5, in addition to specificity/selectivity, it would also be informative to know the sensitivity, i.e., regarding Cre line as “ground truth,” how much of the total targeted population can be successfully captured by the mMORp construct (basically means comparing MOR+ and MOR- % in all Cre+ cells).”***

Expanding on our response above, this comment also concerns a question of the penetrance of our virus, as well as the veracity of the Cre lines used in our study as representing the “ground truth” for *Oprm1* expression in neural cell types throughout the mouse brain. Similarly to what we discussed above, the total penetrance of a particular AAV can also be dependent on a number of factors, such as the tropism of a certain serotype within a given region/cell population, as well as the final titer/concentration of the virus used and the total volume injected, all factors that can affect the overall transduction efficiency and spread of the virus in general. All these factors can influence the total number of cells at a given focal injection site, or within a specific region/structure, that are transduced, thus requiring multiple rounds of testing and optimization in order to determine what combinations of serotype, titer and volume might be necessary to transduce the greatest number of cells with a single injection in a specific brain region, in a specific animal sex, at a specific age of the animal. We have provided a set of sample data from our analyses of the expression of our AAV5-*mMORp1*-hM4Di-mCherry in the mPFC, CeA, VTA and DMS from the *Oprm1*^{Cre-eGFP} line to demonstrate the variability in “penetrance” patterns we noted for this version of our viral construct below. As our overall goal with the work presented in this manuscript was to demonstrate the broad utility of our viral constructs for a large readership base/researchers across multiple scientific disciplines, and in general show that the majority of cells which are transduced by our virus are by and large positive for MOR/the *Oprm1* promoter (which we feel we have achieved in the data throughout the manuscript as a whole), we did not focus as directly on this depth of analysis and testing for every virus we present in the manuscript, for every use case discussed and for every brain region we injected our viruses into for validation and functional studies (see above responses to Reviewer #2 for additional text added to the manuscript regarding tropism and serotypes as well).

Viral penetrance quantification performed within the CeA, mPFC, DMS and VTA of the *Oprm1*^{Cre-GFP} transgenic mouse line (complementing the summary graphs in Fig. 1n and S4e-h). Total *mMORp*-mCherry+/Cre+ or *mMORp*-mCherry-/Cre+ cell counts quantified within a predetermined region of interest (ROI) imaged from the *Oprm1*^{Cre-GFP} transgenic mouse line for a given region are shown as bar graphs on the right, and the summary data for the total percentage of Cre+ cells that were transduced by the *mMORp* virus within said ROIs are shown as parts of a whole graphs on the left. Specifically, quantification was achieved by counting all Cre+ cells within a given ROI, followed by counting the total number of these cells which overlapped with mCherry signal, then comparing *mMORp*-mCherry+/Cre+ counts with the total Cre+ population counted. Results are shown for the CeA (~49.1%, N=3, n=5, a), mPFC (~32.8%, N=3, n=3, b), DMS (~3.2%, N=3, n=3, c) and VTA (~49.5%, N=3, n=5, d). Data presented in bar graphs represent the mean of the individual data points shown overlaying each, with error bar representing the standard error of the mean (S.E.M.).

4. “Line 186: is the *mMORp* sequence also present in rats and shrews? Also related – is the *hMORp* sequence present in macaques?”

The reviewer brings up an apt question regarding the homology of our *m/hMORp* construct sequence across animal species, and how complementary our sequences are to the native *Oprm1* promoter sequence within each model organism in which we tested these viruses. To address this, we conducted multiple sequence alignment comparisons using NCBI’s BLAST analysis tool to examine the overlap and coverage of our *m/hMORp* sequences against those of the native promoter sequence for *Oprm1* in these species, if a full genome was available to conduct these analyses. As such, we were only able to conduct these comparisons for mouse, rat, macaque and human, as a full genome for the Asian house shrew does not exist as of yet. These results are summarized in an updated version of **Supplemental Figure 1**, specifically in **Supp. Fig. 1e**. In a direct response to the reviewer’s questions here: we were able to note that our *mMORp* construct showed ~84% homology with the native rat *Oprm1* promoter sequence, and that our *hMORp* sequence demonstrated a ~96% level of homology in the macaque, indicated that sequence of both of the *mMORp* and *hMORp* sequence appear to align quite accurately with the native *Oprm1* promoter sequence found in the rat and macaque, respectively. Additional text has also been added to the **Discussion** section of the manuscript to draw attention to these comparisons.

e Sequence homology of *mMORp1* & *hMORp1* constructs w/ native MORp sequences

Target Genome	mMORp1 (Mouse)		hMORp1 (Human)	
	Coverage Area	Percent Homology	Coverage Area	Percent Homology
Mouse	100%	100%	29%	66%
Rat	65%	84%	29%	67%
Human	21%	66%	100%	100%
Macaque	N/A	N/A	11%	96%

Panels S1e from Supp. Fig. 1. Table outlining the result of alignment comparisons of the *mMORp1* and *hMORp1* sequences with the native *Oprm1* promoter sequence of mouse, rat, human and macaque conducted using the NCBI Basic Local Alignment Search Tool (BLAST). BLAST comparisons were conducted to compare the *mMORp1* and *hMORp1*

sequences across all four of these species for completeness, and the resulting total coverage and homology comparisons identified between the construct and native promoter sequences of interest are reported as percentages. A value of “N/A” was reported if alignment information was insufficient to generate a proper comparison.

*New text added into the **Discussion** to address the sequence alignment analyses shown above:*

“Indeed, as alignment analyses of the sequences used to generate both of our constructs show (Supp. Fig. 1e) both the mMORp and hMORp sequences demonstrate a high level of homology to the native OPRM1/Oprm1 promoter sequence found in rat, macaque and human (no reference genome is currently available for the shrew), suggesting the possibility to significantly enhance and expand the investigation of the opioidergic system both within and outside of the murine research community” (Lines 442-447).

5. **“Line 189: Please comment on, or summarize the various serotypes used across the manuscript and whether the authors prefer a particular one.”**

As stated above in a previous response to Reviewer #2, the serotypes used for both our MORp viral constructs were primarily determined by what serotypes of each virus provided the most favorable transduction results when tested *in vitro* by our collaborators at Stanford University, that provided the greater titer yield when packaging and scaling up their production, and have historically been shown to have good transduction efficacy in neural tissue in general (Van Vliet et al. Drug Delivery Systems. 2008, PMID: 18369962). In the case of most of the viruses discussed in this manuscript, that serotype proved to be AAV1 more often than not. The robust expression we noted of most of the viral cargo encoded by these AAV1 versions of our viruses (and noted across most of the data summarized in our **Results and Figures**) encouraged us to continue working with this serotype for the majority of our experiments. Where alternative serotypes were used, such as in one set of our cell population subtype labeling based studies used to demonstrate the combinatorial nature of our viral constructs with other commercial available Cre-recombinase dependent viruses (summarized in **Figure 5a-b**), the use of an AAV8-*mMORp1*-mCherry-IRES-Cre serotype was once again determined by a favorable *in vitro* testing and overall titer yield upon production. Similarly, for our specificity/selectivity studies shown in **Fig. 1**, the AAV5 serotype was primarily used because, at the time, the AAV5-*mMORp1*-hM4Di-mCherry virus was the only variant we possessed that encoded a fluorophore other than eYFP, and for these studies specifically, we attempted to stain for both Sun1-sfGFP and the endogenous GFP fluorophores associated with the transgenic Oprm1 Cre lines used for these experiments (see **Methods** below for more details). Tropism, and the ability for our MORp virus to hopefully transduce cells in a pattern comparable to a control virus that was co-injected along with it for a series of confirmation studies summarized in **Supp. Fig. 2k-n**, was only considered in this case to allow the AAV5-*mMORp1*-eYFP virus to complement the AAV5-*hSyn*-mCherry virus chosen for these experiments in order to avoid potential confounds. Other viruses purchased from vendors for use in a few use case studies (i.e. AAV9-*hDlx*-FLEX-eGFP and AAV9-*Ef1a*-fDIO-mCherry) were arbitrarily chosen due to prior availability in our lab, and when the use of these viruses was shown to be successful when administered in tandem with our *MORp* viruses, we did not seek to investigate the use of different serotype combinations for those experiments further. AAV.PHP.eB and AAV.PHP.s serotypes were utilized to make arguments for other potential uses for our constructs in combination with these newly emerging capsid variants to demonstrate the broad application and uses of our constructs to the reader, as we have described in the text. To make sure that these distinctions and reasons are clear to the reader, however, we had added new text into the **Discussion** section to explain our use of these various serotypes using the same language outlined above (for relevant text, see response to Reviewer #2 above).

6. **“Line 233: Fig. 3e and S6b: The authors claimed that DCZ and CNO produced similar analgesic effects, but they look very different to me. It almost seems like CNO only influenced the affective responses to alleviate pain (licking, jumping) while DCZ only affected thermal sensitivity (quick onset paw withdrawal). However, CNO and DCZ should act on the same target, so these figures are hard to reconcile.”**

We thank the reviewer for bringing this matter up, as it could be a point of potential confusion for a more broad audience as well. The routes of administration for CNO and DCZ in these experiments were not the same, as we elected to administer the CNO in series of experiments summarized in **Figure 3** systemically, while the DCZ utilized for the experiments summarized in **Supplemental Figure 9** was administered intrathecally to the sacral/lumbar spinal cord. These two routes of administration can be assumed to produce different behavioral effects based on where the drug in question will be most prominently taken up and/or restricted in its spread. For example, systemic administration of CNO can be presumed to bind hM4Di more directly at neuronal cell bodies and processes within the spinal and at the projection axons located in the brain/brainstem, while intrathecal administration of DCZ can be presumed to have a more robust effect on hM4Di locally at the spinal cord neuronal somas and local processes with little effect at the axonal projections in the brain. As the two drugs may thus be acting more prominently on different neuronal populations, this may explain the subtle discrepancies that were noted in the behaviors commented on by the reviewer, as the affective component

observed for the CNO experiments may be more dependent upon complex stimuli integration and processing within the brain itself as opposed to the spinal cord, while the more robust reflexive behavioral reductions observed in the DCZ experiments could be explained by the role spinal cord cells play in processing and transmitting noxious sensory information (over processing complex affective stimuli). While we did note the difference in routes of administration in **Figure 3b** (summary cartoon explaining the design of these DREADD experiments) we have added new text into the **Results** and figure legends for both **Figure 3** and **Supp. Fig. 9** to more directly call to this difference in route of administration used for each drug, and to briefly comment on how these may precipitate different behavioral results in mice.

*Updated text from the **Results** section making specific note of the routes of administration for both DREADD agonists used in the chemogenetic experiments summarized in **Figure 3** and **Supp. Fig. 9**:*

"We found that mMORp-hM4Di injected animals displayed an increase in mechanical threshold sensitivity in the von Frey filament Up-Down test following either intrathecal deschloroclozapine (DCZ, 10 µg; two-way ANOVA, $F=8.911$, $P=0.0105$; Supp. Fig. 9a) or systemic clozapine N-oxide (CNO, i.p., 3 mg/kg; two-way ANOVA, $F=8.521$, $P=0.0112$) compared to the within-subjects baseline thresholds and mCherry control mice (Fig. 3e)" (Lines 267-271).

7. **"Line 285: Fig. 4g, the x-axis should be minutes?"**

We would like to thank the reviewer for pointing out this typo. This has been corrected in the updated version of **Figure 4g** in the revised manuscript.

8. **"Line 318: Fig. 5c-d: Con/Fon virus has been reported by many labs to be leaky. Do the authors have control experiments showing minimal baseline expression of Con/Fon virus in SST-Cre mice without FlpO?"**

The reviewer's concerns and suggestions here were well founded, as when attempting to address this comment by injecting another round of *Sst-Cre* transgenic mice with only the AAV8-*hSyn*-*Con*/*Fon*-eYFP virus used in our previous co-injection studies on its own, we found this virus to be extremely leaky at both the mPFC and S1 injection sites. We thus decided to remove the use of this virus, and the results of our previous INTRSECT testing that made use of it along with the *Sst-IRES-Cre* knock-in mice from the updated version of the manuscript and replace it with a new use case experiment to demonstrate the utility of the AAV1-*mMORp1*-FlpO viral construct. An updated version of **Figure 5** shows this newly produced data in **panels 5c-d**. Here, we co-injected C57BL/6J male mice with a 9:1 mixture of the AAV1-*mMORp1*-FlpO virus and an AAV9-Ef1a-fDIO-mCherry virus, to drive the expression of the Flp-dependent mCherry reporter only in cells that were successfully transduced by both this reporter virus and our AAV1-*mMORp1*-FlpO virus. Representative images from these successful studies are shown in the aforementioned panels in **Figure 5**, and the **Results and Methods** sections have been updated to include new text to address and discuss these findings in place of our previous *Con*/*Fon* co-injection studies.

Panels 5c-d from updated Figure 5. Representative images showing the successful transduction, and expression, of the AAV1-*mMORp1*-FlpO and AAV9-Ef1a-fDIO-mCherry (mixed 9:1, final titers of $9.1E+11$ and $1.3E+11$ respectively) co-injected into the mPFC (**5c**) and somatosensory cortex, area 1 (S1, **5d**) of C57BL/6J male mice. Higher magnification images of the focal injection sites in both regions are provided as insets over lower magnification images of the regions. Scale bars (not shown here) are 200µm for lower magnification images, and 100µm for insets.

*Updated text from the **Results** section referencing the new data discussed above for our mMORp-FlpO studies:*

*"Next, we co-injected AAV1-*mMORp1*-FlpO (titer: 1.4×10^{12} gc/mL) virus with a pan-neuronal, human Ef1α promoter-driven Flp-dependent mCherry reporter (AAV9-Ef1a-fDIO-mCherry, titer: 2.4×10^{13} vg/mL) into a separate cohort of*

C57BL/6J mice (N=3 male mice) to demonstrate an additional recombinase-based targeting strategy to label putative MOR+ neurons. We found that mMORp-FlpO successfully drove Flp recombinase dependent recombination within Ef1a-fDIO-mCherry transduced cells and observed numerous mCherry labeled cells in both mPFC and S1 (Fig. 5c-d)” (Lines 357-362).

Updated text from the **Methods** section in reference to these new co-injection studies:

“For additional recombinase-driven labeling studies, C57BL/6J mice were injected with a mix of recombinant AAV1-mMORp-FlpO (1.42 x 10¹¹ gc/mL) and recombinant AAV9-Ef1a-fDIO-mCherry (Addgene, # 114471, titer: 2.4 x 10¹³ vg/mL) at a similar ratio of 9µl:1µl, respectively, and given a similar recovery period in order to allow for successful transduction and viral spread to occur prior to tissue collection and processing for histological analysis” (Lines 1596-1600).

9. **“Line 326: Fig.5e: if I understand correctly, the white color denotes the regions co-expressing mMORp-eYFP and FLEX-tdT. I couldn’t see lots of white colors but instead saw blue and red colors preferentially located in different areas. The data do not support the authors’ claim that eYFP and tdTomato are co-expressed in anti-Cre+ cells. If this discrepancy is due to the validity of the Cre line, then the authors should do a MOR staining and show that it correlates well with mMORp-eYFP. Please explain and provide the quantification and zoom-in images in supplementary figures.”**

In order to avoid any potential discrepancies that could be attributed to the validity of an individual Cre line, we have provided an updated version of **Figure 5** that incorporates new imaging data taken from a cohort of C57BL/6J male mice retro-orbitally injected with the AAV.PHP.eB-mMORp1-eYFP virus alone, as opposed to being co-injected along with an AAV.PHP.eB-CAG-TdTomato virus into Oprm1^{Cre} mice as performed previously. These new images (seen in panel **5e**) showing sagittal cross sections of the transduced mouse brain and spinal cord to demonstrate the spread of the PHP.eB version of our viral construct through the CNS following retro-orbital injection, are accompanied by an additional set of images provided in another new supplemental figure, **Supplemental Figure 10**, which shows coronal cross section of brains from the same group of mice retro-orbitally injected with this construct. The PHP.eB virus used for these studies possesses the same *mMORp1* backbone that all the other viral constructs discussed in our manuscript contain, and while the effects of tropism, titer and potential issues from the route of administration and total volume of virus administered could affect factors such as the overall spread and transduction efficacy of this virus, the expression of virally transduced eYFP in select and very defined brain structures known to express higher levels of MOR/MOR+ cell bodies (e.g. mPFC, CeA, VTA, PBN, etc.) and relative absence from areas of lower MOR expression (e.g. BLA), indicate that the selectivity and specificity of our viral construct appears to be maintained in this new viral capsid variant prep. These expression patterns are most evident from the coronal images provided in **Supp. Fig. 10**, and insets have been added over each showing a magnified ROI of one of the key MOR+ structures mentioned above (and which we have focused many of our other validation and use case studies on) to help more clearly demonstrate this more restricted expression. New text has also been added to the **Results** section to address this change in data presentation, as well as the **Methods** updated to reflect these changes as well.

Panel 5e from updated Figure 5. Representative images showing the successful transduction, and expression, of the AAV-PHP.eB-mMORp1-eYFP virus in sagittal cross sections of C57BL/6J male mouse brains and coronal cross sections of the spinal cord approximately 4 weeks after retro-orbital injection. Relative position along the medial-lateral access is provided next to each image to designate position through the brain. Scale bars are 500µm for spinal cord sections, and 1000µm for sagittal sections.

Updated text from the **Results** section referencing these new image sets:

“In adult C57BL/6J mice, we delivered AAV-PHP.eB-mMORp-eYFP (titer: 8.6×10^{12} gc/mL) via retro-orbital injection ($50 \mu\text{l}$, 50:50 mix) in order to facilitate better systemic viral spread and distribution, and observed robust expression of eYFP in cells throughout the spinal cord and brain in both sagittal (Fig. 5e) and coronal tissue preparations (Supp. Fig. 10), with insets shown across representative coronal sections of high magnification pictures of mMORp-eYFP+ cells observed in several of the key structures of interest examined in our intracranial focal injection studies” (Lines 366-371).

Updated text from the figure legend for **Fig. 5** addresses these changes:

“c-d, C57BL/6J mice injected in mPFC (c) or S1 (d) with a 9:1 mix of AAV1-mMORp-FlpO and Flp-dependent AAV9-Ef1 α -fDIO-mCherry. Inset high magnification images show mCherry+ cells. e, CNS expression in representative sections from the spinal cord (coronal) and brain (sagittal along the medial-lateral axis relative to Bregma; coronal sections are shown in Supp. Fig. 10) of C57BL/6J mice injected retro-orbitally with AAV.PHP.eB-mMORp-eYFP virus; scale bars = 500um (spinal cord sections) and 1000um (sagittal sections)” (Lines 682-687).

New Supplemental Figure 10. Representative images showing the successful transduction, and expression, of the AAV-PHP.eB-mMORp1-eYFP virus in sagittal cross sections of C57BL/6J male mouse brains and coronal cross sections of the spinal cord approximately 4 weeks after retro-orbital injection. Relative position along the medial-lateral access is

provided next to each image to designate position through the brain. Scale bars are 500um for spinal cord sections, and 1000um for sagittal sections.

Updated text from the **Methods** section that address these changes as well:

“For pan CNS labeling studies, a 30G insulin syringe was loaded with ~50 μ l of recombinant AAV-PHP.eB-mMORp-eYFP virus (titer: 8.6×10^{12} gc/mL) was used to perform intravenous (retro-orbital) deliveries. C57BL/6J mice were deeply anesthetized with isoflurane gas mixed in oxygen (4%) and then quickly removed from the induction chamber for the injection procedure” (Lines 1602-1605).

10. “Line 330: Fig.5g, the authors need to provide higher-quality images. There are clouds of NeuN mRNAs in DAPI-negative areas, which could mean that the mRNAs weren’t well fixed and couldn’t be used for quantification.”

We thank the reviewer for their comment on this, and have provided an updated panel in **Figure 5 (5g)** to include more high quality images of the cultured mouse DRG neurons from additional rounds of fluorescent *in situ* hybridization studies in which the cloudiness of the NeuN signal from the previous image set should have been resolved. Updated quantification results have also been provided.

Panels 5f-g from updated Figure 5. Representative images showing cultured dorsal root ganglia (DRG) neurons from mice that had been previously injected i.c.v. with either our AAV.PHP.s-mMORp1-eYFP viral construct or an AAV.PHP.s-CAG-TdTomato fluorescent reporter control virus, and undergone fluorescent *in situ* hybridization (FISH) staining for NeuN (*Rbfox3*), *EYFP*, *TdTOMATO* or *Oprm1* transcript expression via RNAscope. Images in the top panel represent positive control assays conducted on cell cultured from mice injected with the reporter virus (**5f, upper**), while images in the bottom panel are associated with wells containing DRG cells cultured from mice injected with our AAV-PHP.s-mMORp1-eYFP construct (**5f, lower**). Quantification and summary bar graphs are shown on the right (**5g**) and report the total number of cells counted in each set of cultures that were co-labeled for either *TdTOMATO* and *Oprm1* transcript, or *EYFP* and *Oprm1* transcript (presented as a percentile out of 100). Consistently, a greater numbers of cells within cultures from mice transduced with our viral construct were noted to be *EYFP/Oprm1* (+) than were *TdTOMATO/Oprm1* (+) in cultured cells from mice transduced with the control virus. Scale bars are equal to 50um, and all data in bar graphs are presented as means with error bars indicating s.e.m.

11. “Line 387: strain number should be 035574.”

This error has been noted and changed in the text of the updated manuscript.

12. “Since this paper reports a new vector tool, it would be great if the authors could share this with the community by placing the construct on addgene with instructions for at least mouse models.”

We are in complete agreement with the reviewer on this, and are already taking steps to prepare a deposit of all the MORp viral constructs discussed in this manuscript with the Stanford GVVC pending the acceptance of the manuscript. All viruses and DNA can be available upon request as well by contacting either Dr. Corder or Dr. Deisseroth.

D. Response to Reviewer #4:

1. "In Fig 2 expression of eYFP following administration of AAV1-mMORp-eYFP in other model organisms (rat and shrew) in candidate brain regions where MOR is known to be expressed. The key issue is fidelity to endogenous MOR expression and so this section would have been strengthened by some co-expression data between the eYFP and endogenous MOR (eg. using *in situ* as they effectively did in the mouse)."

The reviewer brings up an excellent opportunity here for us to further expand upon the specificity of our viral tools by demonstrating that even across different species, our *MORp* constructs are still able to show a high level of fidelity for more exclusively transducing putative MOR/*Oprm1* promoter (+) neurons. To address this, we have added new supplemental data from fluorescent *in situ* hybridization (FISH) studies (via RNAscope) we conducted in a cohort of Sprague-Dawley rats intracranially injected with our AAV1-mMORp-eYFP virus in the CeA in which we examined the co-labeling of transduced neurons for both *EYFP* and *Oprm1* transcript (**Supplemental Figure 7**). Representative images show the overall signal for both *Oprm1* transcript (**S7b**) and *EYFP* transcript (**S7c**) separately, and then merged so as to demonstrate the overall co-labeling of cells for both transcript species that was observed within a high magnification ROI used for quantification (**S7d**). White arrows are also utilized in this merged image to indicate sample cells considered positive for transcript co-labeling. Signals are also known overlaid on individual cellular nuclei via a DAPI stain (**S7a**) for completeness, with scale bars indicating ~100µm or so. These images indicate a very clear and prominent overlap of the native rat *Oprm1* transcript signal and that of the exogenous *EYFP* transcript encoded by our AAV1-mMORp-eYFP virus within the same cells/nuclei across multiple CeA ROIs. Indeed, upon further quantification and analysis, we noted that on average, approximately 71.8% of all *EYFP* (+) cells counted within the CeA of rats injected with our viral construct were also positive for endogenous *Oprm1* transcript (**S7e**), indicating that the specificity of our construct does appear to be highly conserved in rat neural tissue. By contrast though, we were unable to complete these same types of FISH experiments in tissue collected from shrews, as due to the lack of a complete genome for this model organism, no commercial probes exist to examine the expression of endogenous transcripts in tissue from these animals, and the development of custom probes to do so is both something which our lab and any immediate collaborator of our is not equipped to do (especially without a complete genome to work with and guide probe design). For the newly added rat data though, we have update the text in the **Results** section to discuss these studies, and made the necessary changes to our **Methods** section to detail the parameters of these experiments.

New Supplemental Figure 7. Representative images (**S7a-d**) showing transcript staining for native rat *Oprm1* (**S7b**) and exogenous *EYFP* (**S7c**) from successful transduction of the AAV1-mMORp1-eYFP virus in rat CeA cells. A merged image showing the overlap of both *EYFP* and *Oprm1* transcript signals (indicated at select sample cells by white arrows) is shown in **S7d**, while these signal overlay against a DAPI nuclear stain is also shown in **S7a**. Scale bars for all images are ~100µm. Quantification and analyses for the total percentiles of *Oprm1* (+), *EYFP* (+) and *EYFP/Oprm1* (+) cells observed across multiple ROIs counted from the CeA of rat injected with the *mMORp*-eYFP virus (N=1 rat, n=2 ROIs) are shown in **S7e**, with total cell counts shown in a summary bar graph (data presented as means overlaid with individual data points; error bars indicate S.E.M.) and summary parts of a whole charts showing the average percentage of cells positive for each individual transcript (left), and the average percentage of *mMORp*-eYFP cells that were either positive or negative for *Oprm1* transcript co-labeling. These results indicate that the majority of *mMORp*-eYFP (+) cells within the ROIs quantified in the rat CeA were indeed *Oprm1* transcript positive as well.

Updated text from the **Results** section that references this new dataset:

“Additional FISH staining of CeA tissue also showed similar results to those in mouse tissue, with most transduced cell bodies positive for EYFP transcript co-labeled for native rat *Oprm1* transcript (71.8%, n=2, N=1; Supp. Fig. 7a-e). Transfected neurons were also once again observed to be restricted primarily to the targeted regions of interest, indicating nominal levels of off-target spread or transduction of MOR- neurons” (Lines 232-236).

2. ***“In Fig 3 there is quite a lot of inter-individual variation in the behavioural assays in response to CNO eg just looking at E some animals show large shifts in mechanical thresholds, others less so and one animal no change. Given that the authors show that they can assess transduction with mMORp-hM4Di-mCherry was there a relationship between transduction efficacy and behavioural response?”***

Unfortunately, we are unable to access the transduction efficacy across the length of the spinal cord around where our AAV1-*mMORp*-hM4Di-mCherry virus was intrathecally injected and correlate the magnitude of *mMORp*-hM4Di-mCherry expression with their behavioral profiles, as we did not collect the entire of the lumbar portion of the cord where these injections were administered. The few slices we did collect and perform IHC on from each mouse (for the purposes of targeting and expression validation alone) may prove insufficient to complete the kind of correlative analysis that the reviewer is proposing here, and any extra slices collected from these animals was unfortunately not saved. That said, we do not find the variability of the data present in **Figure 3e** to be as inconsistent for an assay such as the von Frey Up/Down test, or in response to the application of CNO, as the reviewer suggests. Indeed, previous publications that members of our lab have authored in which they conducted and analyzed similar assays (i.e. CNO administration to mice expressing chemogenetic receptors in the brain), have shown equal levels of variability in the response of the animals to the drug, and may underscore more the inherent variability across behavior assays that may result from differences in optimized doses of CNO and individual difference in the general temperament of test animals that may influence their behavior on a given assay or test day (Corder et al. Science, 2019. PMID: 30655440). The reviewer’s comment is well taken though, and while these data for our CNO dosing prior to von Frey testing may appear highly variable, it is our hope that the more robust and consistent behavioral results noted both across and within experimental animal groups for the hotplate and formalin hind paw injection studies may suggest less variability in the response of these animals to CNO than may have been suggested from the initial von Frey experiments. As these were the first experiments the animals were exposed to across a multiday series of testing, it is possible that underlying stress to the testing environment and to general manipulation by the experimenter may underlie the variability noted across mice. However, as this variability seemed to wane across future test days and experiments within the same animals for each group, it is possible that the animals were able to habituate to both the experimenter and general manipulation for each behavioral assay on these future tests, allowing the data collected to reflect changes more in line with response to CNO administration alone and nothing attributable to any remaining neophobia for the animals.

3. ***“In Fig 7 the authors show that mMORp viral transduction leads to expression of YFP in human iPSCd nociceptor like sensory neurons and not cardiomyocytes. This should be bench marked relative to endogenous MOR expression (for instance using immunocytochemistry or in situ). Although assumptions can be made that iPSCd nociceptor like neurons express MOR that should be shown (because iPSCs do not always fully capture the expression profile in adult human neurons), furthermore there may well be heterogeneity within this neuronal population and showing a relationship between mMORp driven expression and endogenous MOR would be even more convincing.”***

The reviewer raises an excellent point here, and draws attention to a set of experiments would strengthen the claims we asserted regarding higher levels of transduction of our AAV-PHP.s-*mMORp*-eYFP virus noted in iPSC nociceptor cells compared to cardiomyocytes. In an attempt to correct this, we worked with our collaborators to run both immunohistochemical and *in situ* hybridization assays on new cohorts of nociceptors and cardiomyocytes that we transduced with our AAV1-*hMORp*-eYFP viral construct to test for the expression and potential overlap of both MOR and eYFP signal. However, our attempts to do this either by staining for anti-MOR and anti-eGFP via IHC or staining for *EYFP* and *OPRM1* transcript via RNAscope were unsuccessful, despite running multiple rounds of both assays. For IHC studies, signal for anti-MOR staining proved difficult to resolve at the level of individual puncta on/around cell bodies/nuclei or more broadly in general. This is a drawback of staining for membrane bound receptors, and in particular the use of the anti-MOR Ab we have been working with specifically, that we have addressed elsewhere in our manuscript, and we were ultimately unable to find a way around this for IHC staining. For attempts at conducting ISH, we ran into similar issues with resolving signal for *OPRM1* transcript staining across both sets of cultures, as background/artefact staining and apparent bleed through of the signal for the *EYFP* transcript made determining if we were visualizing true *OPRM1* signal in both sets of cultures difficult. As these cells take multiple months to both culture, differentiate and grow from the base iPSCs into either nociceptors or cardiomyocytes, we decided to try another way to confirm that our nociceptor cultures do indeed express high levels of either the mu opioid receptor or the *OPRM1* gene. To do so, we conduct a gene expression assay using qPCR to examine the relative expression levels of *OPRM1* in both our nociceptor and cardiomyocyte cultures,

and present our findings in the new **Supplementary Figure 15**. We noted a robust increase in the relative gene expression of human *OPRM1* in differentiated nociceptor cells (51.8, n=3) when compared to cardiomyocytes (1, n=3), indicating that the nociceptors differentiated from the cell line utilized across these human iPSC studies appear to express more *OPRM1* than the differentiated cardiomyocytes. While we do not have a qualitative dataset by which to show overlap of signal for MOR or *OPRM1* gene or transcript expression within the same cells that were transduced *in vitro* with either of our AAV-PHP.s-*mMORp*-eYFP or AAV1-*hMORp*-eYFP viruses, we believe that the data presented in **Supp. Fig. 15** from the aforementioned qPCR analyses seem to validate that the nociceptor cell cultures utilized for our transduction studies presented in **Figure 7** do indeed expression greater levels of the mu opioid receptor gene at the very least. These results should also allow us to sufficiently claim that our viral constructs are able to not only transduce putative MOR/*OPRM1* promoter (+) human cell types, but also that our viruses show a similar level of selectivity for these MOR+ cells in human cultures that is comparable to the specificity that we have noted across multiple animal model organisms as well, as our gene express analyses showed cardiomyocyte cultures by contrast expressed little to no *OPRM1* at all while similarly showing little to no evidence of transduction by our viral constructs in culture.

New Supplemental Figure 15 Summary data of the relative gene expression for *OPRM1* in nociceptor and cardiomyocyte cells cultured and differentiated from human iPSC cells (LiPSC-GR1.1 cells). Expression across both cultured cell types is normalized to the house keeping gene GAPDH, and all normalized expression values are displayed as relative to the expression of *OPRM1* in the cardiomyocytes. Averaged, normalized expression of *OPRM1* in cultured nociceptors was found to trend higher than that of the cardiomyocytes (51.83, n=3 [nociceptors]; 1, n=3 [cardiomyocytes]; student's t-test: P=0.2215). Multiple wells of each cell type were combined to obtain enough cDNA to complete each run, with three separate runs conducted in total and final summary data being presented as the average of normalized values calculated for these three rounds of gene expression assays run on separate cohorts of cultured cells. All data in graphs are presented as averages ± the SEM.

4. ***“Generally the methods are well written and comprehensive however there is a lack of detail on the iPSC work. For instance, we know nothing about the donors. How may donors were used, ? healthy, ? sex. Was this the result of just one differentiation or were multiple differentiations performed? Were there checks on neuronal viability following viral transduction (in our hands PHPs evokes some toxicity in iPSCd neurons)?”***

We agree a more detail on the methodology behind the iPSC studies conducted in our manuscript are warranted, and we have revised the **Methods** section to include the following texts outlining more specific details regarding these studies and the cell lines utilized to conduct them:

“Briefly, LiPSC-GR1.1 cells (Lonza) served as the base cell iPSC (single donor, host sex: male, source: umbilical cord). iPSC cell-derived nociceptors were plated at 5,000/well density and co-cultured with 2,000 glial cells in 384-well plate, while iPSC cell-derived cardiomyocytes were plated 10,000/well without co-culture. Both nociceptor and cardiomyocyte derivations were the result of a single round of differentiation. Cultured cells were transduced by directly adding concentrated AAV viral particles at multiple titers (1×10^9 , 1×10^{10} , 1×10^{11} , 1×10^{12} gc/mL) into well plates (MOI= 2×10^5 , 2×10^6 , 2×10^7 , 2×10^8 [nociceptors] and 1×10^5 , 1×10^6 , 1×10^7 , 1×10^8 [cardiomyocytes])” (Lines 1524-1530).

5. ***“Reference below is lacking detail:
 Bohic, M. et al. Developmentally determined intersectional genetic strategies to dissect adult somatosensory circuit function Authors. (2022).”***

We have added the suggested reference to the manuscript and cited it within the text of the **Discussion** section as listed below:

“Examining the extent of the combinatorial approaches that can be taken with the use of our viruses with other transgenic animal lines and tools for the intersectional targeting of unique CNS and PNS cell populations (Bohic M. et al. bioRxiv. 2022, <https://doi.org/10.1101/2022.05.16.492127>) or the dissection of mu opioidergic circuits within the brain will thus require careful testing and planning when using our MORp constructs across any and all applications.” (Lines 509-513).

Change to the Author block:

In order to sufficiently address the reviewer’s comments, we have added six new authors to the manuscript who provided essential new data:

1. Lisa M. Wooldridge
2. Jessica A. Wojick
3. Amrith Rodrigues
4. F. Christopher Bennett
5. Mariko L. Bennett
6. Kate Townsend Creasy

In the Main Text we have changed the author block and author designations to include the new authors as follows:

Gregory J. Salimando^{1,2}, Sébastien Tremblay^{1,2}, Blake A. Kimmey^{1,2}, Jia Li³, Sophie Rogers^{1,2}, Jessica A. Wojick^{1,2}, Nora M. McCall^{1,2}, Lisa M. Wooldridge^{1,2}, Amrith Rodrigues⁴, Tito Borner^{1,5}, Kristin L. Gardiner⁶, Selwyn Jayaker⁷, Ilyas Singeç⁸, Clifford J. Woolf⁷, Matthew R. Hayes^{1,5}, Bart C. De Jonghe^{1,5}, F. Christopher Bennett^{1,9}, Mariko L. Bennett⁹, Julie A. Blendy¹⁰, Michael L. Platt^{1,2}, Kate Townsend Creasy^{4,5}, William R. Renthal³, Charu Ramakrishnan¹¹, Karl Deisseroth^{11,12,13,14*}, Gregory Corder^{1,2,*}

¹Dept. of Psychiatry, Perelman School of Medicine, University of Pennsylvania, Philadelphia, PA, USA

²Dept. of Neuroscience, Mahoney Institute for Neurosciences, Perelman School of Medicine, University of Pennsylvania, Philadelphia, PA, USA

³Dept. of Neurology, Brigham and Women’s Hospital and Harvard Medical School, Boston, MA, USA

⁴Translational Medicine and Human Genetics, Perelman School of Medicine, University of Pennsylvania, Philadelphia, PA, USA

⁵Dept. of Biobehavioral Health Sciences, School of Nursing, University of Pennsylvania, Philadelphia, PA, USA

⁶Dept. of Pathobiology, School of Veterinary Medicine, University of Pennsylvania, Philadelphia, PA, USA

⁷F.M. Kirby Neurobiology Center, Boston Children’s Hospital and Harvard Medical School, Boston, MA, USA

⁸National Center for Advancing Translational Science, National Institutes of Health, Bethesda, MD, USA

⁹Division of Neurology, Dept. of Pediatrics, Children’s Hospital of Philadelphia, Philadelphia, PA, USA

¹⁰Dept. of Systems Pharmacology & Translational Therapeutics, Perelman School of Medicine, University of Pennsylvania, Philadelphia, PA, USA

¹¹CNC Program, Stanford University, Stanford, CA, USA

¹²Dept. of Bioengineering, Stanford University, Stanford, CA, USA

¹³Howard Hughes Medical Institute, Stanford University, Stanford, CA, USA

¹⁴Dept. of Psychiatry & Behavioral Sciences, Stanford University, Stanford, CA, USA

Changes to Competing Interest:

In compliance with Nature Communications request for the disclosure of potential competing financial interests, we have included a statement regarding a recent provisional patent application four of the authors have submitted through their home institutions concerning the sequence design used to create the *mMORp1* and *hMORp1* constructs:

“G.C, K.D., C.R. and G.J.S. are listed as inventors on a provisional patent application filed on November 11th, 2022 through both the University of Pennsylvania and Stanford University regarding the custom sequences used to develop, and the applications of, both the *mMORp1* and *hMORp1* constructs (patent application number: 63/383,462 ‘Human and Murine *Oprm1* Promotes and Uses Thereof”).

REVIEWER COMMENTS

Reviewer #1 (Remarks to the Author):

These kinds of promoters or regulatory elements will allow for the probing of circuits and mechanisms in species other than mice. They also will allow for more tools to be introduced per animal including in mice.

The authors have improved the manuscript by including quantification of overlap in many brain areas as requested and by providing an image with better imaging quality to see the overlap (Fig 1f-m). I still find some images hard to discern with respect to overlap. For example Fig. 1e and Fig. 2- the overlap of the purple (neun) and green (mMORp1-eYFP) is not obvious in many of the images. I found Fig. 5f also particularly challenging. In the latter case, perhaps brightening the images would help. Lastly, Oprm1/mCherry in situ in spinal cord would help ascertain the specificity of the expression pattern of the mMORp1-hM4Di-mCherry construct used in behavior.

Otherwise the authors have done a nice job with the manuscript.

Reviewer #2 (Remarks to the Author):

The authors have addressed all my major and minor concerns to the best of their ability through supporting experiments, new data, text and figure revisions; and through their detailed well thought out response. I have no further comments or concerns, and I highly commend all the authors for the work that went behind revising this manuscript. I recommend this work for publication in N.com.

Reviewer #3 (Remarks to the Author):

The authors have addressed all my concerns, and I endorse the publication in its current form.

Reviewer #4 (Remarks to the Author):

The manuscript is improved especially in terms of the work on model organisms however I'm not convinced that the clarification/extra experiments that they provide on human iPSC derived nociceptors fully support their conclusions.

1. Although it's helpful that they have given more data on the methods used for iPSCs it is now clear that this is based on one differentiation of one iPSC line. I appreciate their point that these are long experiments which is why they have not replicated in a larger number of lines but in that case they need to acknowledge in the discussion that this is exploratory based on an n of 1.

2. The RT-PCR data does not fill me with confidence. They could not get antibody staining of the iPSC nociceptors nor was ISH successful. I agree this could be technical (or the alternative explanation is that OPRM1 is not expressed/expressed at a very low level). The authors therefore moved to RT-PCR and generate a normalised expression which they describe as showing robust expression in iPSC nociceptors VS cardiomyocytes however there is a high level of variance, this is a trend and it is not significant (ie. this does not meet my criteria for robust). They also mention that they have to pool multiple wells which would be unusual for measuring expression of a gene robustly expressed in these neurons. The problem with RT-PCR is that with enough cycles and relative expression results can be distorted/misleading. For instance what is the actual Ct for GAPDH and Ct for OPRM1 in these experiments? This would need to be given. The authors also have their own dataset that they should check for OPRM1 expression in the nociceptors they generate with this protocol: PMID: 37044067 has

RNAseq dataset over a time course from iPSCs through differentiation to nociceptors. This would at least provide independent verification of expression in the nociceptors. Can they please check this to see if there is robust expression of OPRM1 and benchmark against other pain genes (which are already shown in the paper)?

Overall this is a strong paper and my other concerns are allayed but I'd like to see the above clarified/acknowledged.

June 28, 2023

Responses to reviews.

We are writing here to follow up on additional comments provided by reviewers on our manuscript, '**Human *OPRM1* and murine *Oprm1* promoter driven viral constructs for genetic access to μ -opioidergic cell-types**' (NCOMMS-22-39191B). We would like to thank the reviewers for their continued interest in and support of our work as well. The comments provided on our previous resubmission were fair, insightful and we believe will help to improve upon the clarification of the data presented across the manuscript. We would also like to thank the reviewers for their kind words in regards to the previous round of revision work, and hope that we are able to address this next round of comments similarly to their satisfaction.

As before, we will address each of the major comments raised by the reviewers below. Reviewers' comments are paraphrased in **italic black font** with our responses in **blue font**, and any new text that has been added to the manuscript in **green font**. Additional analyses and images are provided within this response letter where applicable for ease of viewing. Overall, the reviewers' comments were taken and addressed directly within the text of the updated manuscript provided; and for the one comment that was not, we offer an explanation as to why we were unable to meet that request.

Here, we summarize the above mentioned analyses and updated images that we have generated to address the reviewers' comments, as well as to correct an images in one of the main text figures of the manuscript which was found to be incorrectly labeled upon further inspection. We hope you and the reviewers find these updates and responses to satisfactorily address any withstanding concerns and comments regarding the manuscript.

Summary of Updated Figures & Analyses Presented:

1. Figure 1f within the main text has been updated to include a corrected image in place of the one used in previous versions of the manuscript. The initial version was found to display a representative image of staining for **AAV1-hMORp1-eYFP** and anti-MOR within the central nucleus of the amygdala, and **not AAV1-mMORp1-eYFP**, as stated within the text and figure legend. This has been replaced in the revised manuscript with a new image taken of sections from a mouse brain in which the correct virus was injected into the CeA and then staining for both anti-MOR and amplified eYFP signal was performed.
2. Updated versions of the images from Figure 1e, Figure 2 and Figure 5f from the main text, with enhanced brightness for select signal channels in order to make overlapping signals more apparent. Figure 5 has also been updated to incorporate new quantification data for the overlap of mMORp-eYFP (i.e. *EYFP*) and *Oprm1* transcript positive DRG neurons that were collected over the course of generating new images with improved NeuN staining and signal.
3. An additional analysis of the NeuN staining displayed within Figure 1e in the main text has been provided to better clarify and demonstrate the overlap of the NeuN signal and signal for mMORp1-eYFP transduced cells. Representative images from Figure 2 have also been provided as additional means of demonstrating this overlap as well.
4. An updated version of Supplemental Figure 15 has been generated to incorporate additional data from our collaborator's paper, "Scalable generation of sensory neurons from human pluripotent stem cells" (Deng et al. 2023, PMID: 37044067), and their extensive transcriptomic analyses of the LiPSC-GR1.1 human stem cell line utilized in our manuscript. This includes transcriptomic data regarding the relative expression of *OPRM1* within the nociceptor-like cells Deng et al. were able to differentiate from this cell line (along with other relevant nociceptor-related genes) across multiple days in culture. Language in the text and figure legends regarding the stem cell work presented in our manuscript has also been updated to better reflect the nature of our own results examining *OPRM1* gene expression levels in these cells.

General Updates to Main Text Figures

Figure 1:

Updated image for Fig. 1f, corrected to a representative image of the overlap between anti-MOR signal and amplified eYFP signal from AAV1-mMORp1-eYFP virally transduced cells within the CeA instead of those transduced by the AAV1-hMORp1-eYFP virus (as incorrectly shown in previous versions of this figure).

Figure 5:

Updated analysis for Fig. 5g and the addition of a new graph to describe the total percentage of *Oprm1* (+) DRG neurons that were positively transduced with the AAV.PHP.s-mMORp-eYFP in comparison to those cells transduced by the virus that were not positive for *Oprm1* transcript in Fig. 5h. The analysis in Fig. 5g (quantification of total *Oprm1* [+] cells co-labeled for tdTomato or eYFP) has been updated to incorporate 3 new animals and 8 new ROIs (N=5, n=14 ROIs) for the control group injected with AAV.PHP.s-CAG-tdTomato virus, and 2 new animals and an additional 7 ROIs (N=4, n=11 ROIs) for the experimental group injected with AAV.PHP.s-mMORp-eYFP. A two tailed unpaired *t*-test, with Welch's correction, show significantly more *Oprm1*+ neurons to have been transduced with, and express the reporter for the AAV.PHP.s-mMORp-eYFP virus than the control AAV.PHP.s-CAG-tdTomato virus (Welsh's *t*-test, $P=0.0437$). All data are presented as individual values overlaying bar graphs, which describe the means of each dataset, \pm SEM ($*P<0.05$). Fig. 5h shows summary quantification of the percent total number of cells positive for eYFP transcript (i.e. positively transduced by the AAV.PHP.s-mMORp-eYFP virus) that were also either positive for *Oprm1* transcript (mMORp-eYFP/*Oprm1*+, 82.5%) or negative for *Oprm1* (mMORp-eYFP+ only, 17.5%) to demonstrate AAV.PHP.s-mMORp-eYFP specificity within mouse DRG neurons. The associated legend text for Figure 5 has been updated in the manuscript to reflect these changes as well, as have necessary details regarding quantification and statistical analyses in the Methods section.

A. Response to Reviewer #1:

1. "I still find some images hard to discern with respect to overlap. For example, Fig. 1e and Fig. 2 – the overlap of the purple (NeuN) and green (mMORp1-eYFP) is not obvious in many of the images. I found Fig. 5f also particularly challenging. In the latter case, perhaps brightening the images would help."

We'd like to thank the reviewer for bringing this to our attention, and for specifically pointing out a few figures in which some of the staining for our IHC and ISH work is still a bit difficult to discern for readers. While we had hoped that our previous round of adjustments were sufficient to show specifically the overlap of NeuN and mMORp1-eYFP in Figures 1 and 2, it is clear that some further adjustments are still necessary in order to make this more apparent. We have thus adjusted the brightness of the NeuN signal across all images displayed in Figure 1e and in Figure 2 to address this and provided these updated images below for the reviewer to examine.

In Fig. 2, we have also included a few markers within the merged images shown to indicate cells in which the signal for both the eYFP and NeuN can be more readily observed at the level of magnification used for these images in order to help readers appreciate what this overlap looks like due to the high signal expressed by the eYFP probe (and our subsequent enhancement of that signal with antibody amplification and staining) against the slightly weaker NeuN signal. We have also included an additional analysis of the NeuN and eYFP overlap within the overlay image from Figure 1e in which we have both greatly enhanced the brightness of the NeuN within this image, and presented the NeuN and eYFP signals in grey scale. This was done to demonstrate that, while the initial dimness of the NeuN signal may suggest that there is not as substantial of an overlap between NeuN stained nuclei and eYFP+ cells transduced by the mMORp1-eYFP virus, when the signal is enhanced within the raw image files, it is apparent that more or less the entire eYFP+ population of cells displayed within this representative image show an overlap of signal with the NeuN stain. Indeed, when quantifying NeuN+ and eYFP+ cells within this image, we show that out of the 85 total cells found to be positive for eYFP, only 1 was found to not overlap with NeuN signal. It is our hope that this additional analysis, and our brighten images for Figure 1 and 2, will satisfy the reviewers concerns about this discrepancy and make the overlap of signals presented within these images easier for readers to discern as well. Similarly, for Figure 5f, we have provided an updated version below in which we have enhanced the brightness of all signals (and adjusted some of the pseudo-coloring) as well.

Also, we suggest a partial reason for difficulty seeing the overlaying fluorescent channels may have resulted from the manuscript file that is shared with the reviewers, since this is a compressed PDF of our Word Document with embedded high-resolution TIFFs via the publisher's web portal PDF converter during upload; we would expect that in a final, published version we would be able to supply the copy editors with our Adobe Illustrator figure files with TIFF images, which should help with image clarity for readers. Lastly, we are confident that the morphology of the transduced, eYFP (+) cells shown in **Figure 1e** are distinctly neurons and not other cell-types, such as glia.

mMORp1 expression in neuronal vs microglial cell-types

Updated image set for Figure 1e, in which the brightness and contrast for signal across all channels have been adjusted slightly in the merged image in order to better accentuate the overlap between the NeuN and eYFP signals.

Updated images for Figure 2, in which brightness and contrast for signals across all channels have been adjusted in the merged images in order to better demonstrate the overlap between the NeuN and eYFP signals. Yellow arrow heads indicate representative cells in which this overlap can be better observed.

Updated images for Figure 5f, in which brightness and contrast for signals for all transcript probe staining has been adjusted and enhanced to better demonstrate the overlap of signals within the merged images. The mMORp-eYFP signal has also been pseudo-colored to red (as opposed to its previous cyan color) in order to provide better contrast against the color scheme selected for the NeuN, DAPI and Oprm1 signals.

Supplemental Analysis (not included in manuscript text):

a 20X images from Figure 1, Panel e in grayscale with adjusted white-point balance (e.g. brightness/contrast levels)

b 40X images in RGB with adjusted white-point and black-point balance

Supplemental analysis of anti-NeuN and AAV1-mMORp1-eYFP overlap in neural tissue. An additional analysis examining the overlap of signal for both NeuN staining and viral eYFP expression within tissue injected with our AAV1-mMORp1-eYFP virus, in this case using the prefrontal cortex (PFC) and our image from **Fig. 1e** as a representative region for demonstrating this overlap. Lower magnification images shown above in greyscale (**a**) display the number of NeuN positive nuclei (upper) and eYFP positive cell bodies (lower) that may not be resolved in raw images taken from the microscope, but which can be better visualized when adjusting brightness and contrast levels for these images. These adjustments better reveal both the greater number of cells that may be present in our image sets that are positive for both signals, and better demonstrate the overlap of these signals (particularly in areas where that overlap may not be apparent without signal adjustments). We demonstrate this further in higher magnification RBG images of the same region below (**b**) in which the white point and black point balances are adjusted to similarly increase the signal for dimmer NeuN and eYFP positive nuclei and cells, respectively. The middle row of images shows much more apparent overlap of these two signals when these balances are adjusted when compared to the upper raw image with minimal adjustments applied. Quantification of the total eYFP/NeuN positive cells (green dots) observed within this representative image is shown in the lower row, with a single eYFP+/NeuN- cell indicated by a red dot and arrow marker. Overall, this analysis demonstrates that with proper adjustments to the raw images taken of these sections on the microscope, that the overlap of both the NeuN and eYFP signals can be better resolved and appreciated, and that in contexts where the cell filling nature of the antibody staining enhanced eYFP signal may prevent this overlap from being readily seen, quantifying and overlaying counts for both eYFP+ and NeuN+ cells reveals a near identical overlap of each marker counted individually, indicating a high level of overlap.

2. “Lastly, *Oprm1/mCherry in situ* in spinal cord would help ascertain the specificity of the expression pattern of the *mMORp1-hM4Di-mCherry* construct used in behavior.”

Unfortunately, the spinal chemogenetics behavior experiment was conducted in September 2021 and we do not have remaining spinal cord tissue for FISH. In 2021, the spinal L3-L5 tissue was collected to confirm injection accuracy and viral spread and was thus sectioned at 50- μ m thick sections for immunohistochemistry, not for *in situ* hybridization applications (via the RNAscope methodology furnished by ACD Bio) that typically require 16-20 μ m thick sections. Any remaining tissue was likely disposed of over 1.5 years ago.

Thus, we have elected to forgo this request with the following justifications:

1. This is a new experimental request that is not connected to the initial review from November 2022, in which we addressed all requests from that review with new data and analyses over 6+ months of revisions.
2. We do not possess, nor we are aware of a validated RNAscope probe for mCherry that will work with our construct, and the process for discovery of this reagent would take an indeterminant amount of time. Multiple versions of a mCherry probe exist on ACD Bio’s RNAscope probe catalog (~\$500+ per probe), and thus may require multiple rounds of injections and FISH prior to finding a probe that works accurately for the detection of the fluorophore in tissue sections. Failure of any of these currently available probes would then require the request for a new probe to be generated (~\$1,500+ per new probe), and further injections and FISH testing after that.
3. If a probe could be found (not a guarantee), we estimate that it would take approximately 3+ months for this one FISH experiment (delivery of new mice, spinal intraparenchymal surgeries, viral incubation time, histology, imaging and analysis) and cost ~\$5,000+ (purchase of animals, RNAscope probes and reagents, technician hourly salary, slides).
4. Critically, in our revised manuscript, per the requests of Reviewer #1 in November 2022, we demonstrated and quantified MORp-driven transgene specificity in n=4 mouse brain regions (PFC, amygdala, striatum, VTA), and in n=4 model species (mouse, rat, shrew, rhesus macaque). Thus, our data strongly demonstrate the robustness and specificity of our tools to transduce endogenous *Oprm1* cell-types in the CNS. It is our hope that other researchers will use and further evaluate the appropriateness of these tools for their own experiments, including the use of the MORp AAVs for spinal cord-related research.

B. Response to Reviewer #4:

1. ***“Although its helpful that they have given more data on the Methods used for iPSCs it is now clear that this based on one differentiation of one iPSC line. I appreciate their point that these are long experiments which is why they have not replicated in a larger number of lines but in that case they need to acknowledge in the discussion that this is exploratory based on an n of 1.”***

The reviewer is absolutely correct in regards to their concerns about the language used to discuss these iPSC experiments within the text, and we agree that it does need to be modified in order to better reflect the exploratory nature of the use cases we present within the manuscript for utilizing our viruses to attempt to target and transduce putative MOR/*OPRM1* promoter positive human nociceptor or nociceptor-like cell populations. As such, we have adjusted the text within the **Results** and **Discussion** that directly address this dataset to make it more explicitly clear that what we are presenting within our final Main

Text figure (and its associated Supplementary Figures) are exploratory/first pass attempts to use these new viral tools to transduce populations of differentiated human cells type from a single iPSC line.

*Text added into the **Results** section of the manuscript in which we first discuss the findings from our human iPSC transduction experiments to make more explicit note of their exploratory nature and use of only a single cell line:*

“Lastly, we wanted to determine whether our *MORp* viruses would be able to provide genetic access to these same putative MOR+ cells within human derived model systems. To test this, we cultured and differentiated human induced pluripotent stem cells (iPSCs) from the LiPSC-GR1.1 lineage to produce either nociceptor-like neuronal cells or non-neuronal cardiomyocytes. Differentiated nociceptor-like cells from this iPSC lineage have previously been shown to possess gene expression profiles consistent with those of nociceptors and sensory neurons observed *in vivo* and to also show altered time courses for the expression levels of select genes across different days in culture, with *OPRM1* specifically reaching peak expression at day 21, and a significant decrease to occur by day 28 (**Supp. Fig. 15a**). When treating these cells with direct administrations of either our AAV-PHP.S-*mMORp-eYFP* or a control *CAG-tdTomato* virus at four different titers ($1 \times 10^9 - 10^{12}$ gc/mL) around days 21-28 in culture, we observed broad transduction of cells with both viruses in nociceptors that increased with titer concentration, with robust expression of both reporters noted at the 1×10^{12} gc/mL titer most prominently (**Fig. 7 a-d**)” (Lines 416 – 427).

*Text added to the **Discussion** section of the manuscript in order to make similar clarification of the nature of these iPSC-based experiments and how the reader should approach interpreting these results in turn:*

“Indeed, the broadening of opioid research models is further extended to *in vitro* culture systems, as suggested in our exploratory use case for human-differentiated iPSCs. Induced pluripotent stem cells from pain patients are currently under development as powerful *in vitro* disease models that provide unique exploration of nociceptor mechanisms of pain, including use in high-throughput screens for novel analgesics and as diagnostics to identify individuals at risk for transitioning from acute to chronic pain. The great majority of peripheral nociceptors express MOR, and as demonstrated by the success of our *MORp* constructs to transduce human iPSC differentiated nociceptors shown to express an apparent increase in gene expression for *OPRM1*, future uses could be leveraged to drive the expression of various gene editing tools or voltage indicators to be used in iPSCs for more complex phenotyping and screening studies. It is important to reiterate again, however, that our initial findings presented in these cultured human nociceptors only represent data acquired from the use of a single cell line shown to present an expression profile for genes such as *OPRM1* and several others indicative of a nociceptor and/or sensory-like cell types over the course of their development, with the expression of *OPRM1* examined in these cells shown to be somewhat variable at the time of collection for the samples used in our experiments. As such, while this does not detract from the clear evidence that *OPRM1* gene is present (and the *OPRM1* promoter active) within these cells, further study across other human-derived cell lines and/or differentiated cell types of both nociceptor and non-nociceptor lineage will be necessary to further demonstrated the utility of our viral tools for additional applications in such culture systems.” (Lines 531 – 548).

2. ***“The RT-PCR data does not fill me with confidence. They could not get antibody staining of the iPSC nociceptors nor was ISH successful. I agree this could be technical (or the alternative explanation is that OPRM1 is not expressed/expressed at a very low level). The authors therefore moved to RT-PCR and generate a normalized expression which they describe as showing robust expression in iPSC nociceptors VS cardiomyocytes, however there is a high level of variance, this is a trend and it is not significant (i.e. this does not meet my criteria for robust). They also mention that they have to pool multiple wells which would be unusual for measuring expression of a gene robustly expressed in these neurons. The problem with RT-PCR is that with enough cycles and relative expression results can be distorted/misleading. For instance what is the actual Ct for GAPDH and Ct for OPRM1 in these experiments? This would need to be given. The authors also have their own dataset that they should check for OPRM1 expression in the nociceptors they generate with this protocol: PMID: 37044067 has RNAseq dataset over a time course from iPSCs through differentiation to nociceptors. This would at least provide independent verification of expression in the nociceptors. Can they please check this to see if there is robust expression of OPRM1 and benchmark against other pain genes (which are already shown in the paper)?”***

The reviewer is once again correct to raise concerns about the language used to describe the results of our RT-PCR experiments when working with the nociceptor and cardiomyocyte cells we cultured/differentiated from the base LiPSC-GR1.1 cell line; we agree that the difference in expression should not be described as “robust”. We have adjusted the language within the **Results**, **Discussion**, and **figure legends** in which this RT-PCR dataset is discussed in order to remove that term and replace it instead with the term “trending”, which as the reviewer suggests, better describes the results obtained from this experiment. We have also provided the raw Ct/Cq values obtained from these RT-PCR experiments using the same differentiated nociceptors and cardiomyocytes below, as requested by the reviewer. These data suggest that there is greater *OPRM1* expression in iPSC nociceptors than in the cardiomyocytes.

The reviewer also raises an important point regarding the RNA-seq data from Deng et al. 2023. Indeed, our collaborators did screen for the expression of *OPRM1* within their previous RNA-seq dataset, and found it is most highly expressed in iPSC nociceptors at 21 days after differentiation (max normalized counts: ~60), and that they also noted that this expression declines precipitously by 28 days after differentiation. We provide this analysis in the revised manuscript as **Supplemental Figure 15a**, and can also mention that *OPRM1* is expressed more highly than *CALCA* (max normalized counts: ~35) in these cells, but less than *TRPV1* (max normalized counts: ~400), as seen in data presented in **Figure 5** in Deng et al. 2023 and reproduced briefly below. To allow sufficient time for viral expression, we analyzed iPSC nociceptors for RT-PCR analysis near 28 days after differentiation, which likely contributed to the low and variable expression of *ORPM1* we observed. Despite this, the data from Deng et al. clearly indicate *OPRM1* to be notably expressed within iPSC nociceptors differentiated from the LiPSC-GR1.1 cell line. It is thus our hope that the heightened levels of expression of other putative “pain” related genes shown within the same LiPSC-GR1.1 differentiated nociceptors from Deng et al. 2023, coupled with the additional data provided, can inspire further confidence as to the nature of the iPSCs we worked with and report on within our manuscript, and demonstrate *OPRM1* to indeed be notably present within these cells at the time points during which we were able to transduce and/or harvest them for analysis to generate the data presented in **Figure 7** in the main text and **Supplementary Figure 15**.

Updated text from the **Results** section (shown above as well), in which we have better clarified the trending, and non-significant, nature of the RT-PCR results we obtained when examining *OPRM1* expression in our cultured nociceptors compared with the cardiomyocytes:

“Differentiated nociceptor-like cells from this iPSC lineage have previously been shown to possess gene expression profiles consistent with those of nociceptors and sensory neurons observed in vivo and to also show altered time courses for the expression levels of select genes across different days in culture, with *OPRM1* specifically reaching peak expression at day 21, and a significant decrease to occur by day 28 (**Supp. Fig. 15a**). When treating these cells with direct administrations of either our AAV-PHP.S-mMORp-eYFP or a control CAG-tdTomato virus at four different titers (1×10^9 – 10^{12} gc/mL) around days 21-28 in culture, we observed broad transduction of cells with both viruses in nociceptors that increased with titer concentration, with robust expression of both reporters noted at the 1×10^{12} gc/mL titer most prominently (Fig. 7 a-d). By contrast, while expression of tdTomato signal remained prominent at both low and high viral titer concentrations in cultured cardiomyocytes, no eYFP signal was noted within these cultures following AAV-PHP.S-mMORp-eYFP treatment (Fig. 7e-h). These results indicate that our current iteration of MORp constructs appear to show selectivity for human cell types known to express MOR (and/or the *OPRM1* gene) when compared to cells shown to possess low expression of this receptor/gene, as demonstrated both from previous studies and our own gene expression analyses of these two cultures, which found nociceptor cells to indeed express higher trending levels of *OPRM1* than the cardiomyocyte cells in which our transduction experiments were performed (**Supp. Fig. 15b**)” (Lines 419 – 434).

Updated text from the **Discussion** section (shown above as well) in which we describe findings from Deng et al. 2023 relevant to *OPRM1* (and other putative pain gene) expression profiles within nociceptors differentiated from the LiPSC-GR1.1 cell line, and how this expression should be sufficient to allow our viruses to transduce these cultured cells more preferably over differentiated cardiomyocytes:

“It is important to reiterate again, however, that our initial findings presented in these cultured human nociceptors only represent data acquired from the use of a single cell line shown to present an expression profile for genes such as *OPRM1* and several others indicative of a nociceptor and/or sensory-like cell types over the course of their development, with the expression of *OPRM1* examined in these cells shown to be somewhat variable at the time of collection for the samples used in our experiments. As such, while this does not detract from the clear evidence that *OPRM1* gene is present (and the *OPRM1* promoter active) within these cells, further study across other human-derived cell lines and/or differentiated cell types of both nociceptor and non-nociceptor lineage will be necessary to further demonstrated the utility of our viral tools for additional applications in such culture systems.” (Lines 540 – 548).

Raw Ct/Cq Values from RT-PCR Experiments:

Nociceptors + Cardiomyocytes (mean Ct/Cq for GAPDH & OPRM1)

Well	Fluor	Target	Content	Sample	Cq	Cq Mean
D04	SYBR	GAPDH	Unkn	Noci	38.54	38.54
D05	SYBR	GAPDH	Unkn	Noci		0.00
D06	SYBR	GAPDH	Unkn	Noci		0.00
D07	SYBR	GAPDH	Unkn	Cardiomyocytes	17.85	17.85
D08	SYBR	GAPDH	Unkn	Cardiomyocytes	17.91	17.91
D09	SYBR	GAPDH	Unkn	Cardiomyocytes	17.75	17.75
H04	SYBR	hOPRM1-3	Unkn	Noci	39.57	39.57
H05	SYBR	hOPRM1-3	Unkn	Noci		0.00
H06	SYBR	hOPRM1-3	Unkn	Noci		0.00
H07	SYBR	hOPRM1-3	Unkn	Cardiomyocytes	23.43	23.43
H08	SYBR	hOPRM1-3	Unkn	Cardiomyocytes	23.35	23.35
H09	SYBR	hOPRM1-3	Unkn	Cardiomyocytes	23.49	23.49

Well	Fluor	Target	Content	Sample	Cq	Cq Mean
D10	SYBR	GAPDH	Unkn	hNoci-2	20.79	20.79
D11	SYBR	GAPDH	Unkn	hNoci-2	20.59	20.59
D12	SYBR	GAPDH	Unkn	hNoci-2	20.70	20.70
D13	SYBR	GAPDH	Unkn	hCardio-2	15.66	15.66
D14	SYBR	GAPDH	Unkn	hCardio-2	15.65	15.65
D15	SYBR	GAPDH	Unkn	hCardio-2	15.67	15.67
F10	SYBR	hOPRM1-3	Unkn	hNoci-2	22.54	22.54
F11	SYBR	hOPRM1-3	Unkn	hNoci-2	22.08	22.08
F12	SYBR	hOPRM1-3	Unkn	hNoci-2	22.11	22.11
F13	SYBR	hOPRM1-3	Unkn	hCardio-2	20.57	20.57
F14	SYBR	hOPRM1-3	Unkn	hCardio-2	20.72	20.72
F15	SYBR	hOPRM1-3	Unkn	hCardio-2	20.41	20.41

Well	Fluor	Target	Content	Sample	Cq	Cq Mean
D10	SYBR	GAPDH	Unkn	Noci-3	33.48	33.48
D11	SYBR	GAPDH	Unkn	Noci-3	32.74	32.74
D12	SYBR	GAPDH	Unkn	Noci-3	32.88	32.88
D13	SYBR	GAPDH	Unkn	Cardio-3	16.07	16.07
D14	SYBR	GAPDH	Unkn	Cardio-3	15.76	15.76
D15	SYBR	GAPDH	Unkn	Cardio-3	16.18	16.18
F10	SYBR	hOPRM1-3	Unkn	Noci-3	35.52	35.52
F11	SYBR	hOPRM1-3	Unkn	Noci-3	35.34	35.34
F12	SYBR	hOPRM1-3	Unkn	Noci-3		0.00
F13	SYBR	hOPRM1-3	Unkn	Cardio-3	25.29	25.29
F14	SYBR	hOPRM1-3	Unkn	Cardio-3	25.35	25.35
F15	SYBR	hOPRM1-3	Unkn	Cardio-3	25.35	25.35

Comparative Gene Expression Values from Deng et al. 2023 Dataset (not included in manuscript):

Gene expression levels for *OPRM1* benchmarked against additional nociceptor-associated genes in LiPSC-GR1.1 differentiated cells. Normalized gene expression data (presented as normalized counts) of *OPRM1*, *CALCA*, and *TRPV1* observed in differentiated nociceptor iPSCs across multiple time points (days) post-differentiation in culture, as reported in Deng et al. 2023 (*CALCA* and *TRPV1* data are reproduced from Figure 5 of this publication; see PMID: 37044067 for more details/information). Gradients in the expression profiles of all three genes can be seen to vary over time in culture, with notable decreases in both *OPRM1* and *TRPV1* observed over time in older nociceptor iPSC cultures, as well as relatively lower expression of *CALCA* even at 56 days in culture when compared with *OPRM1* and *TRPV1* expression profiles overall.

Figure Updates:

Supp. Fig. 15 | Cultured human nociceptors show evidence of gene expression for *OPRM1* in comparison to cultured human cardiomyocytes. Summary data of the relative gene expression for *OPRM1* in nociceptor and cardiomyocyte cells cultured and differentiated from human iPS cells (LiPSC-GR1.1 cells). **a**, Transcriptomic data demonstrating the relative expression level of *OPRM1* observed within nociceptor-like cells differentiated from the LiPSC-GR1.1 line. Expression levels (displayed as normalized transcript counts) for *OPRM1* are shown over a 28 days in culture time course, with cells collected and screened at discrete time points throughout. **b**, *OPRM1* relative expression from qPCR studies conducted with cultured cardiomyocyte and nociceptor cell types differentiated from the LiPSC-GR1.1 line. Expression is normalized to the house keeping gene GAPDH, and all normalized values are displayed as relative to the expression of *OPRM1* in the cardiomyocytes. Averaged, normalized expression of *OPRM1* in cultured nociceptors was found to trend higher than that of the cardiomyocytes (51.83, n=3 [nociceptors]; 1, n=3 [cardiomyocytes]; student's t-test: P=0.2215), indicating a basal level of expression of *OPRM1* in these cell types around the 21 day time point at which they were harvested. All data in graphs are presented as averages \pm the SEM (Lines 1389 - 1401).

With that, we would like to thank you once again for your time and consideration, and we look forward to hearing back on the suitability of our manuscript for publication.

Sincerely,

Gregory F. Corder, Ph.D.

Gregory J. Salimando, Ph.D.

REVIEWERS' COMMENTS

Reviewer #1 (Remarks to the Author):

The authors have satisfied my critiques.

Reviewer #4 (Remarks to the Author):

The authors have now provided further data on OPRM1 expression in iPSC nociceptors which is helpful and they have revised the text of the results and discussion which accurately reflects the data. My concerns are now fully addressed.